# Gut microbiota produces biofilm-associated amyloids with potential for neurodegeneration

Ariadna Fernández-Calvet [1,11], Leticia Matilla-Cuenca[1,11], María Izco [2], Susanna Navarro [3], Miriam Serrano[1], Salvador Ventura [3], Javier Blesa [4,5], Maite Herráiz[6,7], Gorka Alkorta-Aranburu[7,8], Sergio Galera[9], Igor Ruiz de los Mozos [9], María Luisa Mansego [10], Alejandro Toledo-Arana [1], Lydia Alvarez-Erviti [2] & Jaione Valle [1] ✉

Age-related neurodegenerative diseases involving amyloid aggregation remain one of the biggest challenges of modern medicine. Alterations in the gastrointestinal microbiome play an active role in the aetiology of neurological disorders. Here, we dissect the amyloidogenic properties of biofilm-associated proteins (BAPs) of the gut microbiota and their implications for synucleinopathies. We demonstrate that BAPs are naturally assembled as amyloid-like fibrils in insoluble fractions isolated from the human gut microbiota. We show that BAP genes are part of the accessory genomes, revealing microbiome variability. Remarkably, the abundance of certain BAP genes in the gut microbiome is correlated with Parkinson's disease (PD) incidence. Using cultured dopaminergic neurons and *Caenorhabditis elegans* models, we report that BAP-derived amyloids induce α-synuclein aggregation. Our results show that the chaperone-mediated autophagy is compromised by BAP amyloids. Indeed, inoculation of BAP fibrils into the brains of wild-type mice promote key pathological features of PD. Therefore, our findings establish the use of BAP amyloids as potential targets and biomarkers of α-synucleinopathies.

The microbiota of the gastrointestinal (GI) tract contains the largest reservoir of microbes (~1000 different microbial species) and constitutes the most abundant biofilm in the human body[1]. In biofilms, bacterial cells adhere to a substrate and secrete a self-produced extracellular polymeric matrix (ECM) that provides protection against environmental and host insults[2]. Functional amyloids are structural components of the ECM in diverse bacteria[3–5]. The well-

ordered β-strand structure of amyloid fibrils provides resistance to stressful conditions. In addition, the polymerization of amyloidogenic proteins is an energy-efficient strategy for constructing the ECM since it occurs through a nucleation-dependent self-assembly process[6]. An increasing number of amyloids have been identified, purified, and studied as components of the biofilm matrix[5]. However, data on the composition of amyloids that make up the biofilms of

[1]Instituto de Agrobiotecnología (IDAB). CSIC-Gobierno de Navarra, Avenida Pamplona 123, Mutilva 31192, Spain. [2]Laboratory of Molecular Neurobiology, Center for Biomedical Research of La Rioja, Logroño, Spain. [3]Institut de Biotecnologia i de Biomedicina and Departament de Bioquimica i Biologia Molecular, Universitat Autónoma de Barcelona, Bellaterra, Spain. [4]HM CINAC (Centro Integral de Neurociencias Abarca Campal), Hospital Universitario HM Puerta del Sur, HM Hospitales, Madrid, Spain. [5]Instituto de Investigación Sanitaria, HM Hospitales, Madrid, Spain. [6]Department of Gastroenterology, Clínica Universitaria and Medical School, University of Navarra, Navarra, Spain. [7]IdiSNA, Instituto de Investigación Sanitaria de Navarra, Pamplona, Spain. [8]CIMA LAB Diagnostics, University of Navarra, Pamplona, Spain. [9]Department of Personalized Medicine, NASERTIC, Government of Navarra, Pamplona, Spain. [10]Translational Bioinformatics Unit, Navarrabiomed, Complejo Hospitalario de Navarra (CHN), Universidad Pública de Navarra (UPNA), IdiSNA, Pamplona, Spain. [11]These authors contributed equally: Ariadna Fernández-Calvet, Leticia Matilla-Cuenca. ✉e-mail: jaione.valle@csic.es

enteric bacteria are lacking, and their biological roles in health and disease are still unknown. The curli fimbriae are among the best-characterized bacterial amyloid systems produced by enteric bacteria. A dedicated fiber assembly machinery encoded by the *csgBAC-csgDEFG* genes is required for the tight control of the secretion and polymerization of the protein subunits (CsgA) into amyloid fibers[7–9]. The availability of complete bacterial genomes shows that sequences homologous to the *csg* genes are widely distributed among bacteria from the most representative members of the GI microbiota, including Enterobacteriaceae, Proteobacteria, Bacteroidetes, and Firmicutes[10,11]. However, the presence of curli in enteric biofilms is controversial and only indirect evidence, such as the presence of antibodies against curli detected after infection, supports this statement[11–13]. A less complex model of cell surface proteins with amyloidogenic domains, named facultative amyloids, has been described in enteric pathobionts. Bap from *Staphylococcus aureus* and Esp from *Enterococcus faecalis* are prototypical facultative amyloids[14–16]. These proteins contain short stretches with amyloidogenic potential in the N-terminal domain. Upon proteolytic cleavage, the N-terminal region switches to a β-sheet-rich conformation and polymerizes under acidic conditions to form fibrillar structures that promote biofilm formation[15–18]. Bap and Esp belong to a family of proteins named biofilm-associated proteins (BAPs). BAPs are surface proteins with high molecular weights. These proteins contain a core domain of tandem repeats. BAPs are widely distributed in Gram-positive and Gram-negative bacteria and play relevant roles in biofilm formation and virulence[19–21]. It is therefore tempting to speculate that the gut microbiota expresses BAPs with amyloid-like properties as structural components of the enteric biofilm.

Recent advances in determining amyloid structure have revealed a high diversity of fiber architectures[22]. However, many features are shared by most amyloid structures, which opens the possibility to promiscuous interspecies interactions among amyloids. The exposure of the host to the enteric amylome may provide an opportunity to induce disease through mechanisms of molecular mimicry. Recent evidence has demonstrated the influence of the GI microbiota and its alteration (dysbiosis) on brain metabolism and functioning[23]. Furthermore, and even more surprisingly, dysbiosis has been implicated in the etiopathology of neurodegenerative disorders such as Alzheimer's disease (AD), Parkinson's disease (PD), amyotrophic lateral sclerosis (ALS) and prion diseases[24–27].

All of these neurodegenerative diseases commonly exhibit progressive neuronal loss and dysfunction accompanied by the deposition of abnormally processed or misfolded proteins (amyloids): alpha-synuclein (α-Syn) in PD patients, amyloid-β (Aβ) and hyperphosphorylated Tau in AD patients and TDP-43 in ALS patients. The causative agent or agents responsible for initiating protein misfolding and the molecular basis of pathogenesis remain to be fully elucidated.

Herein, using a multidisciplinary approach, we investigated the distribution of BAPs in the gut microbiota and their putative impact on neurodegenerative diseases. We performed a series of in silico, biochemical, biophysical, and genetic experiments on a subset of identified BAPs to validate their capacity to form amyloids. We also provide evidence for the presence of BAP-related amyloid aggregates in the human fecal microbiota. Using in vitro and in vivo models of α-synucleinopathy, we demonstrated that BAP-amyloidogenic domains induce α-Syn aggregation and promote key pathological features of PD. Finally, the reanalysis of large-scale metagenomic data from control and PD gut microbiomes revealed a correlation between the presence of BAP-encoding genes and PD, suggesting that BAPs may contribute to the mechanisms of PD.

## Results

### BAP-like genes are part of the accessory genomes of the microbiome

BAPs have been shown to produce amyloidogenic peptides[15,16]. We hypothesized that the presence of BAP-like genes in the human microbiota could be a source of amyloids with potential implications for human health. To evaluate the presence of BAPs in the human gut microbiota, we performed a local BLASTP search against the human gut database[28] using the *S. aureus* Bap sequence as a query, which represents the protein model of the BAP family. The results led to the identification of 30 proteins with identities ranging from 13 to 56% to *S. aureus* Bap (Table S1). Interestingly, 19 of these 30 proteins fulfilled the criteria for high molecular weight surface proteins and for the presence of repeat domains, which are parameters that define the BAP family (Fig. 1a). We found BAP orthologues in members of the order Lactobacillales (*Lactococcus lactis*, *Lactobacillus acidophilus*, *L. johnsonii*, *Enterococcus faecalis*, *E. faecium*, *Streptococcus salivarius*), Enterobacterales (*Escherichia coli*, *E. fergusonii*, *Citrobacter freundii*, *Enterobacter hormaechei*, *Chania multitudinisentens*, *Providencia alcalifaciens*), and Bacillales (*S. hyicus*, *Terribacillus goriensis*), among the most abundant.

Although BLASTP analysis confirmed the presence of BAPs in the human microbiota, the abundance and distribution of BAP-encoding genes in the human population cannot be inferred. Therefore, we performed shotgun sequencing to quantify BAP-like genes in human fecal samples available from an ongoing cohort study of healthy individuals and irritable bowel syndrome patients (IBS) (Table S2 and Table S3). Figure 1b shows that 11 different BAP genes were present in 49 human samples, although the bacterial species that should harbor these genes were more widely represented (Fig. 1c). These findings suggest that BAP genes may be part of the accessory genome, which is consistent with the findings of previous studies showing that both the *bap* and *esp* genes are encoded by pathogenicity islands that are present in certain isolates of *S. aureus* and *E. faecalis*, respectively[20]. To confirm that the remaining BAP-like genes belonged to the accessory gene repertoire, we used a pangenomic approach to map BAP-genes to the core or accessory genome of the corresponding bacterial species. For this purpose, we used the MGnify platform, which includes biome-specific collections of metagenomic-assembled and isolated genomes and their corresponding pangenome analyses (https://www.ebi.ac.uk/metagenomics)[29]. We focused on the Unified Human Gastrointestinal Genome (UHGG) v2.0.1 biome[30] to determine the presence/absence of BAP-like genes among all the genomes present in the corresponding species cluster. The presence/absence binary matrix of the bacterial species shown in the Supplementary Dataset was queried with the corresponding BAP-like gene reference. The percentage of strains carrying a particular BAP-like gene in a specific species cluster was subsequently calculated. Figure S1 shows that the BAP-like genes of all the analyzed species, as well as *E. faecalis*, are encoded in the accessory genomes of the gut microbiota. BAP-like genes from Gram-positive species are less represented than those from Gram-negative species. Taken together, these results demonstrate that BAP-like genes are encoded in the accessory genome of bacterial species inhabiting the human gut.

### BAP domains form amyloid-like structures that mediate biofilm formation

To predict the amyloidogenic propensity of the BAP orthologues from the microbiota, we performed a stringent computational analysis using five well-known algorithms, namely WALTZ, AGGRESCAN, TANGO, MetAmyl, and FoldAmyloid[31–35]. Potential amyloidogenic regions predicted by at least four of the algorithms were considered for further analysis. We found that BAPs had domains whose properties are compatible with an amyloid nature (Fig. 2a and Table S4). We used the

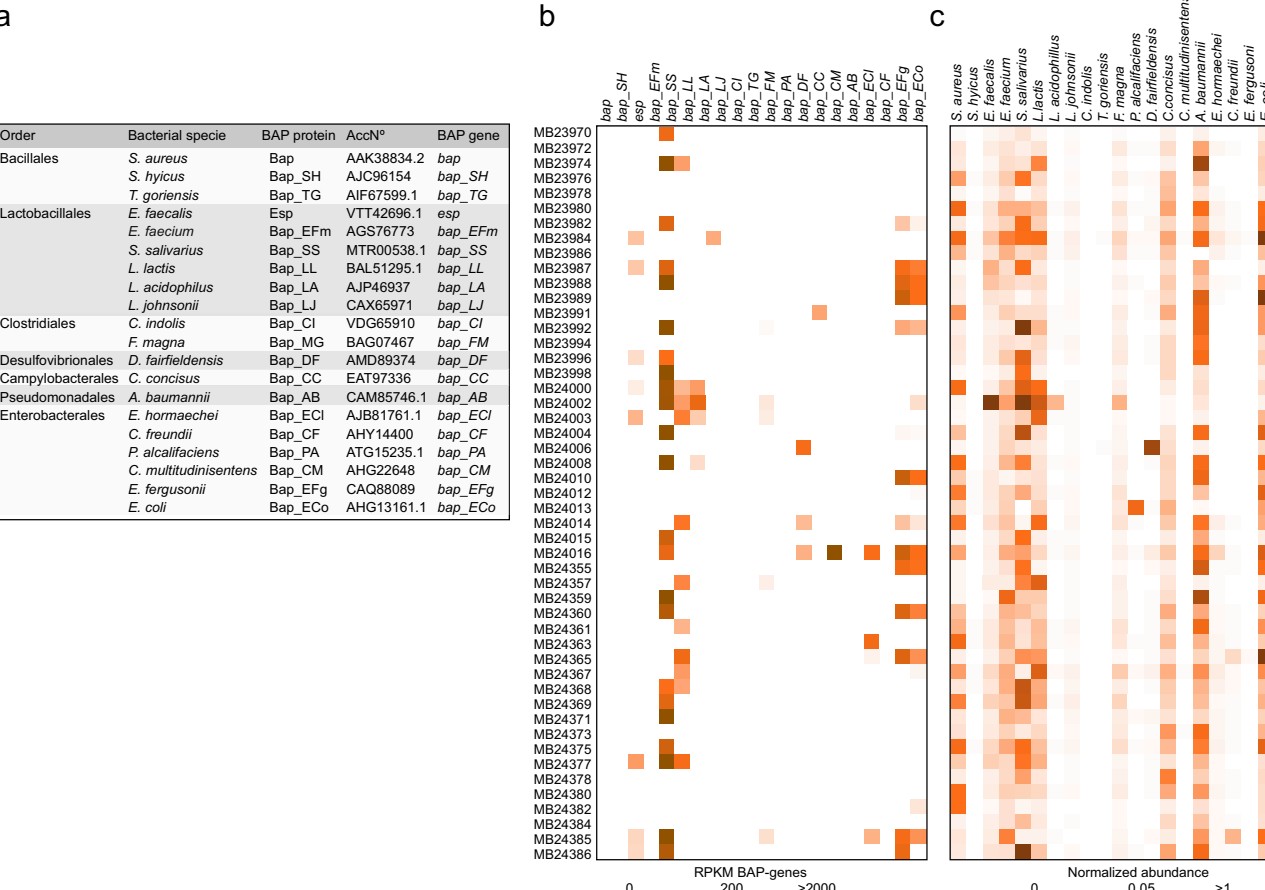

**Fig. 1 | BAP-like genes are part of the accessory genomes of the microbiome.**
**a** BAPs from gut microbiota. **b** Reads per kilobase of BAP-coding genes per million mapped reads (RPKM) among a cohort including healthy individuals: MB23986, MB23991, MB24012, MB24013, MB24014, MB24015, MB24359, MB24360, MB24367, MB24368, MB24377, MB24384, MB24385, MB24386 and IBS patients: MB23970, MB23972, MB23974, MB23976, MB23978, MB23980, MB23982,
MB23984, MB23987, MB23988, MB23989, MB23992, MB23994, MB23996, MB23998, MB24000, MB24002, MB24003, MB24004, MB24006, MB24008, MB24010, MB24016, MB24355, MB24357, MB24361, MB24363, MB24365, MB24369, MB24371, MB24373, MB24375, MB24378, MB24380, MB24382.
**c** Heatmaps showing the normalized abundance of taxon at level of species.

C-DAG system to confirm the amyloidogenicity of the predicted domains[36]. We cloned 22 BAP-derived domains, including the most representative domain of each protein, which were fused to the signal sequence of CsgA in the pExport plasmid (Fig. 2a and Table S5). The plasmids used were subsequently introduced into the curli deficient *E. coli* VS39, which expresses the *csgG* gene required for chimera export. We also included the *S. aureus* Bap_B and *E. faecalis* Esp_N domains as the prototypical amyloidogenic domains and the *S. aureus* Bap_A and *E. faecalis* Esp_C domains which did not aggregate[16]. To analyze the presence of extracellular amyloid-like aggregates, we determined the ability of the strains expressing BAP-derived domains to bind Congo red (CR) dye (Fig. 2b). Quantification of CR bound to the cells using a quantitative CR binding assay showed that 19 of 26 *E. coli* strains expressing BAP-derived domains bound significantly greater amounts of CR than did *E. coli* without the pExport plasmid (Fig. S2a). To further confirm the presence of amyloid-like fibers, we extensively analyzed the ultrastructure of the extracellular material by transmission electron microscopy (TEM). As shown in Fig. 2c, fiber-like extracellular material was associated with CR-positive strains (Fig. 2c), which reacted specifically with gold-labeled anti-His antibody (Fig. S2b). In contrast, no fibrillar structures were detected in the CR-negative strain Bap_CM2 (Fig. 2c).

To elucidate the role of the amyloidogenic domains in biofilm formation, we constructed chimeric proteins comprising some of the predicted amyloid domains and the carboxy terminal domain of clumping factor A (R_ClfA) of *S. aureus*, which contains the LPXTG motif for anchoring to peptidoglycan. The chimeric proteins were expressed in a biofilm-negative background (*S. aureus* Δ*bap*). The results showed that the heterologous expression of a chimera in *S. aureus* Δ*bap* induced biofilm formation (Fig. S3). Overall, we concluded that BAP orthologues from the gut microbiota carry domains with amyloidogenic properties that could mediate biofilm formation when expressed in the gut intestinal tract.

## BAP domains assemble into fibrillar β-sheet-rich structures with amyloid-like properties

To experimentally demonstrate the amyloid-like properties of individual BAP domains, we selected the most promising candidates (those showing statistically significant CR-binding, *p*-values < 0.01) to generate the corresponding recombinant domains (Fig. 2a and Table S6). As a negative control, we included the Bap_CM2 domain which does not bind CR. Purified recombinant soluble proteins were incubated in acidic and neutral pH buffers and the formation of macromolecular structures was analyzed via light scattering[37]. An increase in the light scattering signal was observed for all the recombinant proteins under at least one of the conditions tested except for rBap_TG and rBap_CM2 where small or undetectable

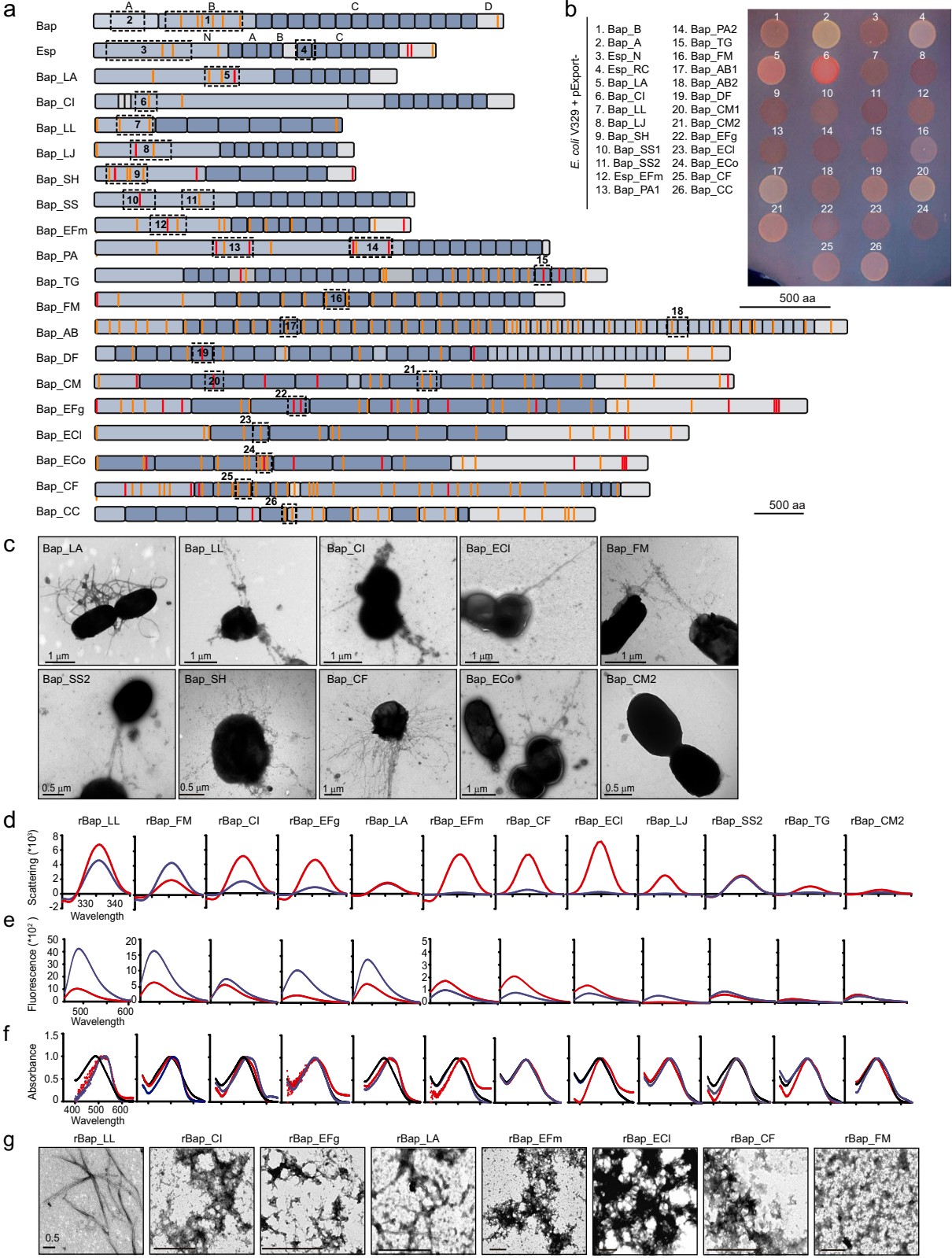

**Fig. 2 | BAP domains form amyloid-like structures. a** Schematic representation of BAPs. N-terminal domain (A and B), core C-repeats and the carboxy-terminal domain (D). Color lines represent the amino acid stretches predicted to be amyloidogenic by five (red) or four (orange) of the algorithms used. Dash boxes represent the amyloidogenic regions that were expressed at C-DAG system. **b** Congo red binding of *E. coli* that export amyloid domains. Bap_B (1) and Esp_N (3) were used as positive controls. Bap_A (2) and Esp_RC (4) were used as negative controls. **c** Transmission electron micrographs of negatively stained fiber-like structures. Images from representative experiments are shown. Synchronous light scattering (**d**), Th-T emission fluorescence (**e**) and CR absorbance spectrum (**f**) of amyloid domains samples resuspended in phosphate–citrate buffer either at pH 7.0 (blue) and 4.5 (red). Free CR absorbance is represented in black. **g** Representative transmission electron micrograph of amyloid domains after incubation at pH 4.5 are shown. Scale bars, 0.5 μm. Source data are provided as a Source Data file.

aggregates were obtained (Fig. 2d). The secondary structure content of the aggregates was assessed by recording FTIR spectra in the amide I region (1700–1600 cm$^{-1}$). Deconvolution of the spectra indicated that, in most cases, the signal was dominated by a band occurring at 1620–1630 cm$^{-1}$, attributed to the presence of intermolecular β-sheets (Fig. S4). This signal was the largest contributor to the absorbance spectrum in most of the aggregates at pH 4.5 (Fig. S4a, b). In the case of rBap_LL, rBap_FM, and rBap_CI, a band at ~1625 cm$^{-1}$ also dominates the spectrum at neutral pH. Interestingly, an antiparallel β–sheet band (~1690 cm$^{-1}$) was detected only for rBap_CF, rBap_ECl and rBap_LA, suggesting that the detected β–strands tended to adopt a parallel distribution (Fig. S4). Overall, our data are consistent with the assembly of BAP-derived amyloidogenic domains into supramolecular β-sheet-enriched structures. We then used the amyloid-indicating dyes Thioflavin-T (Th-T) and CR to confirm the amyloid nature of the detected aggregates. A strong Th-T fluorescence signal was observed for the rBap_LL, rBap_FM, rBap_CI, rBap_EFg, and rBap_LA domains after incubated under neutral and acidic conditions (Fig. 2e). On the other hand, rBap_EFm, rBap_CF and rBap_ECl showed slight increases in Th-T fluorescence emission intensity when incubated under acidic conditions. The CR binding to amyloid-like aggregates was determined by analyzing the absorbance spectra recorded between 380 and 680 nm. rBap_LL, rBap_FM, rBap_CI and rBap_LA exhibited a redshift to ~540 nm in their spectra at both neutral and acidic pH values (Fig. 2f). On the other hand, rBap_EFm, rBap_EFg and rBap_ECl exhibited CR binding only when incubated at acidic pH (Fig. 2f). These results are consistent with the Th-T fluorescence signal emission. A slight redshift in the spectrum was observed for rBap_LJ under neutral conditions. Finally, the morphological features of the amyloid-like aggregates were examined using TEM. Negative staining revealed that rBap_LL, rBap_FM, rBap_CI and rBap_LA assemble into supramolecular structures with fibrillar features under both, neutral and acidic pH conditions (Fig. 2g and Fig. S4c). The length of the fibrils ranged from 3 μm (rBap_LL and rBap_LA) to 0.2 μm (rBap_EFg). The remaining proteins essentially exhibited protofibrillar forms at acidic pH (Fig. 2g), except for rBap_SS2, rBap_LJ, and rBap_CM2, where no fibrillar structures were detected

(Fig. S4d). In summary, the results indicate that the formation of fibrillar and protofibrillar amyloid-like structures is an intrinsic feature of most BAP-derived domains. The BAP domains formed by rBap_LJ, rBap_SS2 and rBap_TG, did not exhibit amyloid characteristics, at least under the assayed conditions.

## BAP amyloid-like structures are present in the human fecal microbiota

Having demonstrated that recombinant BAPs form amyloid-like fibrils, we wondered whether these amyloid structures could be naturally detected in the human gut. Therefore, fecal samples from the same cohort used for metagenomic analyses were subjected to a density gradient and differential centrifugation to isolate the fecal microbiota and subsequently purify its insoluble protein fraction. Taking the advantage of the insolubility of amyloid fibers in detergents, we incubated the samples several times in 1% SDS to improve their purity. Then, we used a filter retention assay to test for the presence of SDS-stable amyloids. The cellulose acetate membrane retains amyloid aggregates, whereas soluble proteins pass through the filter[38]. Immunological detection using antibodies against BAPs (Esp_N from *E. faecalis*, Bap_B from *S. aureus*, Bap_LL from *L. lactis* and Bap_LA from *L. acidophilus*) allowed the detection of retained amyloid-like fibers of Esp in 26% of the samples (8/30), Bap_LL in 3% of the samples (1/30) and Bap_LA in 6% of the samples (2/30) (Fig. 3 and Fig. S5a). As expected, Bap_B could not be detected in the human gut microbiota samples. The specificity of the amyloid signal was determined by testing the susceptibility of BAP-positive samples to formic acid and trifluoroacetic acid/hexafluoroisopropanol treatments (Fig. S5b) and using fecal samples from mice in which the gut microbiota was depleted with a broad-spectrum antibiotic treatment (Fig. S5c). We also showed that the majority of the BAP-positive amyloid samples also reacted with the anti-amyloid oligomer (A11) antibody that recognizes transient folding species, including CsgA oligomers[39], and/or the anti-amyloid fibril OC antibody, which recognizes an epitope that is common to amyloid fibrils and that is absent from prefibrillar oligomers (Fig. 3 and Fig. S5d). All these results showed the presence of BAP structures with amyloidogenic properties in the insoluble fraction of the human microbiota.

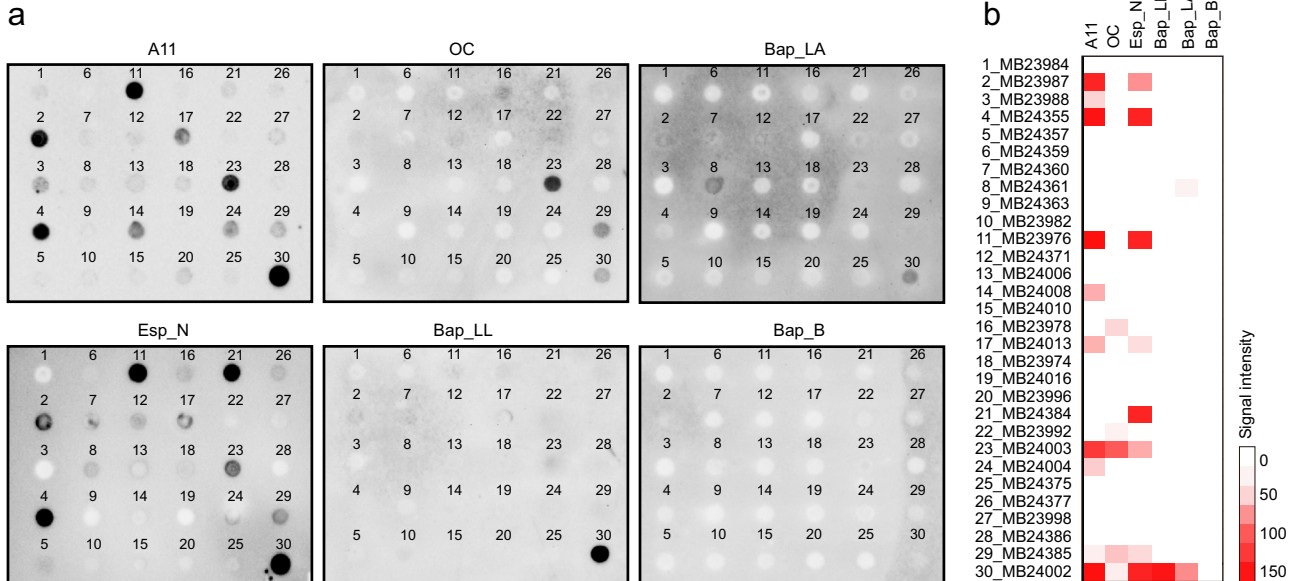

**Fig. 3 | BAP amyloid-like structures are present in the human fecal microbiota. a** Detection of amyloids in human fecal samples using filter trap and dot-blot assays. SDS-resistant amyloids were subjected to vacuum filtration through a 96-well dot blot apparatus with an acetate cellulose membrane. Immunoblotting was then performed with 4 antibodies against BAPs: Esp_N, Bap_LL, Bap_B, Bap_LA. Anti-amyloid oligomers (A11) and anti-amyloid fibrils OC antibodies were used to detect amyloid conformation. **b** Heatmap showing the relative signal intensity of the blots determined by densitometry using Image Lab Software.

## BAP-derived amyloidogenic domains promote human α-Syn aggregation in *Caenorhabditis elegans* muscle model of PD

Given that domains from BAPs form amyloid-like structures, we investigated whether these domains can heterologously induce α-Syn aggregation. We used the nematode *C. elegans* NL5901 as a model of PD[40]. We screened BAP-related amyloid domains for their effects on PD pathogenesis by feeding *C. elegans* NL5901 with individual *E. coli* VS39 strains from the C-DAG system (Fig. 2b) and assessing α-Syn aggregation. Quantification of the number of α-Syn aggregates showed that the majority of *C. elegans* fed BAP-derived amyloid-like fibers had a significantly greater number of visible α-Syn aggregates than animals fed *E. coli* VS39 did ($*p < 0.05$; $**p < 0.01$; $***p < 0.001$) (Fig. 4a, b). Interestingly, nematodes fed *E. coli* expressing the nonamyloid domain Bap_CM2 did not show an increase in the number of α-Syn aggregates. In addition, we performed locomotion assays in liquid media to test whether the increase in α-Syn aggregates affected the fitness of the animals. By measuring *C. elegans* thrashing (number of body bends per unit of time[41]), we showed that the majority (70%) of nematodes that accumulate α-Syn aggregates have impaired motility compared to animals fed *E. coli* VS39 (Fig. 4c). To limit the effect of the BAPs on amyloid properties and to avoid any bias related to the bacterial background, we supplemented regular feeding plates of *C. elegans* NL5901 containing the nonpathogenic *E. coli* strain OP50 with preformed fibrils (PFFs) from the prototypical BAP amyloids Bap_B and Esp_N (rBapB-PFF and rEspN-PFF), the monomeric Bap_B and Esp_N proteins (rBapB-MON, and rEspN-MON) and the nonamyloid domain rBap_CM2. Analysis of *C. elegans* NL5901 by fluorescence microscopy revealed a significant increase in the number of α-Syn aggregates in animals supplemented with preformed BAP amyloids but not in nematodes fed rBapB-MON, rEspN-MON and rBap_CM2 ($***p < 0.001$; $*p < 0.05$) (Fig. 4d, e). Consistent with the immunofluorescence results, biochemical analysis revealed increased levels of the α-Syn protein in nematodes fed BAP amyloid-like fibers (Fig. 4f and Fig. S6a). Remarkably, the α-Syn RNA expression level was not affected (Fig. S6b). Taken together, these results demonstrated that BAP amyloids from the gut microbiota induce α-Syn aggregation in the *C. elegans* muscle model of PD.

## Prototypical BAP amyloids induced the formation of intracellular α-syn aggregates in human neuronal cells

To investigate whether BAP-like amyloids can directly influence α-syn aggregation in a cross-seeding process, we incubated seeds generated

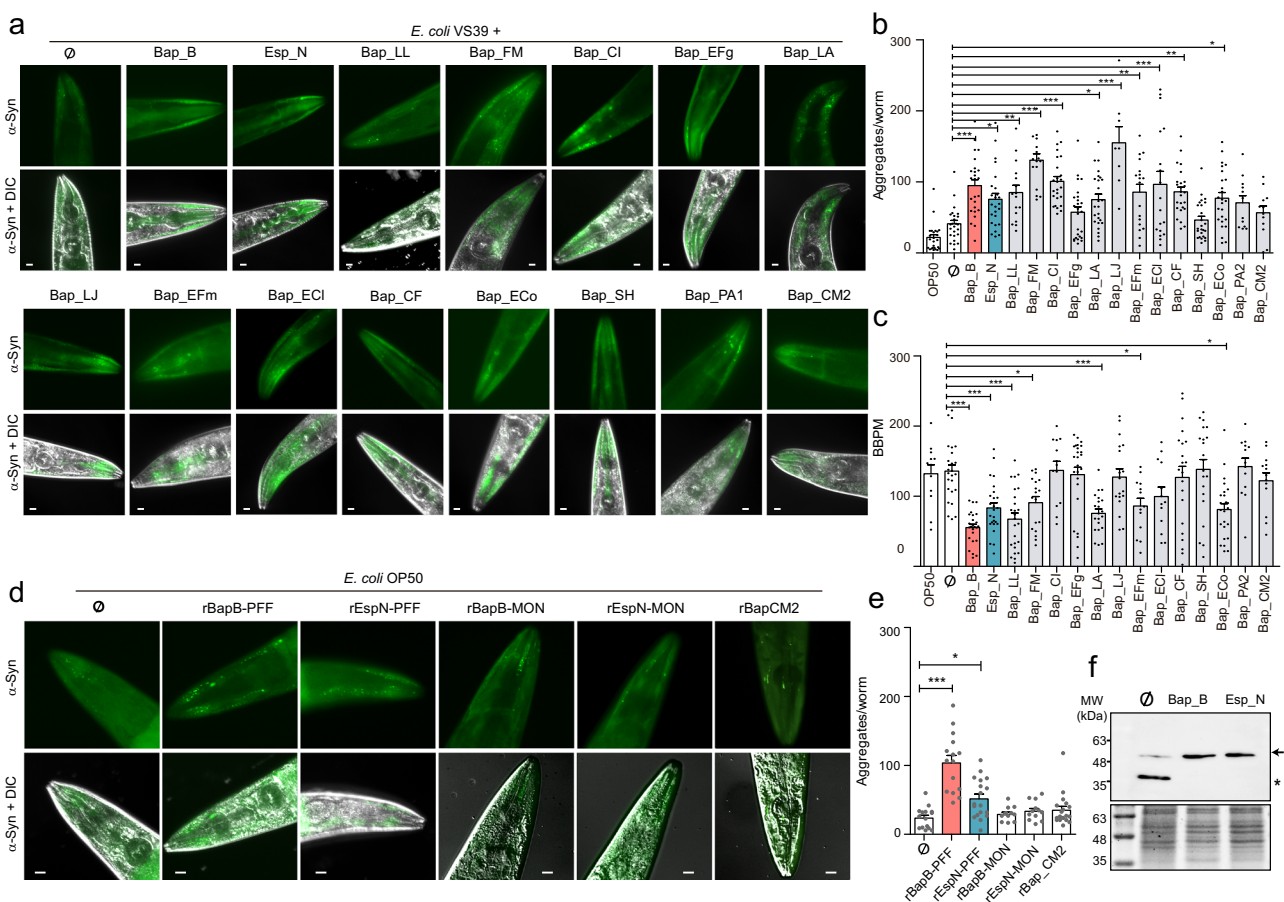

**Fig. 4 | BAP-derived amyloidogenic domains promote human α-Syn aggregation in *C. elegans* model of PD. a** Representative images of α-Syn muscle aggregates obtained by fluorescence microscopy in the head of *C. elegans* NL5901 fed *E. coli* VS39 expressing BAP amyloids. Scale bars, 10 μm. **b** Quantification of α-Syn muscle inclusions per worm. $N = 25$ worms per condition except for Bap_EFm, Bap_ECl ($N = 18$); Bap_FM, Bap_LL ($N = 16$); Bap_LJ, Bap_PA, Bap_CM2 ($N = 12$). **c** Worm-thrashing representation as the number of bends per minute (BBPM). $N = 25$ worms per condition except for Bap_LA, Bap_EFr, Bap_SH ($N = 21$); Bap_FM, Bap_LJ ($N = 18$); Bap_CI, Bap_EFm, Bap_ECl, Bap_PA, Bap_CM2 ($N = 13$). **d** Representative images of α-Syn aggregates in *C. elegans* NL5901 fed recombinant BAP proteins. Scale bars, 10 μm. **e** Quantification of α-Syn muscle inclusions per worm. rBap-PFF ($N = 15$), rEspN-PFF ($N = 18$), rBapB-MON and rEspN-MON ($N = 12$) and the nonamyloid domain rBap_CM2 ($N = 18$). Non-treated (Ø, $N = 15$). **f** Immunodetection of α-Syn:YFP in the protein fraction of *C. elegans* NL5901 fed *E. coli* expressing Bap_B and Esp_N amyloids (upper panel). Arrow and * indicate α-Syn monomeric and sub-monomeric forms, respectively. Anti-GFP antibody was used to detect α-Syn. Stained SDS-PAGE was used as loading control (lower panel). For panels (Fig. 4b, c, and e), data are shown as means and error bars are shown as the SE of means. For panels (Fig. 4b, c, and e), the data were analyzed using one-way ANOVA ($***p < 0.0001$) with Bonferroni's multiple comparison test. For all panels, $*p < 0.05$, $**p < 0.01$, $***p < 0.001$. MW; molecular weight. Source data are provided as a Source Data file.

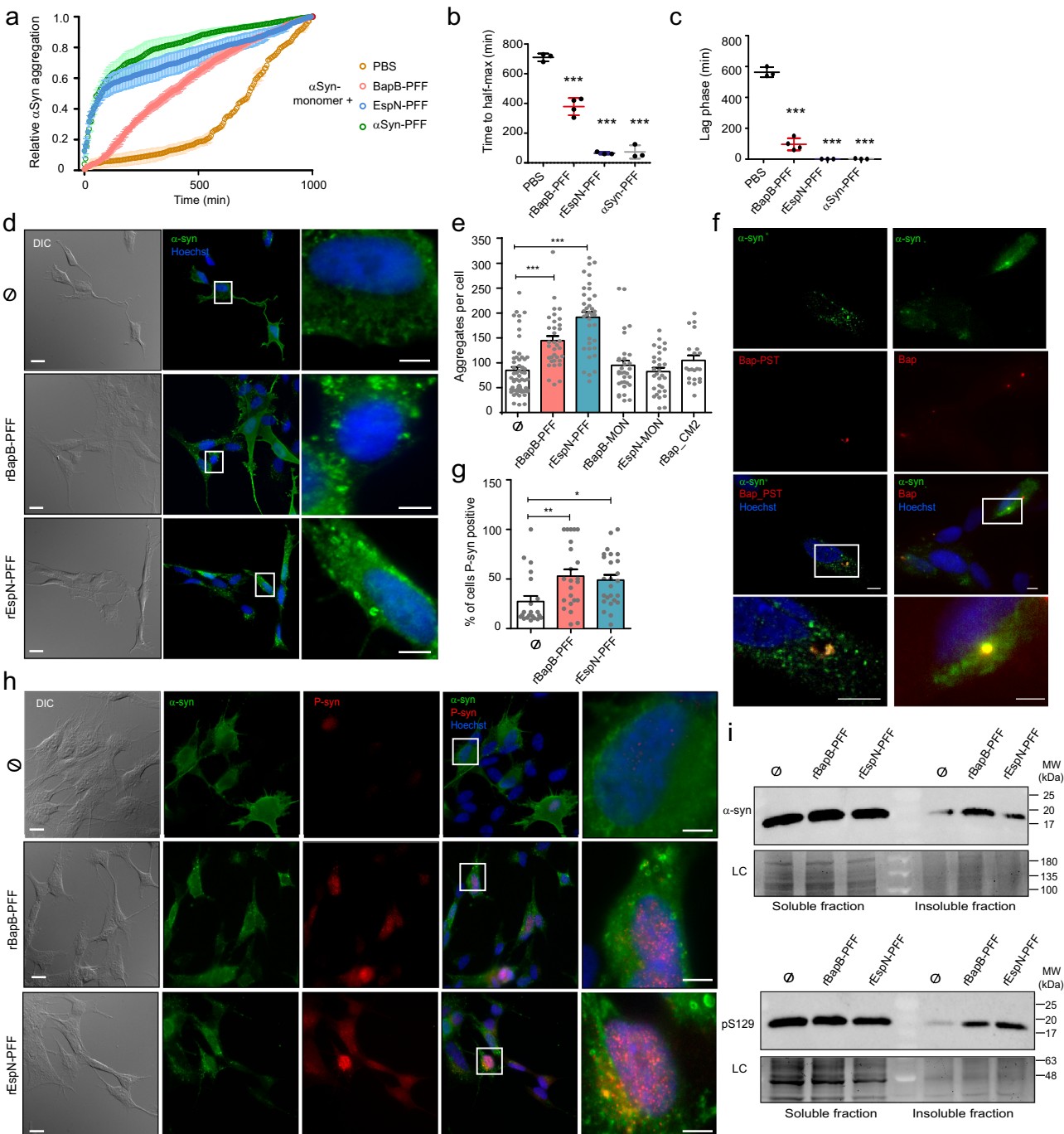

**Fig. 5 | BAP-derived amyloids exacerbate α-Syn aggregation and toxic synu-cleinopathy in SH-SY5Y cells. a** α-Syn aggregation in the presence of PBS ($N = 3$), rBapB-PFF ($N = 4$), rEspN-PFF ($N = 4$) and α-Syn-PFF ($N = 3$) (10:1 molar ratio) as measured by Th-T fluorescence over time. Time to reach exponential lag phase (**b**) and half maximum (**c**). Data were analyzed using one-way ANOVA with Dunnett's multiple comparison test. ***$p < 0.0001$. **d** Immunostaining of α-Syn in SH-SY5Y treated with BAP amyloids for 10 days. Higher magnifications of the highlighted regions are shown. Scale bars, 5 µm. **e** Quantification of α-Syn inclusions per cell. SH-SY5Y were treated with rBapB-PFF ($N = 34$) and rEspN-PFF ($N = 37$). As negative controls monomeric rBapB-MON ($N = 32$), rEspN-MON ($N = 30$) and the nonamyloid domain rBap_CM2 ($N = 20$) were used. Ø, non-treated ($N = 59$). **f** Colocalization of BapB and α-Syn using Bap-PST (left panels) or anti-Bap antibodies (right panels). Higher magnifications of the highlighted regions are shown in the lower panels.

Scale bars, 5 µm. **g** Percentage of SH-SY5Y cells positive for pS129-α-Syn. Graph represents the percentage of pS129-α-Syn positive cells per microscopic field analyzed. rBapB-PFF (fields analyzed $N = 23$, cells $N = 329$); rEspN-PFF (fields analyzed $N = 25$, cells $N = 310$); Ø, non-treated (fields analyzed $N = 21$, cells $N = 328$). **h** Representative images of pS129-α-Syn (red) and total α-Syn (green) immunostaining. Higher magnifications of highlighted regions are shown in lower panels. Scale bars, 5 µm. **i** Detection of α-Syn in SH-SY5Y cell extracts treated with BAP amyloids. Stain-free gel portions are shown as a loading control (LC). MW; molecular weight. Data were analyzed using one-way ANOVA (***$p < 0.0001$) with Bonferroni's multiple comparison test (**e**–**g**). For all panels, data are shown as means, and error bars are shown as the SE of means. *$p < 0.05$, **$p < 0.01$, ***$p < 0.001$. Source data are provided as a Source Data file.

after sonication of rBapB-PFF or rEspN-PFF with monomeric α-syn in vitro at concentrations lower than those at which they self-aggregated (Fig. S7a). The addition of preformed fibrils (α-Syn-PFF) was used as a positive control. The results showed that Bap_B and Esp_N amyloids accelerated α-Syn fibrilization in a similar way to that of α-Syn-PFF (Fig. 5a–c). The effect of rBap-PFF on α-Syn aggregation was dose-dependent and small seeds produced by sonication of the preformed fibers were needed (Fig. S7b). On the other hand, in vivo cross-seeding was tested using the human neuroblastoma cell line SH-SY5Y, which overexpresses human α-Syn and differentiates into neurons[42]. First, we showed that BAP amyloids did not affect the cell viability (Fig S8a). Next, we investigated whether preformed amyloid fibers could be taken up by neurons. To do this, differentiated SH-SY5Y cells were incubated with GFP tagged rBapB-PFF for 24 h[43]. Fluorescence microscopy analysis over time revealed that internalized rBapB-PFF colocalized with LysoTracker™ Red at 6 h post-treatment (Fig. S9a). This observation suggested that BAP amyloids are internalized and likely processed via the endosomal pathway toward perinuclear lysosomes. We then investigated whether intracellular BAP-like amyloids induced α-Syn aggregation. After the addition of rBapB-PFF and rEspN-PFF and incubation of the cells for 10 days, immunofluorescence microscopy confirmed the formation of α-Syn aggregates, including large perinuclear inclusions, formed in differentiated cells. In contrast, a few dispersed small aggregates were observed in the untreated cells and in cells that were incubated with monomeric BAP proteins, and with the nonamyloid domain rBap_CM2 (Fig. 5d and Fig.S9b). Quantification of the α-Syn aggregates showed that the number of aggregates per cell was significantly greater when the cells were treated with BAP fibers (Fig. 5e). We next tested whether BAP amyloids colocalize with α-Syn. BAP amyloids labeled with the fluorescent dye ProteoStat (rBapB-PST) were used to treat SH-SY5Y cells[15]. By the immunostaining of α-Syn, we observed colocalization of rBapB-PST with α-Syn (Fig. 5f). To avoid possible complications with the rBapB-PST signal, we directly visualize the colocalization of BAP and α-Syn through immunofluorescence double staining. Once again, BAP amyloids colocalized with α-Syn, and the rBapB-PFF and α-Syn signals appeared to overlap completely (Fig. 5f).

To further characterize the endogenous α-Syn aggregates, we probed SH-SY5Y cells treated with rBapB-PFF and rEspN-PFF with an antibody specific for α-Syn phosphorylated at serine 129 (P-S129). This post-translational modification of α-Syn has been found in the Lewy bodies of PD brains. Immunofluorescence staining confirmed that the number of P-S129 α-Syn-positive cells was significantly greater after the addition of rBapB-PFF and rEspN-PFF amyloids than in untreated control cells (Fig. 5g, h). Interestingly, P-S129 α-Syn strongly accumulated in the nuclei of neurons treated with rBapB-PFF or rEspN-PFF (Fig. 5h). Finally, immunoblot analyses of Triton X-100-soluble and Triton X-100-insoluble cell lysate fractions confirmed the presence of higher levels of α-Syn and P-S129 α-Syn in the insoluble fraction of cells incubated with BAP-derived amyloids (Fig. 5i and Fig. S10). We concluded that the treatment with BAP amyloid-like aggregates resulted in an increase in the number of phosphorylated aggregates in SH-SY5Y α-Syn cells, suggesting that the endogenous α-Syn aggregates in this cell culture model exhibit Lewy body-like characteristics.

### Intracerebral inoculation of BAP-like amyloids enhances α-Syn pathophysiology

Since BAPs efficiently promote the aggregation and fibrillation of soluble endogenous α-Syn in both the *C. elegans* muscle model of PD and in neuronal cells, we wondered whether BAP-amyloids would induce α-Syn pathology and PD features in a mouse model. Previous studies demonstrated that intrastriatal injections of fibrillar α-Syn into mice induce Parkinson's disease (PD)-like Lewy body (LB) pathology[44,45]. Based on that, we determined the short-term stability of the rBapB-PFF (10 μg/mouse) injected into striatum of C57BL6/

C3H F1 mice. Immunostaining analysis of Bap aggregates at 24 h and 72 h post-injection showed that the rBapB-PFF aggregates were stable, and were localized at regions other than the injection site (Fig. S11a–c). The long-term analysis of the motor function of mice showed that the intrastriatal injection of rBapB-PFF resulted in impaired fine motor performance (Fig. 6a). Animals injected with a single dose of amyloid rBapB-PFF showed a progressive decline in performance in the negative geotaxis test, starting at 90 days post-injection and becoming significant at 180 ($p \leq 0.05$) and 210 ($p \leq 0.05$) days, and an early rotarod deficit at 60 days ($p \leq 0.05$) that was maintained until day 210 ($p \leq 0.05$) (Fig. 6a). In addition, the rBapB-PFF-injected mice performed poorly in the wire suspension test (Fig. 6a). These behavioral changes were associated with a unilateral loss of dopaminergic innervation in the posterior striatum (18%; *$p \leq 0.05$) (Fig. 6b, e) and a 26% loss of dopaminergic neurons in the SNpc (*$p \leq 0.05$) (Fig. 6c, f). Moreover, there was a significant 10% decrease in the neuronal density (observed by NeuN staining) in the ipsilateral striatum (*$p \leq 0.05$) (Fig. 6d). Our results showed that the increase in phosphorylated α-Syn staining in the ipsilateral amygdala was not statistically significant (Fig. S12a). However, significantly greater levels of phosphorylated α-Syn were observed in the ipsilateral and contralateral SNpc in the rBapB-PFF-treated mice than in the control mice (*$p \leq 0.05$) (Fig. 6g, h). The presence of microglial activation (Iba-1 staining) or astrocyte activation (GFPA staining) was assessed in the ipsilateral SNpc. Iba-1 and GFAP-positive cells were unaffected in the rBapB-PFF-injected mice, suggesting that neurodegeneration was not associated with a neuroinflammatory response to Bap fibrils (Fig. S11d, e and S12b). The analysis of cytokine releases in the cell culture model also suggested that BAP amyloids did not trigger an inflammatory activation (Fig. S8b). Taken together, these results indicated that Bap amyloids can exacerbate α-Syn pathology, which contributes to fine motor deficits.

To investigate the mechanisms associated with α-Syn pathology we assessed α-syn mRNA expression level in the midbrains of mice treated with BAP amyloids and we found that it was unaffected (Fig. 6i). However, α-Syn protein expression level was significantly increased (78%; $p \leq 0.05$) (Fig. 6j, k). In human-derived dopaminergic cells α-Syn is predominantly degraded in lysosomes by chaperone-mediated autophagy (CMA). We measured the mRNA expression levels of 2 key proteins in the CMA pathway, LAMP-2A and Hsc70, in the midbrains of mice treated with BAP amyloids. There was a significant decrease (55%; $p \leq 0.05$) in *lamp*-2a mRNA expression with a concomitant decrease in LAMP-2A expression protein levels (19%; $P \leq 0.05$) (Fig. 6i–k). In order to confirm that CMA was compromised by BAP amyloids, we assessed α-Syn turn-over in cell models. Under control conditions α-Syn half-life was 36.6 h (Fig. 6l, m); however, rBapB-PFF exposure increased significantly α-Syn half-life to 57.8 h (57%; $p \leq 0.05$) (Fig. 6l, m) with a subsequent increase in α-Syn expression levels (103%; $p \leq 0.05$) (Fig. 6n, o). The decrease in α-Syn turnover was associated with a decrease in LAMP-2A expression levels (21%) (Fig. 6n, o). The results showed that CMA activity is compromised by amyloid rBapB-PFF in cell and animal models.

### Specific BAP genes are enriched in the microbiomes of PD patients

Considering that BAPs can induce PD-related pathologies and that BAP genes are encoded in the accessory genomes of the microbiome, suggesting that only certain strains have the potential to produce BAP amyloids, we investigated whether the presence of BAP-encoding genes could be associated with patients suffering PD. For this purpose, we took advantage of a recent large-scale metagenomics study involving a cohort of 490 PD patients and 234 neurologically healthy controls[46]. We performed a comprehensive analysis of the metagenomic data to investigate the presence of BAP-like gene regions and

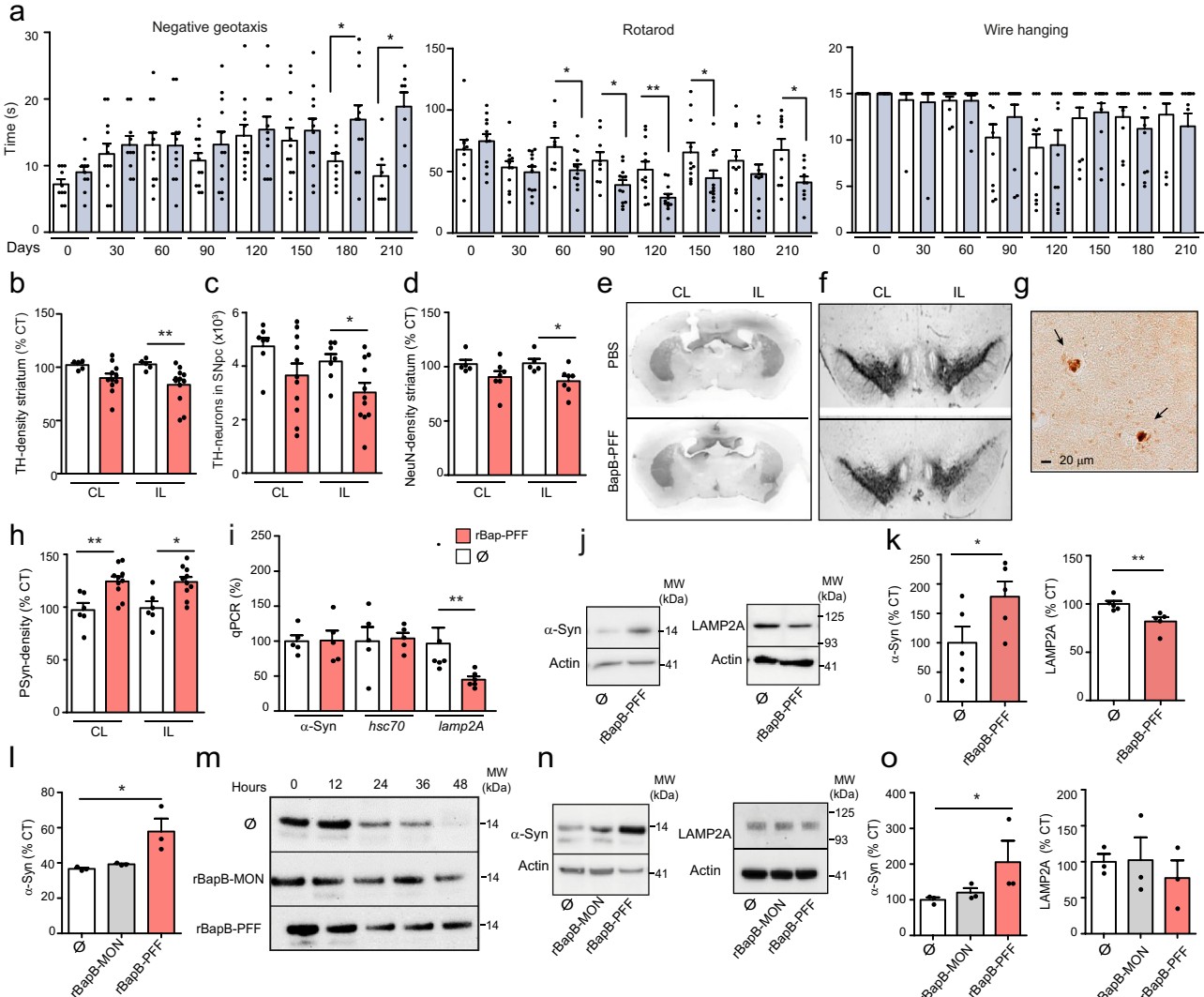

**Fig. 6 | Intracerebral inoculation of BAP-like amyloids enhances α-Syn pathophysiology in C57BL6/C3H F1 mice. a** Motor function assessed by quantifying time to turn around, wirehang and riding time. Blue bars, mice injected with rBapB-PFF. *N* = 12. Two tailed unpaired *t* test, *p* = 0.031, **p* = 0.0023, *p* = 0.038, *p* = 0.0177, **p* = 0.005, *p* = 0.049, *p* = 0.0178. Bars represent the means and standard errors of means. Motor data are compiled from two independent cohorts. **b** Anterior striatal TH innervation was quantified by optical density. Ø, *N* = 5; rBap-PFF, N = 11. **p* = 0.072. **c** Stereological quantification of TH-immunoreactive neurons in the SNpc. Ø, *N* = 7; rBap-PFF, N = 11. *p* = 0.0462. **d** Striatal NeuN was quantified by optical density. Ø, *N* = 5; rBap-PFF, *N* = 7. *p* = 0.0177. **e** Images for TH staining in posterior striatum. **f** TH staining of dopaminergic neurons in coronal sections of the SNpc. **g** Representative images of intraneuronal phospho-α-Syn inclusions in the ipsilateral (IL) SNpc of rBapB-PFF injected mice. **h** Quantification of Ser129 phospho-α-Syn densitometry in the contralateral (CL) and IL of midbrain. Ø,

*N* = 6; rBapB-PFF, *N* = 10. **p* = 0.047, *p* = 0.011. Data were analyzed using two-tailed Mann–Whitney test (**b**–**d**, **h**). **i** mRNA level of α-syn, *hsc70* and *lamp2A* genes. *N* = 5, **p* = 0.0041. **j** Western-blots analyzing α-Syn and LAMP-2A expression. **k** Quantification of total α-Syn and LAMP2A levels normalized to beta-actin. *N* = 5, *p* = 0.0476, **p* = 0.004. Data were analyzed using one-tailed Mann–Whitney test (i, k). **l** Average of the α-Syn half-life. Disappearance of α-Syn was quantified as the mean of three experiments. *p* = 0.0273. **m** Representative images of α-Syn turn-over during 48 h. **n** Western-blots analyzing α-Syn and LAMP-2A expression in SH-SY5Y. **o** Quantification of total α-Syn and LAMP2A levels normalized to beta-actin. *N* = 3, *p* = 0.05. Data were analyzed using Kruskal–Wallis with Dunn's multiple comparison test (**l**, **o**). For all panels, data are shown as means, and error bars are the SE of means. *p* < 0.05, **p* < 0.01, ***p* < 0.001. Mice intrastriatally injected with rBapB-PFF (red bars), monomeric rBapB-MON (grey bars) or PBS (white bars). MW; molecular weight. Source data are provided as a Source Data file.

whether their presence is associated with PD. The raw metagenomic shotgun sequencing data (BioProject ID PRJNA834801) were reanalyzed to determine the presence of the BAP-encoding genes listed in Table S2.

The results of the binary logistic regression revealed that PD patients had a significantly greater odds ratio of having BAP genes such as *bap_FM* (OR: 2.2; *p* < 0.05), *esp* (OR: 3.2; *p* < 0.01), *bap_EFm* (OR: 4.6; *p* < 0.01), *bap_ECo* (OR: 1.7; *p* < 0.05), *bap_EFg* (OR: 1.9; *p* < 0.01) and *bap_LJ* (FC: 2.6; *p* < 0.001) (Fig. 7a). We also found enrichment of the *csgA* gene (OR: 1.5; *p* < 0.05), which has been shown to be substantially enriched in the PD metagenome[46]. The

analysis was adjusted for variables such as sex, age, collection method and total sequences, and retained significance for the association of BAP genes such as *bap_EFm*, *esp*, *bap_EFm*, *bap_ECo*, *bap_EFg* and *bap_LJ* with PD (Tables S7 and S8). Although BAP-genes were also found in neurologically healthy controls, the abundance (RPKM) of these genes per sample was significantly lower than that in PD patients (Fig. 7b and Table S9), suggesting that the concentration of BAP-encoding genes in the microbiota is also a variable to consider. Alternatively, it cannot be excluded that neurologically healthy controls carrying BAP-encoding strains could develop PD in the future. In conclusion, these results confirm that several BAP genes

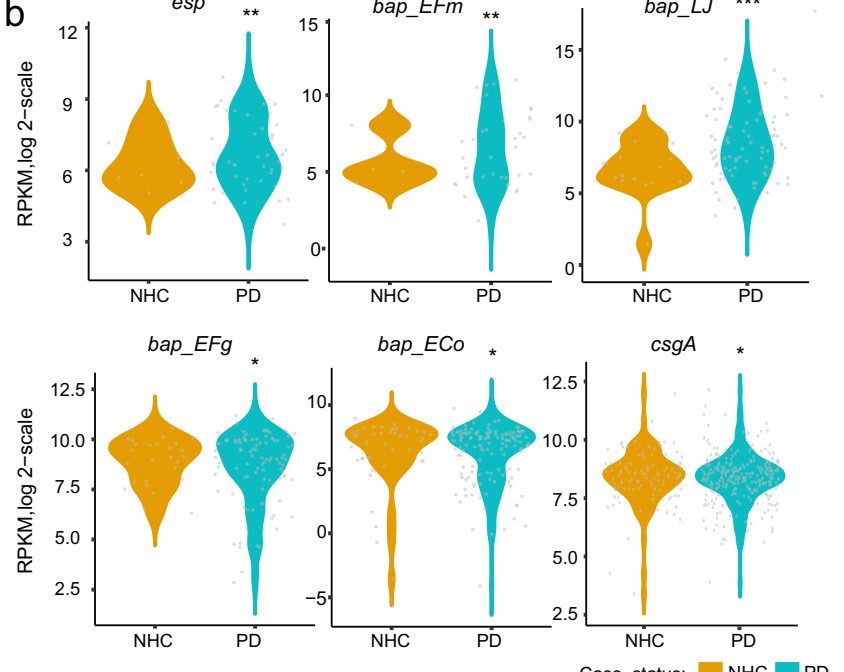

| Gene | PD summary stats | % | NHC summary stats | % | P | OR (95%) |
|---|---|---|---|---|---|---|
| *bap_FM* | 40 | 8% | 9 | 4% | 3.90E-02 | 2.2 [1-5.3 ] |
| *bap_LL* | 110 | 22% | 48 | 21% | 6.30E-01 | 1.1 [0.8-1.7 ] |
| *bap_CC* | 7 | 1% | 2 | 1% | 7.30E-01 | 1.7 [0.3-16.7 ] |
| *esp* | 44 | 9% | 7 | 3% | 2.80E-03 | 3.2 [1.4-8.5 ] |
| *bap_ECo* | 136 | 28% | 44 | 19% | 1.00E-02 | 1.7 [1.1-2.5 ] |
| *bap_EFm* | 36 | 7% | 4 | 2% | 1.40E-03 | 4.6 [1.6-17.8 ] |
| *bap_CM* | 0 | 0% | 0 | 0% | 1.00E+00 | 0.5 [0-24.2 ] |
| *bap_EFr* | 4 | 1% | 1 | 0% | 1.00E+00 | 0.5 [0.2-94.8 ] |
| *bap_SH* | 0 | 0% | 0 | 0% | 1.00E+00 | 0.5 [0-24.2 ] |
| *bap_TG* | 0 | 0% | 0 | 0% | 1.00E+00 | 0.5 [0-24.2 ] |
| *bap_ECl* | 71 | 14% | 28 | 12% | 4.20E-01 | 1.2 [0.8-2.1 ] |
| *bap_LA* | 17 | 3% | 4 | 2% | 2.40E-01 | 2.1 [0.7-8.5 ] |
| *bap_DF* | 127 | 26% | 71 | 30% | 2.10E-01 | 0.8 [0.6-1.2 ] |
| *bap_PA* | 0 | 0% | 0 | 0% | 1.00E+00 | 0.5 [0-24.2 ] |
| *bap_AB* | 1 | 0% | 1 | 0% | 5.40E-01 | 0.5 [0-37.6 ] |
| *bap_EFg* | 93 | 19% | 26 | 11% | 7.30E-02 | 1.9 [1.2-3.1 ] |
| *bap_LJ* | 83 | 17% | 17 | 7% | 3.20E-04 | 2.6 [1.5-4.8 ] |
| *bap* | 0 | 0% | 0 | 0% | 1.00E+00 | 0.5 [0-24.2 ] |
| *bap_CI* | 0 | 0% | 0 | 0% | 1.00E+00 | 0.5 [0-24.2 ] |
| *bap_SS* | 289 | 59% | 129 | 55% | 3.40E-01 | 1.2 [0.8-1.6 ] |
| *csgA* | 279 | 57% | 109 | 47% | 1.10E-02 | 1.5 [1.1-2.1 ] |

**Fig. 7 | Specific BAP-genes are enriched in the microbiomes of PD patients.**
**a** Percentage of the presence of *bap* genes in bacterial species from gut microbiota in Parkinson's disease patients (PD) ($N = 490$) and in neurologically healthy controls (NHC) ($N = 234$). Data were reanalyzed by binary logistic regression using the glm() function with binominal distribution in R. OR: Odds ratio: 95% CI: 95% Confidence Interval; P: P-value. Odds ratio with significant binary logistic regression are highlighted. **b** Violin plots reporting the relative abundances calculated as the log2 of RPKM among NHC and PD patients from metagenomic data analysis. BAP positive samples were considered for the analysis. To identify statistically significant differences in RPKM between PD and NHC groups, linear regression was performed using MaAsLin2 test. For multiple testing corrections, the Benjamini-Hochberg FDR method was used. Relative abundance of *csgA* was used as control; *esp*, **$p = 0.003$; *bapEFm*, **$p = 0.002$; *bapLJ*, ***$p = 0.0001$; *bapEFm*, *$p = 0.016$; *bapECo*, *$p = 0.014$; *csgA*, *$p = 0.015$. Source data are provided as a Source Data file.

could be associated with PD and open up the possibility of using these genes as biomarkers of PD disease.

## Discussion

With the emergence of the field of neuromicrobiology, the amyloids expressed and secreted into enteric biofilms have been proposed to act as active molecules associated with neurodegeneration[47–49]. However, analysis of the entire amylome in the GI tract has been overlooked, and studies aimed at determining the role of enteric amyloids in disease are still lacking. Here, we identified the presence of BAP amyloid-like structures in the enteric microbiota, which impact α-Syn aggregation and pathology. Moreover, we provide evidence that an enrichment of BAP-associated genes appears to be a feature of the PD metagenome.

Polymicrobial biofilms have been shown to interact with GI surfaces[50–52]. Biofilms are mainly found in the ileum and cecum, and, to

a lesser extent, in the distal colon but are also attached to fecal particles[53,54]. Most related studies have focused on determining the composition of the bacteria that form part of enteric biofilms; however, our knowledge of the components of the biofilm matrix is incomplete. The availability of complete bacterial genomes has made it possible to show that biofilm-associated proteins are widely distributed among bacteria from the gut (Fig. 1a). BAPs from Gram-positive bacteria have a modular structure with predicted aggregation-prone domains predominantly at the N-terminal region. Proteolytic cleavage releases the N-terminal domain containing amyloidogenic peptides, which subsequently self-assemble into amyloid-like fibrillar structures. Conversely, in Gram-negative bacteria, BAPs have evolved to display a tandem repeat core bearing predicted aggregation-prone domains. The exact mechanism by which the amyloid domains of these extremely large proteins assemble into fibrillar structures is unknown. The presence of an amyloid-prone region localized in tandem repeats can favor multiple intermolecular and intramolecular interactions between repeat units, increasing the amyloid propensity of the protein and the stability of the resulting fibrils[55].

The pH of the gastrointestinal tract is highly variable, ranging from 1.7 to 4.7 in the gastric tract and 5.9 to 7.8 in the small bowel and values fluctuate between pH 5 and pH 8 in the colon[56]. pH values determine bacterial colonization of the host and biofilm formation[15,16,57,58]. Most of the amyloid domains of BAPs have acidic isoelectric points (Table S6), implying that the lack of net charge at acidic pH would facilitate the conversion of the protein from its soluble state to the amyloid-like state. Notably, Bap_LL, Bap_FM, and Bap_CI also exhibited amyloidogenic behavior at neutral pH. This finding could be an important selective advantage for the bacteria allowing them to better adapt to a wide range of microenvironments.

We demonstrated the presence of BAP-related amyloidogenic structures in human fecal samples. To the best of our knowledge, this is one of the main demonstrations of the presence of bacterial amyloids in the human gut microbiota. Only the presence of curli structures in the GI tract has been demonstrated using in vivo infection models[11]. However, the production of curli amyloids in vivo has been questioned because their expression is induced in the laboratory at temperatures below those found in the body. Using a filter-trap retention assay, we detected BAP amyloid-like structures in the SDS-insoluble protein fraction isolated from the gut microbiota that were sensitive to formic acid and TFA:HFIP treatment. The use of an antibody against the amyloid region of Esp, together with the amyloid conformational antibodies A11 and OC, revealed the presence of Esp-amyloidogenic structures. The analysis was performed on stool samples available from an ongoing cohort study of healthy individuals and patients suffering from irritable bowel syndrome (IBS). Individuals with IBS have a 48% greater risk of developing PD than healthy patients[59,60]. Constipation is a major symptom of IBS, and altered intestinal permeability and microbiome disturbance contribute to the pathogenesis of this disease. Similarly, gastrointestinal dysfunction, particularly constipation, is a nonmotor symptom in PD and in a substantial percentage of patients, it precedes the onset of motor symptoms by several years[61]. Therefore, IBS may define a prodromal condition for PD and the analysis of the microbiome in IBS patients may help to identify etiologic factors that may trigger the onset or progression of PD. In this study, we have identified BAP-related amyloids in fecal samples from IBS patients. More remarkably is that 8 of the 10 samples positive for A11 belong to patients with IBS. Therefore, it would be of great interest to follow up those patients and determine a possible association between the presence of amyloids in IBS patients and the development of PD.

Increasingly, studies show that individuals with neurodegenerative diseases have altered microbiomes, and the relative abundance of certain microbial species correlates with the severity of functional symptoms[62-68]. In the case of PD, 30% of the species, genes and pathways exhibited altered abundances[46]. Consistent findings across studies include the increase in *Lactobacillus* and *Enterococcus* spp. and members of the *Enterobacteriaceae* family in PD patients[46,69,70]. *F. magna*, which forms a slow-growing biofilm in the oral cavity and intestine, is also enriched in the feces of PD patients[71]. Our work showed that the BAP genes are part of the accessory genome of PD-associated bacterial species. Interestingly, we detected an enrichment of the presence and abundance of the BAP genes in the PD metagenomes. BAP genes appear to be located at the accessory region of the microbiome, suggesting that only certain strains could produce BAP amyloids[72]. It is therefore tempting to speculate that amyloids expressed by some PD-associated species may act as key factors triggering and/or exacerbating this disease. The fact that BAP amyloids were detected in the microbiome of the control group suggests that pathogenic amyloids may need to reach a threshold concentration in the intestine to induce pathology. This hypothesis was supported by the fact that cross-seeding of α-Syn is concentration-dependent (Fig. S7). The threshold concentration of BAP amyloids could also have collateral effects on the intestine, such as triggering a perturbation of the gut barrier, modulating the physiology of the consortium of gut bacteria by inducing changes in the bacterial composition or promoting interspecies amyloid aggregation, all of which may encourage an environment conducive to disease development.

Although recent studies have shown the existence of a wide variety of amyloid fiber architectures[22], bacterial functional amyloids can be considered as canonical molecular mimics of human molecules that can play a pathological role in the central nervous system (CNS) in certain situations, either by nucleating the polymerization of human amyloid proteins (cross-seeding) or priming an inflammatory response[47,49,73]. This is the case for curli and human amyloid Aβ, both of which stimulate Nos2 production (a hallmark of inflammation) in macrophages and microglia through a TLR2-dependent mechanism[74,75]. However, this process does not appear to be the central mechanism by which BAP amyloids induce α-Syn aggregation. The lack of a neuroinflammatory response in the substantia nigra pars compacta (SNpc), 72 h after the interstitial injection of rBapB-MON or rBapB-PFF excludes an inflammatory mechanism similar to the one elicited by LPS injection[76]. Moreover, rBapB-PFF injection did not result in long-term glia activation or increase in inflammatory markers 7 months after injection. However, involvement of neuroinflammation in PD pathogenesis is supported by in vivo studies and post-mort PD brain analysis[77] and we cannot exclude an earlier neuroinflammatory response in SNpc that preceded neurodegeneration similar to the activation described in α-Syn PFF models[77,78]. Alternatively, bacterial amyloids could act as aggregation nuclei that induce fibrillation of certain non-homologous amyloids in a process called cross-seeding. Recent studies have shown that purified curli amyloids catalyze α-Syn aggregation in vivo[64], and, in vitro studies have shown that bacterial amyloids seed the fibrilation of Aβ, thus suggesting that cross-seeding aggregation might contribute to the onset of AD[79]. Our study adds to the repertoire of proteins that could effectively play a role in the polymerization of human amyloids. We showed that BAP amyloids colocalized with α-Syn inside human neurons and increase human α-Syn aggregation using in vitro and in vivo models. However, the exact mechanisms by which bacterial amyloids colocalize with human α-Syn to engage in cross-seeding, and particularly, how this process affects subsequent amyloid aggregation in the brain, are still unknown.

The gut-brain axis provides a bidirectional way to communicate between the GI tract and the CNS through several mechanisms, including through neuroendocrine cells, activation of immune cells and the enteric nervous system (ENS), with the vagus nerve acting as the main neural communication pathway[80]. During homeostasis, bacteria stimulate the afferent neurons of the ENS, and the resulting vagal

signals induce anti-inflammatory responses. However, in certain situations such as those involving age[81,82] and/or altered gut microbiota caused by pathogens or diseases[83,84], gut barrier function may be disrupted, presumably allowing the translocation of microbiota-derived products that can interact with enteric neurons and, more excitingly, reach the brain itself. Current data suggest that the PD process starts in the ENS and spreads to the lower brainstem via the vagus nerve[85]. It is therefore conceivable that bacterial-derived products translocated from the gut, including BAP amyloids or bacteria expressing BAPs, may exert their effects on the CNS via the ENS and/or have a direct effect by spreading to the brain in a "prion-like" manner. A recent study revealed that commensal bacteria from the gut can be translocated through the vagus nerve to the brain in mice with intestinal dysbiosis[86]. Assuming that amyloids may migrate from the gut to the brain through anatomical paths, such as spinal or vagal neurons[87,88], in this study, we forced this scenario using direct intra-cerebral inoculation of BAP preformed amyloids (rBapB_PFF) in wild-type nontransgenic mice. We chose to target the dorsal striatum, an area connected to several CNS nuclei, including midbrain dopaminergic neurons. Previous studies have shown that the injection of α-Syn-PFF into this area induces the formation of α-Syn aggregates, the loss of dopaminergic neurons and motor abnormalities[44]. Our data showed that rBapB-PFF spread in the brain 72 h after the interstitial injection and demonstrated that rBapB-PFF presence in SNpc was virtually undetectable (Fig S11). These results support the hypothesis that rBapB-PFF seeds recruit soluble α-Syn monomers initiating a prion-like spread of pathologic α-Syn previously described in PD brains and α-Syn PFF animal models[89,90]. Although this effect has been observed using preformed BAP fibers, it cannot be ruled out that monomeric proteins may aggregate in vivo over time causing synucleopathologies in the future.

There is increasing evidence that α-Syn degradation pathways are compromised in PD, in particular LAMP-2A decrease in PD midbrain has been reported in several studies[91,92]. Moreover, in vitro studies demonstrated that impaired chaperone-mediated autophagy (CMA) function resulted in α-Syn decreased degradation and intracellular accumulation[91,93]. We demonstrated that BAP amyloids exposure leads to α-Syn pathology and CMA impairment. Although it cannot be excluded a direct effect of rBapB-PFF upon LAMP-2A, current data pointed to a central role of pathologic α-Syn in CMA dysfunction as a decrease in LAMP-2A has been reported in midbrain in the α-Syn-PFF model[94]. Our results pointed to a common mechanism in BAP amyloids model, α-Syn-PFF model and PD.

In conclusion, our study fills important gaps in our present knowledge of the amylome composition and its role in disease and advances the understanding of the connection of biofilm-associated amyloids from the gut microbiota with human neurodegenerative diseases. These findings may have important implications for earlier diagnosis and more effective therapies targeting the early stages of pathology.

## Methods

All mouse procedures were performed in accordance with institutional protocol guidelines at the Ethics Committee of CSIC and the Ethics Committee on Animal Experimentation of the Center of Biomedical Research of La Rioja (CIBIR). Mice were maintained accordingly to the National guidelines (RD 53/2013), under protocols approved the Comité de Ética y Bioseguridad del IdAB (Ref. number 1357/2022) and the Ethics Committee on Animal Experimentation of CIBIR (permit number LAE-04) and were conducted according to the National Institute of Health (NIH) Guide for the Care and Use of Laboratory. The study involving patients was conducted in accordance with the Declaration of Helsinki, and the protocol was approved by the Ethical and Scientific Committees 2020.153 and 222/2022, University of Navarra. All included patients gave their consent to participate in the study. Participants did not receive any compensation for participating in the study.

### Oligonucleotides, plasmids, bacterial strains, and culture conditions

Bacterial strains, plasmids, and primers used in this study are listed in Tables 1 and 4. *Escherichia coli* strains were grown in Luria-Bertani (LB) broth or Tryptic Soy Broth supplemented with glucose 0.25% (w/v) (TSBg) (Conda-Pronadisa). When required, the medium was supplemented with ampicillin (Amp) 100 µg/ml, erythromycin (Erm) 10 µg/ml, chloramphenicol (Cm) 20 µg/ml, L-arabinose 0.2 % (w/v) and iso-propyl β-D-1-thiogalactopyranoside (IPTG) 1mM.

### Library construction of BAP domains using the C-DAG system

To obtain the *E. coli* strain collection for C-DAG system, coding DNA of predicted amyloidogenic domains were synthetized by GeneArt (Invitrogen, ThermoFisher Scientific) and were cloned *NotI/XbaI* into the pExport plasmid (Table S5 and S10). The final plasmid collection was transformed in *E. coli* VS39 pVS76 strain and positive clones were selected on LB agar containing Amp (100 µg/ml), Cm (20 µg/ml). Induction of protein expression and the presence of amyloid-like material was assessed on LB agar containing Amp (100 µg/ml), Cm (20 µg/ml), 0.2 % (w/v) L-arabinose, 1 mM IPTG, and 0.8 % (w/v) Congo red dye (CR). Plates were incubated at 30 °C for two days for evaluating colony color phenotype. To quantify the amount of CR bound to fibers generated from C-DAG strains, liquid overnight cultures grown at 37 °C were diluted 1:30 into 150 µl of low-salt LB medium containing Amp100 µg/ml, Cm 20 µg/ml, 0.2% (w/v) L-arabinose, 1 mM IPTG and 25 µg/ml of CR dye, and placed into flat clear-bottomed black 96-well plates (Thermo Fisher Scientific). *E. coli* VS39 pVS76 was used as negative control of CR binding. Plates were placed into a BioTek Synergy H1 microplate reader at 37 °C and CR fluorescence (ex, 525 nm; em, 625 nm) and bacterial growth (Optic Density of 600 nm [$OD_{600nm}$]) were read every 10 min for 16 h with shaking between reads (800 rpm). At the endpoint, CR fluorescence:$OD_{600nm}$ ratio was calculated to normalize CR fluorescence values according to bacterial growth. At least three independent experiments were performed in triplicate. For transmission electron microscopy (TEM), *E. coli* cells exporting amyloidogenic domains were washed twice with 1x phosphate buffered saline (PBS) and then fixed with 2% (v/v) paraformaldehyde (Sigma) for 1h at room temperature. Formvar/carbon-coated nickel grids were deposited on a drop of fixed sample during 5 min and rinsed three times with PBS. Negative staining was performed using 2% (v/v) uranyl acetate (Agar Scientific, Stansted, UK). Observations were made with a JEOL 1011 transmission electron microscope (Fig. S13).

### Protein production, purification, and in vitro fibril formation

For recombinant expression, encoding DNA was PCR amplified from pExport plasmids (Table S10) using high fidelity Phusion DNA polymerase (Thermo Scientific) and primers listed in Table S6. PCR products were cloned in pET46-Ek/LIC vector by annealing or pET28a vector using *NcoI/XhoI* restriction enzymes. Overnight cultures of *E. coli* BL21 DE3 containing the expression plasmid were diluted 1:100 and grown to an $OD_{600nm}$ of 0.6. IPTG were added to a final concentration of 0.1 or 1 mM and cultures were shaken (150 rpm) at 18 °C overnight or 5 h at 37 °C. After centrifugation for bacterial harvesting (6200 × *g* for 10 min), pellets were resuspended in lysis buffer (1x phosphate buffer, 20 mM imidazole, 0.2 mg/ml lysozyme, 20 µg/ml DNase, 1 mM MgCl2, 1 mM PMSF), sonicated with 30-s incubation on ice between cycles (3 cycles of 30 seconds; 40 % amplitude; 5 cycles of 30 s; 50 % amplitude), and centrifuged (20,200 × *g* for 30 min at 4 °C). Supernatants were filtered (0.45 µm) and 10 µg/ml of DNase and RNase were added. Proteins were purified by Ni affinity chromatography using HisTrapTM FF columns (GE Healthcare).

The concentration of the purified BAP amyloidogenic domains was determined by the Bradford Protein Assay (BioRad). Bap_SH, Bap_PA1, and Bap_Eco domains could not be purified into a soluble state. For the in vitro fibril formation, purified amyloid domains were diluted to 2 µM in 2 ml of phosphate-citrate buffer at pH 4.5 and were incubated at 37 °C under shaking (200 rpm) in a glass tube.

## Generation of chimeric proteins and biofilm phenotype

BAP-domains were amplified using primers shown in Table S10. The signal peptide of Bap from *S. aureus* was amplified using primers 5′-GGATCCTTTATTTTGAGGTGAGTAAATATGGG and 5′-AGCAC-TATTTTGTACTTCCGC and fused to BAP-domains by overlap PCR. To allow anchoring of BAP-domains to the bacterial cell wall, the R region of clumping factor A (*clfA*) gene containing an LPXTG motif was amplified from *S. aureus* Newman strain using primers 5′-GGTACCGACTACAAAGACCATGACGGTGATTATAAAGATCATGA-CATCGACTACAAGGATGACGATGACAAGGAAATTGAACCAATTCCA-GAGGAT and 5′ GAATTCTTACACCCTATTTTTTCGCC and fused to BAP domains. The resulting fragments were cloned into pCN51 vector. Chimeric proteins containing the BAP-amyloidogenic domains were expressed under the control of the Pcad-cadC promoter; a cadmium inducible promoter in *S. aureus* Δbap[95]. For the biofilm formation assay strains were grown overnight at 37 °C and then diluted 1:40 in TSBg. Cell suspension was used to inoculate sterile 96-well polystyrene microtiter plates (Thermo Scientific). After 24 h of static incubation at 37 °C, wells were gently rinsed two times with water, dried, and stained with crystal violet for a few minutes. To quantify biofilm formation capacity, crystal violet adhered at the bottom of the wells was resuspended with 200 µl of a solution of ethanol:acetone (80:20 v/v) and the absorbance was quantified at 595 nm.

## Fecal sample collection and metagenome analysis

Samples and data from patients included in the study were provided by the Biobank of the University of Navarra and were processed following standard operating procedures approved by the Ethical and Scientific Committees 2020.153 and 222/2022 (Table S3). Samples were kept at −80 °C until further processing. 35 individuals (between 17 and 82 years old; age average of 45 years old) with irritable bowel syndrome (IBS) and 14 control were enrolled in the study. Fecal samples were homogenized in PBS 1X. Bacterial DNA from 200 mg of fecal samples was extracted using the Maxwell® RSC Fecal Microbiome DNA Kit. Extractions were performed according to manufacturer instructions with the introduction of a previous mechanic disruption step with bead-beating. The sequencing library was prepared by random fragmentation of the DNA followed by 5′ and 3′ adapter ligation. Each fragment was amplified into distinct, clonal clusters through amplification. Templates used for sequencing by Illumina SBS technology. Host DNA was filtered out from raw data. The host DNA was filtered out by mapping against the host reference genome using KneadData v0.74. Trimmomatic v0.38 was used to remove the Illumina sequencing adapter and trim low-quality end bases. To assess the quality of raw data and trimmed data, FASTQC v0.11.6 was performed. Centrifuge v1.0.4 was performed for taxonomy assignment of trimmed data with primary assignments set as 1. The count and portion of the reads assigned to each taxon number were calculated at each taxon ranks (Kingdom, Phylum, Class, Order, Family, Genus, Species, and Subspecies). Normalized abundance of taxon at level of species was calculated as: Abundance = Ra/Rt *G; where Ra: Read count assigned to the taxon; Rt: Total read count; G: Average genome size of the taxon. After calculating abundance, a relative abundance which is the ratio of the abundance, compared with total abundance sum was calculated. The abundance of BAP was calculated as reads per kilobase per million mapped reads (RPKM).

## Filter retention assay

Microbial cells were enriched from the fecal samples by differential centrifugation. Briefly, fecal samples were normalized to 32,4 mg/ml of wet weight in PBS and 10 ml of the suspension were centrifuged at 500 × g for 2 min at 4 °C to remove food debris. The supernatant was then filtered through a 20 µm pore membrane. After that, the filtrate was collected and centrifuged at 26,000 × g for 10 min at 4 °C. Pellets were resuspended in a 1/10 volume of the original culture of Tris buffer 50 mM pH 8, and trypsin (2 µg per sample). Digestion with trypsin was performed under shaking conditions for 2h at 37 °C. Insoluble protein fraction was separated by two subsequent centrifugations (18,800 × g, 10 min and 18,800 × g, 30 min). Taking advantage of the insolubility of amyloid fibers in detergents, pellets were washed with SDS 1% to improve their purity. Insoluble proteins were diluted 1:10 in Tris buffer 50 mM pH 8 and 50 µl were subjected to vacuum filtration through a 96-well dot blot apparatus (Bio-Rad) with 0.45 µM pore size cellulose acetate (Sterlitech). The resultant membranes were blocked in PBS-Tween 0,1% and 5% (w/v) non-fat dry milk overnight at 4 °C. To detect Bap and Esp amyloids, rabbit polyclonal antibodies against rBap_B[15] and chicken polyclonal antibodies against rEsp_N[16] were used. To detect Bap_LL and Bap_LA amyloids rabbit polyclonal antibodies raised against purified rBap_LL and rBap_LA proteins were supplied by Proteogenix company. Antibodies were diluted 1:5.000. Anti-amyloid fibrils (OC) antibody (StressMarq; SPC-507S) and anti-amyloid oligomers (A11) antibody (StressMarq; SPC-506) diluted 1:1.000 were used to detect fibers and oligomers respectively. Bound antibodies were detected with horseradish peroxidase-conjugated anti-rabbit (Invitrogen; ref: 31460, 1:5.000), anti-chicken (Abcam; ref: Ab97135, 1:20.000) and anti-mouse (Life technologies; ref: A16072, 1:5.000) antibodies. For the detection of Esp amyloids in stool samples, 6 weeks CD-1 female mice ($N = 3$) were subjected to a broad-spectrum antibiotic treatment (vancomycin 0.5 g/l, neomycin 1 g/l, ampicillin 0.5 g/l) to deplete the gut microbiota. After antibiotics treatment, rEspN_PFF were orally administered for 2 days. The SDS resistant fraction purified from fecal samples of the mice ($N = 3$) were subjected to a filter retention assay. Immunoblotting was performed using anti Esp_N antibodies.

## Preparation of protein aggregates

Sodium phosphate/citric acid buffer were prepared at pH 4.5 and 7.7 by mixing 0.1 M citric acid and 0.2 M $Na_2HPO_4$. PD-10 desalting columns were used for sample buffer exchange, and the recombinantly purified proteins were then diluted to a final concentration of 0.5 mg/ml in either pH buffers. For protein aggregation reactions, samples were incubated under agitation at 600 rpm for 7 days at 37 °C.

## Light scattering

Light scattering was measured exciting at 330 nm, and recording in the range of 320 – 340 nm using an excitation and emission bandwidth of 5 nm, using a Jasco FP-8200 fluorescence spectrophotometer (Jasco Corporation, Japan)

## Amyloid dyes binding

Thioflavin T (Th-T) and Congo Red (CR) dyes were used to determine the amyloid nature of protein aggregates. For Th-T binding assay, incubated proteins were diluted 1:10 in citric acid/ $Na_2HPO_4$ buffer containing 25 µM Th-T. Th-T emission fluorescence was detected on a Jasco FP-8200 fluorescence spectrophotometer (Jasco Corporation, Japan) using an excitation wavelength of 445 nm and recording in the range of 460–600 nm, with an excitation and emission bandwidth of 5 nm. For CR binding assay, incubated proteins were diluted 1:10 in citric acid/ $Na_2HPO_4$ buffer, containing 20 µM CR. Optical absorption spectra were recorded from 375 to 700 nm in a Specord200 Plus spectrophotometer (Analytik Jena, Germany). Spectra of incubated proteins without dye were acquired to subtract scattering.

## Fourier transform infrared spectroscopy (FTIR)

After 7 days of incubation, 100 µl of aggregated proteins were centrifuged at 12,000 g for 30 min, and pellets were resuspended in 10 µl of MQ water. FTIR spectra were acquired using a Bruker Tensor 27 FTIR (Bruker Optics, USA) supplied with a Specac Golden Gate MKII ATR accessory. Samples were placed on the ATR crystal and dried under $N_2$ flow. Each spectrum consisted of 32 acquisitions measured at a resolution of 1 cm$^{-1}$ within the amide I region (1700 to 1600 cm$^{-1}$). Data were acquired and normalized using the OPUS MIR Tensor 27 software (Bruker Optics, USA), with the Min/Max normalization method. Data was deconvoluted by automated peak fitting using the Peak Fit software. The resulting area, amplitude and central values of the fitted bands were plotted.

## Transmission Electron Microscopy (TEM)

Aggregated proteins were visualized with a JEM 1400 transmission electron microscope (JEOL ltd, Japan) operating at 80 kV. For sample preparation, 10 µl of incubated proteins were deposited onto a carbon-coated copper grid for 10 min, and the excess liquid was wiped with filter paper. Then, 2% (w/v) uranyl acetate was added for negative staining. Grids were exhaustively scanned using a CCD GATAN ES1000W Erlangshen camera (Gatan Inc., United States) (Fig. S13). For immunogold labeling, grids coated with the sample were washed and incubated for 45 minutes on a drop of PBS containing 1:10 anti-polyhistidine antibody (Sigma; ref: H1029). After washing with PBS, grids were incubated 45 min with gold-conjugated (10nm) goat-anti-rabbit secondary antibody.

## Cross-seeding of α-Syn

To assess the cross-seeding between BAP amyloids and human amyloids, the active human recombinant α-Syn monomer (SPR-321, StressMarq) was diluted to 10 µM into 100 µl of PBS containing 2 µM Th-T supplemented with 1 µM BAP preformed fibrils (PFF) or 10 nM active human recombinant α-Syn-PFF (SPR-322, StressMarq). All PFF were sonicated in an ultrasonic bath for 1 h immediately prior to use. Samples were placed into flat-bottomed black 96-well plates (Costar) with a 1/8″ diameter Teflon ball (Polysciences, Inc.) for homogenous mixing. Plates were read in a BioTek Synergy H1 plate reader at 37 °C and Th-T fluorescence (ex, 440 nm; em, 475 nm) was measured every 5 min for 30 h with shaking between reads (548 rpm). Th-T fluorescence intensities were normalized to the plateau value. Th-T emission spectra were recorded at the end of the experiment between 470 to 600 nm (step of 5 nm) with a fixed excitation of 440 nm.

## Maintenance of *Caenorhabditis elegans* and in vivo model of α-Syn aggregation

The nematode *C. elegans* NL5901 pkls2386[Punc-54::α-Syn::YFP + unc-119(+)] was grown at 23 °C on 100 mm Nematode Growth Medium (NGM) agar plates seeded with 300 µl of an overnight culture of *E. coli* OP50 as a regular food source (Brenner, 1974). To analyze in vivo α-Syn aggregation, synchronized L4 worms were recovered from NGM plates and washed three times with M9. For a diet switch, L4 worms were incubated for 1 h on unseeded NGM plates with 50 µg/mL of kanamycin to remove residual *E. coli* OP50 and washed again. Next, L4 worms were transferred to freshly seeded NGM plates with either bacteria-producing amyloid-like fibers or OP50 supplemented with purified amyloid-like fibers. For the first type of diet, *E. coli* VS39 transformed with C-DAG system was grown on LB agar containing Amp (100 µg/ml), Cm (20 µg/ml), 0.2 % (w/v) L-arabinose and 1 mM IPTG to produce extracellular BAP-type amyloid fibers. After 2 days at 30 °C, bacteria were recovered from LB plates and resuspended to an OD$_{600nm}$ of 4 in M9. Then 500 µl of the resuspension was plated onto NGM agar and let dry before worms were placed on food. For the alternative diet, L4 worms were transferred to NGM plates seeded the day before with 300 µl of a mixture of an overnight culture of *E. coli* OP50 and 40 µg of

rBapB-PFF or rEsp_N-PFF preformed fibers. Under these conditions, adult worms were transferred to a new freshly seeded NGM plate every two days. At day 10, adult worms were recovered from NGM plates and washed three times with 10 ml of M9 for locomotive and α-Syn aggregation analysis.

## Locomotive analysis of *C. elegans*

Adult worms were transferred to an unseeded NGM plate to remove bacteria from the cuticle of the worms. After 1 h, ~10 worms per well were transferred to a 24-well plate with 200 µl of M9, allowed to recover for 2 min and then movies of swimming worms were recorded with the same settings for 1 min with a Zeiss Stemi 508 stereo microscope and Zeiss Axiocam 208 color camera. Recorded worms were tracked with the WrmTrck plugin from ImageJ (https://www.phage.dk/plugins/wrmtrck.html) to measure the body bends per minute (BBPM). At least two independent experiments were performed.

## Quantification of α-Syn aggregation in *C. elegans*

For fluorescence microscopy imaging, 1 ml of worm suspension was fixed with 4 % (v/v) paraformaldehyde (PFA) for 20 min at room temperature, washed with PBS and resuspended in 300 µl of PBS. Then, 50 µl of fixed worms were mounted onto glass slides and images were captured at 40X with a Leica DMi8 fluorescence microscope and Hamamatsu ORCA Flash 4.0 LT camera. Image processing was performed with Icy software (https://icy.bioimageanalysis.org/). α-Syn aggregates were quantified between the tip of the head and the end of the pharyngeal bulb, in at least 30 individuals per group, using Spot Detector tool from Icy software. For immunoblot analysis of α-Syn aggregation, 9 mL of the worm suspension were pelleted (800 × *g*, 1 min) and worm pellets were flash freeze with liquid nitrogen and stored at −80 °C. Thawed worm pellets were processed for sequential protein extraction[96]. Briefly, the nematode pellets were weighted and resuspended in 150 µl of cold RIPA lysis buffer (50 mM Tris pH 8.0, 150 mM NaCl, 5 mM EDTA, 0.5% (w/v) SDS, 0.5% (w/v) sodium deoxycholate, 1% (v/v) Triton X-100, 1 mM PMSF, 1x protease inhibitor cocktail) per 50 mg of weighted pellet. The suspension was then transferred to a 2 mL-Lysing Matrix Tube containing a 1:1 (v/v) proportion of glass beads (0.2–0.3 µm) and mechanically homogenized in a FastPrep-24™ 5G homogenizer (MP Biomedicals) with 10-s incubation on ice between cycles (4 cycles of 30 s at 5 m/s speed). Homogenates were centrifuged at 21,100 × *g* for 20 min at 4 °C and the supernatant was transferred to a clean tube and stored at −80 °C. The amount of proteins was determined by the Bradford Protein Assay (BioRad). 5 µg of proteins were loaded and separated electrophoretically in 12% [w/v] SDS-PAGE stain-free gel (BioRad) and then transferred to a nitrocellulose membrane for 30 min at 1 A. The membrane was fixed with PFA 4% (v/v) in PBS for 30 min at room temperature, washed in PBS and blocked overnight at 4 °C with 5% (w/v) non-fat milk diluted in PBS-T (PBS 1X with 0.1% (v/v) Tween 20). Total α-Syn:YFP was detected with a mouse anti-GFP (JL-8) primary antibody (Takara) diluted 1: 5.000 (v:v). Goat anti-mouse secondary antibody conjugated to horseradish peroxidase (Thermo) were diluted to 1:5.000. Antibodies were diluted in 4% (w/v) non-fat milk dissolved in PBS-T.

## Quantitative PCR from *C. elegans*

Total RNA from *C. elegans* was extracted using TRIzol reagent (Thermo Fisher). For transcriptional analysis, cDNA was reversed transcribed from total RNA using PrimeScript RT reagent kit (Takara) from synchronized L4 animals. Quantitative real-time PCR of α-Syn was performed using a TB Green Premix Ex Taq kit (Takara) in Applied BiosystemsTM QuantStudio 5 machine using primers H-aSyn-Fw (5′- CACAGGAAGGAATTCTGGAAGATA) and H-aSyn-Rv (5′-GCTTCAGGTTCGT AGTCTTGAT). Values were normalized to the

internal control tba-1using primers tba-1-qPCR-Fw (5′- AGGTAC-CACGTGCCGTATGT) and tba-1-qPCR-Rv (5′- TTATGCA-GATTGGAGCAGGCGG). All data shown represent the average of three independent replicates.

### SH-SY5Y cell culture and neuronal differentiation

Human neuroblastoma SH-SY5Y cells (ECAC: 94030304) constitutively expressing full-length human wild type α-Syn with a C-terminal HA tag were grown at 37 °C with 5% $CO_2$, in Dulbecco's Modified Eagle's Medium (DMEM)/F-12 with Glutamax 1x (Gibco) supplemented with 10% heat-inactivated fetal bovine serum (Sigma), 1% non-essential amino acids (MEM NEAA) (Gibco) and 1% penicillin/streptomycin (Gibco). To induce cell differentiation to dopaminergic neurons and assess bacterial amyloid effect on α-Syn aggregation, SH-SY5Y cells were seeded at $2.5 \times 10^3$ cells per well on 13 mm circular coverslips in 24-well plates containing growth medium supplemented with 30 μM retinoic acid (RA) (Sigma). RA-containing growth medium was replaced every 2-3 days for 6 days to achieve differentiation. The differentiated cells were then treated for 2 days with 10 μg/well (0.02 μg/μl) of rBapB-PFF and rEsp_N-PFF and cells were maintained until day 10 post-treatment with the replacement of the RA-containing growth medium every 2–3 days. For immunofluorescence microscopy, cells were washed in PBS, fixed with 4% (v/v) PFA for 20 min at room temperature, permeabilized for 15 min at −20 °C with 100% methanol, washed in PBS and incubated for 30 min at 37 °C with 10% normal goat serum (NGS) as blocking solution. Total α-Syn was stained with a mouse anti-α-Syn LB509 antibody (Invitrogen; 180215) diluted 1: 200 (v:v). Phosphorylated α-Syn was detected with a rabbit anti-pS129-α-Syn EP1536Y primary antibody (Abcam; ab51253) diluted 1:200 (v:v). DNA was stained with Hoechst 33342 diluted 1:1000 (v:v). Goat anti-rabbit conjugated with alexa-488 (Invitrogen; ref: A11008) and goat anti-mouse conjugated with alexa-568 (Invitrogen; ref: A11031) secondary antibodies were diluted 1:200 (v:v) in 2% NGS. After immunostaining, coverslips were washed in PBS and mounted onto glass slides. Fixed samples were imaged at 100X with a Leica DMi8 fluorescence microscope and Hamamatsu ORCA Flash 4.0 LT camera. Image processing was performed with Icy software. α-Syn aggregation was quantified in at least 25 cells per group using Spot Detector tool from Icy. Each experiment was performed in triplicate.

### Protein extraction and Western-blot analysis of differentiated SH-SY5Y cells

For Western-blotting, cells were seeded to $4 \times 10^4$ cells/well in 6-well plates. Differentiated SH-SY5Y cells were treated with 40 μg/well of rBapB-PFF and rEspN-PFF as described above. At day 10 post-treatment cells were scraped in cold PBS, placing the 6-well plate on ice. The cellular homogenates recovered from 3 wells were pooled and centrifuged at 1400 g for 5 min at 4 °C. Pellets were weighted, resuspended to 100 mg/ml in cold High-Salt Buffer (50 mM Tris base, 750 mM NaCl, 5 mM EDTA, 1x protease inhibitor cocktail and 1x phosphatase inhibitor cocktail [Thermo]), incubated 10 min at 4 °C and centrifuged at $18,800 \times g$, for 20 min at 4 °C. Supernatant corresponding to soluble fraction was stored at −80 °C. Pellets were resuspended to 100 mg/ml, according to cell homogenates weight, in cold HSB with 1% (v/v) Triton X-100, incubated 20 min at 4 °C and centrifuged at $21,100 \times g$, for 10 min at 4 °C. Pellets were washed in cold PBS and resuspended in cold SDS/urea (8 M urea, 2% (v/v) SDS, 1× protease inhibitor cocktail and 1x phosphatase inhibitor cocktail [Thermo], 50 μg/mL DNase) to 200 mg/mL according to cell homogenates weight, incubated 1h at 37 °C and centrifuged at $21,100 \times g$, 20 min at 4 °C. Supernatants corresponding to insoluble fraction were stored at −80 °C. Soluble and insoluble fractions were separated on 12% (w/v) SDS-PAGE and transferred to a nitrocellulose membrane for 30 min at 1 A. Membranes were fixed with PFA 4% (v/v)

in PBS for 30 min at room temperature, washed in PBS and blocked overnight at 4 °C with 5% (w/v) non-fat milk diluted in PBS-T. Total α-Syn was detected with α-Syn LB509 primary antibody (Invitrogen; 180215) diluted 1: 1000 (v:v). Phosphorylated α-Syn was detected with a rabbit anti-pS129-α-Syn EP1536Y primary antibody (Abcam; ab51253) diluted 1:1000 (v:v). Goat anti-rabbit and goat anti-mouse secondary antibodies conjugated to horseradish peroxidase (Thermo) were diluted 1:5000 and 1:2500 (v:v), respectively. Antibodies were diluted in 2 % (w/v) non-fat milk dissolved in PBS-T. LAMP-2A was detected using anti-LAM2A (Invitrogen 51-2200) (Fig. S14).

### Analysis of α-synuclein half life

α-Synuclein turnover was investigated in SH-SY5Y cells treated with BAP-amyloids by inhibiting protein synthesis with 50 μg/ml cycloheximide (Sigma-Aldrich). Samples were removed at 12-h intervals up to 48 h. Sample loadings from equal cell numbers were calculated using the fluorescent DAPI assay of DNA content. For immunoblotting, cell lysates containing equal cell numbers were loaded separated on NuPAGE Novex 4%-12% Bis-Tris Gel (Invitrogen) and transferred to a polyvinylidene difluoride membrane. Blots were incubated with anti-human α-synuclein antibody LB509 (1:1000) and then with peroxidase-conjugated goat anti-rabbit (1:5000) antiserum and enhanced chemiluminescence. Western blots of α-synuclein were analyzed using ImageJ analysis software, and half-life was calculated using a linear regression model; average half-life was calculated from regression lines drawn for each individual experiments. All experiments were performed in triplicate (Fig. S15).

### Mice and study design

Eight- to nine-week male C57BL6/C3H F1 mice were purchased from Charles River. All animals were randomly distributed in the cages and before any procedure, the cages were randomized to each group by a person not involved in the study. Animals were housed in individual polycarbonate cages with stainless steel covers, with a maximum of five mice per cage. All mice were allowed to ad libitum access to standard pellet diet (Special Diet Service, UK) and normal tap water, and mice were maintained in constant environmental conditions of humidity ($55 \pm 10\%$) and temperature ($22 \pm 2$ °C) on a 12-h light/dark cycle. All the in vivo experiments were blinded and the investigators responsible for data collection and analysis were blinded. Mice received two 2.5 μl injection of sonicated rBap-PFF in sterile PBS into the dorsal striatum (10 μg total; AP + 0.2, ML -2.0, DV − 3.4 and −2.6 from the skull) by stereotaxic injection. Control animals were injected with an equal volume of sterile PBS. Animals were sacrificed 7 months after injection.

### Behavioral assessments

To evaluate the motor function, mice were tested on wire hang, rotarod and negative geotaxis tests before treatment, at 30-day intervals during the study and prior to sacrifice. Tests were performed on the same animals. All tests were conducted between 09:00–13:00 h in the lights-on cycle to eliminate time of day differences in behavior and mice were allowed to rest for 30 min between tests. Mice were habituated to the testing room 1 h before tests and the apparatus were cleaned with 70% ethanol in between to minimize odor cues.

**Wire hang test.** To assess neuromuscular strength and motor coordination, the wire hang test was performed. Each mouse was placed on a wire lid of a conventional housing cage. The lid was lightly agitated to encourage the animals gripping of the bars and then was turned upside down. The latency of mice to fall off the wire grid was measured and averaged over two trials (15 min apart). Trials were stopped if the mouse remained on the lid for over 15 min.

**Negative geotaxis.** Motor coordination was assessed in an inclined plane. Each mouse was placed with its head facing downward on a wire grid that was set at a 45° angle to the plane. The behavior of the animal was observed during 30 s and scored as follows: 0 = turns and climbs, 1 = turns and freezes, 2 = moves, but fails to turn, 3 = does not move[97]. The latency to turn 180° to an upright position and initiate climbing was recorded for all animals that received a score of 0. If the mouse was unable to turn, the default value of 30 s was taken as maximal severity of impairment.

**Rotarod.** Mice were placed on the rod (Rotarod LE8205, Panlab) for over 300 s which accelerated from 4 to 40 rpm. The stay on time (s) of each mouse was recorded as the latency period to fall, and mice were test 4 times with 10 min rest intervals.

## Immunohistochemistry

30-μm-thick coronal brain sections were prepared using a cryostat. Free-floating sections were washed with TBS and were incubated with 3% hydrogen peroxidase to inactivate the endogenous peroxidase. Afterwards, sections were blocked in blocking solution (0.1 M TBS containing 5% normal goat serum and 0.04% Triton X-100) followed by incubation for 24 h at 4 °C with the primary antibodies: anti-Tyrosine Hydroxylase (Abcam; ref: ab112, dilution 1: 2.000), anti-α-Syn (phospho S129) (Abcam; ref: ab51253, dilution1: 1.000), anti-NeuN (Abcam; ref: ab104225 dilution 1: 5.000), anti-Iba-1 (Wako; ref: 019–19741, dilution 1:500), anti-GFAP (Sigma, ref: MAB360, dilution 1:1000). The next day, brain sections were rinsed and incubated with the biotinylated secondary antibodies of the appropriate species. After wash steps, sections were incubated with ABC Peroxidase Staining Kit (Thermo Scientific), washed again and revealed with a solution of 3,3'-diaminobenzidine substrate (Dako). Finally, brain sections were mounted onto microscopic slides, dehydrated and covered with DPX mounting medium and coverslips. The total number of dopaminergic (TH-positive) neurons in the substantia nigra pars compacta (SNpc) was assessed by stereology as previously described[98]. Striatal optical density (OD) of TH immunostaining was measured as an index of the density of striatal dopaminergic innervation. Six representative rostrocaudal sections (at three levels of the striatum: anterior, medial and posterior) were examined for each animal, and regions of interest in the striatum were delineated and pixel densities were estimated using ImageJ software. The presence of phosphorylated α-Syn inclusions were assessed at 20X magnification on coronal sections (120-μm intervals between sections) at multiple rostrocaudal levels corresponding to SNpc, frontal cortex and thalamus. Phosphorylated α-Syn staining was measured in representative rostrocaudal sections of SNpc, striatum and amygdala at 10X magnification. Regions of interest were delineated and pixel densities were estimated using ImageJ software. For evaluation of astroglial and microglial cells, sections were stained with GFAP and Iba-1 antibodies, respectively. Estimation of the percentage of immunostained area by Iba1 and GFAP-positive cells was carried out using the threshold method of the ImageJ software.

## Western blots

Frozen midbrain were homogenized in a 10 mM Tris/HCl (pH 7.4) buffer containing 0.1% sodium dodecylsulfate, a protease inhibitor mixture (Halt protease inhibitor cocktail, Thermo Scientific), phosphatase inhibitor (Halt phosphatase inhibitor cocktail, Thermo Scientific) and DNAse (RQ1 DNase, Promega). Proteins levels were determined by the Pierce BCA protein assay (Pierce BCA protein assay kit, Thermo Scientific) using bovine serum albumin as the standard. Tweenty μg of proteins were solubilised in LDS buffer (Invitrogen) and reducing agent (Invitrogen) and separated on NuPAGE Novex 4–12% Bis-Tris Gels (Invitrogen), transferred to PVDF membrane and analyzed by Western blot using the following primary antibodies: anti α-Syn LB509 (Invitrogen; 180215), anti-LAMP-2A (Invitrogen; 51-2200), and anti-beta-actin (Abcam; ab8227) antibodies. Membranes were incubated with horseradish peroxidase-conjugated secondary

## Quantitative PCR

Total RNA was isolated from midbrain samples using the RNeasy Mini kit (Qiagen) as per the manufacturer's protocol. Reverse transcription (RT) was performed with High -Capacity cDNA Reverse Transcription kit (Applied Biosystems). Transcripts obtained from the reverse transcriptase reaction were analyzed for *lamp-2a* and *hsc70* quantification by quantitative real-time PCR. qPCR experiments were performed on a StepOneTM Real-Time PCR system (Applied Biosystems) using NZY-Speedy qPCR master mix (NZYtech). Values were calculated using the standard ΔΔCt method relative to actin. The primer sequences for lamp-2a (forward: 5'-TATGTGCAACAAAGAGCAGA-3'; reverse: 5'-CAG-CATGATGGTGCTTGAGA-3') and hsc70 (forward: 5'-ATTGATCTTGG-CACCACCTA-3'; reverse: 5'-ACATAGCTTGAAGTGGTTCG3') and the sequence for mouse actin (forward: 5'-TCTACAATGAGCTGCGTGTG-3'; reverse: 5'-GGTGAGGATCTTCATGAGGT-3') α-Syn (forward: 5'-GCCAA GGAGGGAGTTGTGGCTGC-3'; reverse: 5'-CTGTTGCCACACCATGCA CCACTCC3').

## Metagenomic data reanalysis

We used the raw shotgun sequencing data obtained from individuals diagnosed with PD, along with Neurologically Healthy Controls (NHC) available at the NCBI Sequence Read Archive (SRA) repository under the BioProject ID PRJNA834801[46]. We performed a quality control checks on the raw sequencing reads using FastQC v0.12.1 to filter low quality and adapter-contaminated reads. We used FastQ Screen v0.15.3 to check the presence of other organism's contaminations within the samples. We proceeded to remove a Nextera transposase adapter using Cutadapt v4.34 with –trim-n, --quality-cutoff 25 and –a CTGTCTCTTATA –A CTGTCTCTTATA to clean forward and reverse reads from that adapter. We employed Centrifuge v1.0.4 for a general taxonomic profiling and functional annotation. The high-quality 150 nt pair end reads were aligned against a custom database, consisting of a reference genome collection comprising different bacteria that contained BAP gene region using BWA v0.7.17-r1188 (default parameters) (Table S2). The alignment process enabled us to determine the proportion of BAP presence in the samples by examining their positional information with SAMtools v1.157. Reads that contained less than 70 matches in their sequence were removed. We employed Pysam to assess the alignment quality of the reads against the reference database. The presence of BAP was determined as a binary variable for each sample and organism based on the total number of reads in BAP-like gene (TNBAP) TNBAP ≥ 1 was considered as BAP positive.

## Statistical analysis

To identify significant differences in specific features between two groups of samples, non-parametric Mann–Whitney test or a parametric $T$ test were used. To identify significant differences between more than two groups, non-parametric Kruskal–Wallis test or a parametric one-way ANOVA test were used. To identify statistically significant differences in BAP-like gene presence between PD and NHC groups for each organism, binary logistic regression was performed using the glm() function with binominal distribution in R. For the PD-associated BAP-positive species in which a significant association was found, the model was adjusted by age, sex, and technical variables (stool collection method and total sequence count (standardized). To identify statistically significant differences in RPKM between PD and NHC groups for each organism, linear regression was performed using MaAsLin2 test[58]. Multiple testing corrections were carried out using the Benjamini–Hochberg FDR method. No statistical method was used to predetermine the sample size. Data are shown as means, and error

bars are shown as the SE of means. Analyses were performed using GraphPad Prism® version 6. P values < 0.05 were considered significant Asterisks indicate the level of statistical significance: $*p < 0.05$, $**p < 0.01$, $***p < 0.001$.

## Reporting summary

Further information on research design is available in the Nature Portfolio Reporting Summary linked to this article.

## Data availability

Sequencing data and metadata are available online through European Nucleotide Archive (ENA) with the study accession PRJEB66343. Sequencing data from individuals diagnosed with PD and Neurologically Healthy Controls (NHC) were obtained from the NCBI Sequence Read Archive (SRA) repository under the BioProject ID PRJNA834801[46]. All the data presented in this study are available in the paper or in the Supplementary Information. Source data are provided with this paper. Additional data is available from the corresponding authors upon reasonable request. Source data are provided with this paper.

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

## Acknowledgements

The authors would like to thank the Microprogen consortium. We particularly acknowledge the patients for their participation and the Biobank of the University of Navarra for its collaboration. We thank Prof Iñigo Lasa and Paco García del Portillo for helpful critiques. We are grateful to Ainara Aginaga, Adrian Román, Samuel Peña and Idoia Glaria for technical support. Imaging was performed in Plateforme IBiSA de Microscopie Electronique Université de Tours Imaging Facility. L.M. was a Universidad Pública de Navarra predoctoral fellow. This research was supported by Grants PID2021-124248OB-I00 from the Ministry of Science and Innovation, and PC133-134-135 MICROPROGEN from the Government of Navarra. L.A.-E. and J.B. are supported by Miguel Servet contracts (CPII20/00027 and CP19/00200) from ISCIII.

## Author contributions

Conceptualization J.V., L.A.-E., A.T.-A. Designed the experiments, J.V., L.A.-E., S.V., A.T.-A. Investigation, A.F.-C., L.M.-C., M.S. Performed animals' experiments, M.I., L.A.-E. Performed neurobiological analysis, J.B., L.A.-E., M.I. Performed biophysical experiments, S.N., S.V. Human sample collection, M.H. Metagenome analysis, G.A.-A. Reanalysis of metagenomics data from NHC and P.D. S.G., I.R.dl.M., M.L.M., A.T.-A. Writing—Original Draft, J.V. Writing—Review and Editing, all authors; Supervision, J.V., L.A.-E., S.V., I.R.dl.M., A.T.-A. All authors approved the final manuscript.

## Competing interests

The authors declare no competing interests.
