## [Peer Review File · Nature Communications]

Reviewers' expertise and comments:

Reviewer #1. Bacterial amyloids / alpha Syn aggregation.

(Remarks to the Author):

The study by Fernandez-Calvet A, et al. identifies the presence of many bacterial species with amyloid forming biofilm-associated proteins (BAP) both in healthy individuals and in individuals with irritable bowel syndrome (IBS). Using various biophysical techniques, the authors characterize the ability of BAPs to form aggregates and show those aggregates can induce the aggregation of α -Synuclein. The authors also describe SDS-insoluble aggregates in fecal samples from individuals with IBS that are A11 antibody active. For the most part the experiments are nicely done but it was sometimes difficult to appreciate the logic that went into the work and I not fully convinced the work rises to the high levels of a Nature Communications article because at least some of the major take-home message (that bacterial amyloids can exacerbate alpha-synuclein pathologies) have been published previously by other groups.

1. The manuscript contains several editorial and grammatical hiccups that could be addressed with a careful review. There were parts of the manuscript that I needed to read several times before completely understanding the authors' intent. This could be cleaned up with careful and thorough grammatical revisions.
2. All of the plasmid constructs used for inducing biofilm formation are under the control of either IPTG-- or CdCl₂--regulated promoters. Is there a possibility that the overexpression of the BAP proteins forces the amyloid assembly on the bacterial cell surface?
3. The protein variants used in the study were tagged with either a curli-dependent amyloid generator (C-DAG) or the carboxy terminal domain of clumping factor A (R_ClfA) to target the proteins to the cell surface. Can the BAPs identified in the study get to the cell surface themselves or do they require the added tags?
4. Table S2 is discussed before Tabel S1. Similarly, Fig. 2A appears in the text before discussing all the panels in Figure 1. It would be good to rearrange the figures/tables in the order that they are discussed in the text.
5. In the Discussion it was stated that "the production of curli amyloids in vivo has been questioned because their expression is induced in the laboratory at temperatures below those found in the body". The Congo red plate shown in Fig. 2B was grown at 30 °C. Can the authors comment on that? Was BAP expression investigated at 37 °C?
6. Table S5 is missing.
7. In the Methods section, "Generation of chimeric proteins and biofilm phenotype: BAP-domains were amplified using primers shown in Table S1". But there are no primers listed in Table S1.

8. For the cross seeding assays, pre-formed fibrils were sonicated for an hour prior to the experiments. Was there any protein degradation/shearing after prolonged sonication?
9. The authors mentioned that “As a control we used curli fibers that reacted with anti-A11 antibodies, but not with anti-EN (Fig. S6)”. But A11 binding to CsgA aggregates was rather weak (Fig. S6B). It was previously shown by Wang X, et al. *J. Biol. Chem*, 2007, 282, 3713-19 that CsgA oligomers binds to A11 antibody than CsgA fibrils. There might be other publications that show A11 binding to curli-like aggregates that I am not familiar with or remembering, but it might be worth figuring out exactly what A11 is binding to for these BAPs.
10. For the ease of reading, it would be good to provide abbreviation expansions below Table S9
11. In the discussion it was mentioned “accumulation of endogenous phosphorylated α -Syn, a post-transcriptional modification associated with PD pathology”. Change post-transcriptional to post-translational.
12. It would be good to include page numbers.

Reviewer #2. Microbiome and neuro diseases / alpha Syn.

(Remarks to the Author):

This is a conceptually interesting article describing the ability of biofilm associated proteins (BAPs) to interact with alpha-synuclein and accelerate its accumulation in model systems. This work builds on a plethora of existing literature surrounding interactions between heterologous amyloidogenic proteins (aSyn, Abeta, Tau, SAA, IAP, etc, etc). Cross-kingdom interactions are more limited to studies of curli/CsgA from Enterobacteriaceae and FapC in Pseudomonas. This study extends the literature to include potential capabilities of the BAPs in an interaction with aSyn. While initially enthusiastic, given what is already known in the field regarding such cross-kingdom interactions, I was anticipating a more in depth and mechanistic study. In conjunction with some significant technical concerns, despite the conceptual interest, I would not support the publication of this manuscript in its current state.

Major

Mechanism of BAP-synuclein interactions

While the assays displayed begin to show the association of BAP with aSyn accumulation (and some aggregation), how this is occurring is not clear. Do BAPs alter aSyn expression? Do they impact proteostasis? Are they actively cross-seeding? Unfortunately, even with the multiple model systems used, there is not much depth in this regard to begin interpreting how a BAP-induced acceleration might be occurring.

A) For instance, in Fig 4A- it is unclear why there is no YFP detected in the transgenic worms treated with the null vector *E. coli*. Shouldn't expression by this constitutive muscle promoter be the same across groups, and thus we should see the same amount of total synuclein expression in the muscle tissue, rather than see a larger total amount with the different BAPs? Some clarity on the total amount of synuclein formed here across the different lines, as well as gene expression by qPCR (or similar) to understand where this striking effect might be occurring?

B) In Fig 5A-C. Dose curves are needed, varying the amount of recombinant BAP fibrils, to identify concentration dependence and more conclusively show the potential acceleration. Do monomers of the BAPs have the same effect, or must these be in a PFF state? Controls without monomeric aSyn are also needed to see how the BAPs alone alter the ThT signal. Both these would help support interpretations of cross-seeding and begin to identify possible mechanistic activities (Are BAPs catalytic? Are they incorporated?)

C) Do the BAPs co-localize with aSyn in the in vitro models? This would also help support a cross-seeding event, rather than BAP-mediated impacts on proteostasis or inflammatory pathways that could result in aSyn accumulation as well. In the discussion (pg 15, line 27-30) the authors state that they show a co localization of aSyn with BAPs, however this does not appear to be supported by any direct experimental data?

D) Further characterization of the in vivo outcomes would also help support possible mechanisms of action of BAPs on synuclein. These in vivo observations should be bolstered with further molecular characterization- western blots for pSyn and detergent soluble aSyn for instance, qPCR for aSyn expression, etc. All of which might support direct interactions with existing murine synuclein or effects on other cellular/molecular pathways which could more indirectly effect synuclein accumulation in mice.

As an example, the endpoint observation of equivalent area of GFAP and Iba helps to support a claim that neuroinflammatory activation is not involved, however, earlier timepoints would better show this. BAPs may incite a strong, yet acute, inflammatory activation that triggers the subsequent degeneration and pathology, before being resolved by the endpoint where this Iba/GFAP equivalency is observed. Controls with non-amyloidogenic BAP would help to confirm this too. Analysis of cell culture models for inflammatory pathway activation or cytokine release would also help to clarify a role for inflammatory activation.

Presence of BAPs in human microbiomes

A) The mining of the existing, large scale metagenomic study of PD is very useful. I am slightly confused regarding the analysis however. As an example, the authors show in Fig 7 that the BAP encoded by *Lactobacillus johnsonii* is present in 83 PD vs 17 Control samples. However, the study

these data come from states that *L. johnsonii* is present in only 11 PD vs 4 Control samples (Supplementary Data Table 1 of the published, cited study). Similar discrepancies appear with *E. faecalis* (44 vs 7 here and 42 vs 11 in the cited study), and likely others. Clarity is needed in the differences in the analysis that would detect, for instance, so much more *L. johnsonii* in the same dataset (8x more here). In the case of LJ, are the BAP_LJ gene coming from other similar species and being combined? The reason for prevalence differences are too unclear.

B) Perhaps showing the bacterial species specifically and the predicted encoded BAPs side by side would help to clarify the differences between the presence of the organism and the genes of interest. I do understand that orthologs may be encoded by species that are not fully annotated and that strains of the same species may not have the specific gene. In those cases, the authors should be careful in their discussion of bacterial species vs the BAP itself (such as on pg 12 lines 7-11, re: Fig 7).

C) A discussion around the implications of the prevalence of BAPs between PD and control, vs the limited differences in actual abundance of those BAPs in those who are carrying them, irrespective of their PD status. Are the presence of BAP a potential trigger or a consequence?

D) There are also some discrepancies between the sequencing and the blotting in Fig 1 vs Fig 3. For instance, MB23984 has no reactivity for any of the tested amyloids (including EspN and BapLA), yet it was one of the higher samples for gene content with EspN. This is true for other samples as well – MB24377 for instance is also high in EspN and BapLA by gene content, but shows no signal in the detection assay. How well do the sequencing data align with the protein measurements? It suggests that encoding does not correspond to functional protein here, or that perhaps the amyloid staining in Fig 3 is not reacting to the bacterial amyloids.

E) It is fairly well established that injection of protein aggregates of various types into the brain, or other anatomical sites, can result in amyloid pathology. Given these are proteins produced in the gut, what is the biological relevance of injecting them into the brain?

Specificity for the amyloidogenic properties of the BAPs

A) In many instances, BAPs are compared to PBS buffer controls in all of the model systems, from the in vitro aggregation assays to the mouse intracerebral injections. Rather than PBS alone, perhaps the use of a BAP that does not form amyloid as readily (like CM2?) could be a useful control for amyloidogenic capacity and specificity or a mutant BAP that does not form amyloid.

B) Similarly, purified, monomeric BAP (rather than PFF) would be essential in the cell culture and mouse systems, building on what is understood in the aSyn PFF models. Do monomers never induce pathology? Or do they simply take longer?

Minor (generally in order of appearance)

In Fig 2B there are no indications of which colonies 1-26 are the controls and the tested proteins. I think it corresponds to the list presented in Supplemental Fig S2, but this was difficult to interpret as a reader.

Fig S2 shows significance compared to the blank, but this seems a bit unfair. Wouldn't the better comparison be to the vector/expression controls of BapA and EspRC?

The relevance of anti-His immunogold reactivity is unclear to me Fig S3. Is this region histidine rich? Perhaps a phrase to explain to non-experts the relevance of this observation in the results.

Fig 3- controls are needed for specificity. Stool samples from a microbe free environment (germ-free mice?) for instance? To ensure that the antibodies indeed do not react to epitopes derived from host tissues. Cross-reactivity between amyloid-detecting antibodies can occur. Given the reactivity comes almost completely from individuals with IBS, could this be a reaction to something from the host rather than the microbes? Synuclein? Also a technical question, as I am not familiar with these specific types of blots. But in western blotting, the lighter color can sometimes indicate too strong reactivity that then blocks the detection substrate. How is equal loading confirmed? Could samples be run with and without a formic acid treatment to ensure some specificity of the amyloid signal?

PBS controls in Fig 5C do not appear to have the same N displayed as in 5B (2 vs 3).

Fig 5H, whole blots are needed. Are oligomeric forms of aSyn observed in these fractions? Was pS129 probed from the same blot as the total aSyn? Are the observed bands forming at the same MW between the two targets? This would also help support possible mechanistic inferences.

The quantification of pSyn density in Fig6H is not clear. Given the sparsity of aggregates seen in Fig 6G, counts should be provided- are these in TH+ cells?

Pg 13, lines 16, 20 (and possibly elsewhere)- Gram should be capitalized.

Citation 77 does not support a prion like spread of BAPs or any protein (discussion, pg 16 line 11)

Reviewer #3. Bacterial amyloids / Structural and functional analyses.

(Remarks to the Author):

Fernández-Calvet and coworkers report a comprehensive study the prevalence of Biofilm associated protein (Bap) – like bacterial amyloid sequences in the human intestinal microbiota, and evaluate their potential neurodegenerative activity.

Through a phylogenomic mining of sequenced gut microbiomes the authors identify a series of Bap-like proteins and map associated candidate amyloidogenic degradation fragments. They then analyse their prevalence in the metagenomes of 35 individuals with irritable bowel syndrome and 14 control individuals, finding a high prevalence for some of the identified proteins. A selection of the identified protein fragments are cloned into a curli-based export system to monitor production of extracellular amyloid-like fibers, and cloned for recombinant expression. ThT, CR, EM and FTIR are used to monitor the presence and extend of amyloid formation on cells or in vitro using purified recombinant protein fragments.

The authors demonstrate that some of the identified Bap fragments are amyloidogenic and that stools of patients have variable levels of A11-reactive fibers. They then go on to evaluate the cross-seeding activity towards alpha-Synuclein in vitro, as well as in cellulo and in mice. Doing so, the authors identify the relatively widespread presence of amyloidogenic proteins in the human gut microbiome, and demonstrate their principle potential to facilitate the formation of amyloidogenic deposits of alpha-Syn.

The setup and findings of the study are novel and will be of interest to the growing community seeking to determine the potential of gut microbiota and their proteomes to enhance human protein aggregation disorders and associated pathologies. The study is generally well executed and documented. A few points need further clarification prior to publication.

Specific points of attention:

- 1) The alpha-Syn cross-seeding experiments use 1 μ M Bap pre-formed fibrils or 10 nM active human recombinant a-Syn-PFF. Did the authors try lower concentrations of rBap? It would be important to determine if there is a critical concentration to the Bap – a-Syn cross-seeding activity.

2) Furthermore, in the cellular or in vivo experiments, cells were exposed to 10 µg rBap-PFF / well (please provide a concentration) or mice were injected with 10 µg rBap-PFF / mouse, resulting in a quite high local concentration. How does this relate to expected circulating concentrations of rBap fragments? Did the authors try lower doses? It is advisable to include a discussion on the concentrations used and how they relate to expected in vivo conditions, and whether there is any critical minimal concentration to induce disease.

3) Control animals were injected with PBS. Can the authors exclude an inflammatory activity of endotoxin co-purified with their recombinant Baps? The authors should include an injection of a recombinant non fiber forming Bap fragment as a control.

Additional points of attention:

Page 3 - Ln 24-26: The term 'facultative' bacterial amyloid was coined by Van Gerven et al. 2018 J. Mol. Biol. 430, 3657–3684.

Page 3 - Ln32: gram-positive and gram-negative should be Gram-positive and Gram-negative. Check throughout text.

Page 4 – Ln 3: The authors write: "Amyloid protein folding patterns are highly conserved in human and bacterial amyloids." What is this statement based on? More caution should be made in this statement. To date, the only structures available are for the obligate bacterial amyloid curli (Sleutel et al. 2023, Nat Commun 17;14(1):2822), which forms beta-solenoids rather than the serpentine amyloid folds seen in pathological amyloids. This may be different in facultative bacterial amyloids, where the amyloid state is an off-pathway beta aggregation of often protein fragments, and does not form the native fold of the amyloidogenic proteins. There are however no structural studies to date to confirm or disconfirm this claim.

This statement on structural similarity is repeated in the Discussion, Page 15, Ln10-11. Statements on cross-seeding ability are correct, but any claims of structural similarity should be toned down and be rephrased.

Ref. 21 needs formatting: "Rogers, G. B. et al. From gut dysbiosis to altered brain function and mental illness: mechanisms and pathways. 21, 738–748 (2016)."

We thank the editor and all the reviewers for the comments and suggestions on our manuscript. We have tried to address all of the major points with new experiments in order to improve the presentation of the work.

Reviewers' expertise and comments:

Reviewer #1. Bacterial amyloids / alpha Syn aggregation.

(Remarks to the Author):

The study by Fernandez-Calvet A, et al. identifies the presence of many bacterial species with amyloid forming biofilm-associated proteins (BAP) both in healthy individuals and in individuals with irritable bowel syndrome (IBS). Using various biophysical techniques, the authors characterize the ability of BAPs to form aggregates and show those aggregates can induce the aggregation of α -Synuclein. The authors also describe SDS-insoluble aggregates in fecal samples from individuals with IBS that are A11 antibody active. For the most part the experiments are nicely done but it was sometimes difficult to appreciate the logic that went into the work and I not fully convinced the work rises to the high levels of a Nature Communications article because at least some of the major take-home message (that bacterial amyloids can exacerbate alpha-synuclein pathologies) have been published previously by other groups.

Author response:

We greatly appreciate the time and effort the reviewer has spent reviewing the manuscript, their careful criticism and their constructive comments. However, we disagree about the lack of novelty of our manuscript regarding bacterial amyloids and alpha-synuclein pathologies. Although some studies have shown that bacterial amyloids, specifically curli amyloids, exacerbate α -syn pathologies, our work goes a step further:

- We show that bacteria inhabiting the human gut produce biofilm-associated proteins (BAPs) with the potential to form amyloids. Importantly, we used clinical samples to prove the presence of BAP amyloid-like fibrils in the insoluble fraction isolated from the human gut. We have performed novel experiments that reinforce the observation of amyloid nature of the purified fibres. To the best of our knowledge, this is the first demonstration of bacterial amyloids in the human gut microbiota.
- Another relevant difference is that our study includes the reanalysis of metagenomic data from Parkinson's disease patients and neurologically healthy controls and show that the abundance of certain BAP-encoding genes in the gut microbiome correlates with Parkinson's disease. BAP-encoding genes appear to be located at the accessory parts of the genomes of the microbiome, suggesting that only certain strains would have the potential to produce BAP amyloids. This underscores the importance of analysing the genomic content of the microbiota rather than focusing on the presence of specific bacterial species. Our results may have important implications for earlier diagnosis and more effective therapies targeting the initial stages of the pathology.
- In the new version of the article, we investigated in depth the mechanism of BAP- α -Syn interactions. We demonstrated that BAP amyloids colocalise with α -Syn expressed by human-derived dopaminergic cells and seed α -Syn aggregation. We also found that BAP amyloid exposure significantly increased the α -syn half-life. The decrease in α -syn turnover is associated with a decrease in LAMP-2A levels, suggesting that chaperone-mediated autophagy activity is compromised by BAP amyloids in cell and animal models, a situation described and associated with Parkinson's disease.

Based on all these results, we consider that our work contributes new knowledge to the field of bacterial amyloids and their relationships to diseases.

1. The manuscript contains several editorial and grammatical hiccups that could be addressed with a careful review. There were parts of the manuscript that I needed to read several times before completely understanding the authors' intent. This could be cleaned up with careful and thorough grammatical revisions.

Author response:

We have tried to improve the readability of the manuscript as a whole and the text has been carefully revised and corrected by Springer Nature Author Services. We have included the Editing Certificate.

2. All of the plasmid constructs used for inducing biofilm formation are under the control of either IPTG-- or CdCl₂--regulated promoters. Is there a possibility that the overexpression of the BAP proteins forces the amyloid assembly on the bacterial cell surface?

Author response:

The reviewer suggests that overexpressing BAP with inducible promoters may force the formation of amyloid structures on the bacterial cell surface. Although this is an interesting assessment, we respectfully disagree with the reviewer for the following reasons:

-Previous works from our group demonstrated that *Staphylococcus aureus* and *Enterococcus faecalis* wild-type strains produced BAP-related amyloids without overexpressing *bap*-encoding genes (Response Figure R1) (Taglialegna et al., 2016; Taglialegna et al., 2020).

Response Figure R1. Immunogold labeled electron micrographs BAP amyloid fiber formed by *S. aureus* V329 cells (A and B) and *E. faecalis* 11279 (C and D). Images were taken from doi.org/10.1371/journal.ppat.1005711 and doi.org/10.1038/s41522-020-0125-2 respectively. Scale bar of panel A represents 100 nm. Scale bar of panels B, C, D represents 200 nm.

-The inducible promoter of the pCN51 plasmid shows a basal expression in the absence of the inducer (cadmium) (Charpentier., et al., 2004). This feature has already been used to produce basal expression of certain targets (Villanueva, et al., 2018; Villanueva et al., 2021). Therefore, we performed a biofilm formation assay in the absence of the cadmium in the culture medium. The results showed that the basal expression of chimeric BAP proteins in *S. aureus* Δ *bap* also induced biofilm formation. We have included these results as part of Supplementary Figure S4.

Figure S4. Expression of BAP N-terminal domains is sufficient to mediate biofilm phenotype in a heterologous strain. A) Bacterial clumping of *S. aureus* Δ bap mutant expressing chimeric N-terminal domains of BAPs. Images of the bottom part of the tubes are shown (bottom panel). Cultures were grown overnight in TSBg supplemented with 1 μ M CdCl₂ under shaken conditions (200 rpm) at 37 °C. B) Biofilm formation of *S. aureus* Δ bap cells that express the BAP N-terminal domains. For biofilm formation, bacteria were cultured overnight in TSBg supplemented with cadmium 1 μ M CdCl₂ at 37 °C in microtiter plates under static conditions. C) Biofilm formation without cadmium supplementation. As positive controls, Bap_B and Esp_N chimeric protein were used. As negative control, a chimeric protein using the nonamyloid domain Bap_CM2 was constructed. Data are shown as means, and error bars are shown as the SE of means. Statistically significant differences were determined using non parametric Kruskal Wallis test * $p < 0.05$, ** $p < 0.01$, *** $p < 0.001$.

-Finally, we included the biofilm formed by a strain expressing the nonamyloid domain Bap_CM2, as a negative control. Overexpression of the nonamyloidogenic domain in the pCN51 plasmid did not induce biofilm formation. This result suggests that only the overexpression of domains with amyloid properties induce biofilm formation. We have included this control in the new version of Figure S4.

3. The protein variants used in the study were tagged with either a curli-dependent amyloid generator (C-DAG) or the carboxy terminal domain of clumping factor A (R_ClfA) to target the proteins to the cell surface. Can the BAPs identified in the study get to the cell surface themselves or do they require the added tags?

Author response:

By using SignalP 6.0, we found that many of these proteins contain standard secretory signal peptides transported by the Sec translocon and cleaved by signal peptidase I (Lep) (Sec/SPI) (Table R1). We have included this information in Table S1. To confirm that BAPs are exposed to the cell surface without the addition of any tag, we performed immunofluorescence of *S. aureus*, *E. faecalis*, *L. lactis* and *L. acidophilus* wild-type strains using anti-Bap, anti-Esp, anti-BapLL and anti-BapLA antibodies, respectively. As shown in Figure R2, Bap, Esp, BapLL and BapLA are localised at the cell surfaces of *S. aureus*, *E. faecalis*, *L. lactis* and *L. acidophilus*, respectively (Response Figure R2). These results showed that BAPs are exposed on the cell wall without the need to add tags for secretion or anchor them to the cell wall.

Table R1

BAP	Signal peptide					
	Other	SP (Sec/SPI)	Lipoprotein SP (Sec/SPII)	TAT SP (Tat/SPI)	TAT Lipoprotein SP (Tat/SPII)	Pilin-like SP (Sec/SPIII)
Bap	0.0008	0.9984	0.0002	0.0002	0.0002	0.0002
Bap SH	0.0004	0.9989	0.0002	0.0002	0.0001	0.0001
Bap TG	0.7001	0.2903	0.0069	0.0009	0.0008	0.0001
Esp	0.0965	0.9006	0.002	0.0004	0.0003	0.0003
Bap EFm	0.0242	0.9737	0.0013	0.0004	0.0002	0.0002
Bap SS	0.0182	0.9729	0.0079	0.0004	0.0003	0.0003
Bap LL	0.3723	0.6067	0.0112	0.0066	0.0022	0.001
Bap LA	1	0	0	0	0	0
Bap CI	0.0025	0.9963	0.0005	0.0003	0.0002	0.0002
Bap FM	0.0449	0.9468	0.007	0.0006	0.0004	0.0003
Bap DF	1.0001	0	0	0	0	0
Bap CC	1.0001	0	0	0	0	0
Bap AB	1	0	0	0	0	0
Bap ECl	1.0001	0	0	0	0	0
Bap CF	1.0001	0	0	0	0	0
Bap PA	1	0	0	0	0	0
Bap CM	1.0001	0	0	0	0	0
Bap EFg	1	0	0	0	0	0
Bap ECo	0.9999	0.0002	0	0	0	0

The SignalP 6.0 server were used to predict the presence of signal peptides (SP)

Response Figure R2. Immunofluorescence showing surface localization of BAP proteins. *S. aureus* V329, *L. acidophilus* BL17, *L. lactis* MG1363 and *E. faecalis* 65140D7 cells were fixed and labelled with anti-Bap, anti-BapLA, anti-BapLL and anti-Esp antibodies, respectively and DAPI.

4. Table S2 is discussed before Tabel S1. Similarly, Fig. 2A appears in the text before discussing all the panels in Figure 1. It would be good to rearrange the figures/tables in the order that they are discussed in the text.

Author response:

The reviewer is correct. We have rearranged figures and supplementary tables accordingly.

5. In the Discussion it was stated that “the production of curli amyloids in vivo has been questioned because their expression is induced in the laboratory at temperatures below those found in the body”. The Congo red plate shown in Fig. 2B was grown at 30 °C. Can the authors comment on that? Was BAP expression investigated at 37 °C?

Author response:

The reviewer makes an interesting assessment. To promote slowed bacterial growth and reduce the toxicity of amyloids, a Congo Red binding assay on agar plates was conducted at 30 °C. However, the quantification of CR bound to bacteria expressing BAP was performed at 37 °C (Supplementary Figure S2A). For that, overnight cultures of the *E. coli* collection that export the predicted BAP amyloidogenic domains were diluted in LB medium supplemented with antibiotics (Amp100 µg/ml and Cm 20 µg/ml), inducers (L-arabinose 0.2% and IPTG 1 mM) and CR dye and placed into clear-bottomed black 96-well plates. A BioTek Synergy H1 microplate reader was used to analyse CR fluorescence and bacterial growth at 37 °C. The results showed that BAP also formed amyloid-like structures at 37 °C since most of the strains bound the CR dye. The assay was performed at 37 °C, as indicated in the legend of Figure S2.

6. Table S5 is missing.

Author response:

Table S5 has been included as Supplementary Dataset

7. In the Methods section, “Generation of chimeric proteins and biofilm phenotype: BAP-domains were amplified using primers shown in Table S1”. But there are no primers listed in Table S1.

Author response:

The primers used to amplify the BAP domains are listed in Table 1 (from MIC 1189 to MIC 2826). However, to circumvent any uncertainty regarding the presence of the primers, we have included them in a separate section of the Table 1 (now Table 10).

8. For the cross seeding assays, pre-formed fibrils were sonicated for an hour prior to the experiments. Was there any protein degradation/shearing after prolonged sonication?

Author response:

The reviewer is correct. To perform the cross-seeding assay, recombinant BapB and EspN proteins were incubated under defined conditions to generate preformed fibrils (BapB-PFF and EspN-PFF). These fibrils were then sonicated (1 hour) to generate the seeds. We used a long sonication time based on previous works (Polinski et al., 2018; Creed et al., 2022). We did not perform shorter sonication periods because the 1-hour sonication resulted in cross-seeding of monomeric α -Syn. However, considering the reviewer's comment, we have included a control using nonsonicated BAP fibres. In this case, we observed that the intact BAP fibres produce a weaker cross-seeding effect of monomeric α -Syn aggregation than sonicated preformed fibrils. We have included these results as a new Supplementary Figure S7.

Figure S7: Aggregation kinetics by measuring Th-T fluorescence over time. A) Th-T fluorescence of the seeds obtained after sonication (60 minutes) of rBapB-PFF, rEspN-PFF and α -syn-PFF to show that the seeds alone do not alter Th-T signal. As positive control aggregation kinetic of monomeric α -syn was shown. B) Aggregation kinetics of monomeric α -syn in the absence or presence of the indicated amount of seeds obtained after sonication (60 minutes) of rBapB-PFF, rBapB-PFF nonsonicated or monomeric rBapB (rBapB-MON).

9. The authors mentioned that “As a control we used curli fibers that reacted with anti-A11 antibodies, but not with anti-EN (Fig. S6)”. But A11 binding to CsgA aggregates was rather weak (Fig. S6B). It was previously shown by Wang X, et al. J. Biol. Chem, 2007, 282, 3713-19 that CsgA oligomers binds to A11 antibody than CsgA fibrils. There might be other publications that show A11 binding to curli-like aggregates that I am not familiar with or

remembering, but it might be worth figuring out exactly what A11 is binding to for these BAPs.

Author response:

The reviewer is completely correct. We purified CsgA fibres from *E. coli* VS39 pExport-CsgA using the protocol described in (Dueholm, M. S., et al. 2013). We showed that the CsgA mature fibrils were labelled with OC antibody, but not with A11 antibody (Fig. S5D and S5E). Prompted by the reviewer's comment, we also conducted an assay to analyse the binding of the A11 antibody to Esp amyloids. We purified the insoluble fraction from *E. coli* VS39 pExport-Esp, using the protocol of Dueholm, M. S., et al. The results showed that the A11 antibody can recognise intermediate amyloid folding stages of Esp, which were sensitive to formic acid, while the OC antibody is able to slightly detect mature Esp fibres (Fig. S5D and S5E). These results confirm that the signal detected by the A11 antibody in the insoluble fraction of the human faecal samples could correspond to intermediate amyloid oligomers of Esp. We have included these results as part of the new version of Figure S5.

Figure S5. BAP amyloid-like structures are present in the human fecal microbiota. A) Quantification of BAP-related amyloids from microbiota. The relative signal intensity of the blots was determined by densitometry using Image Lab Software. GraphPad Prism Version 6 was used for density analysis. B) Insoluble protein fractions purified from human stool samples were treated with 100% formic acid (FA) or a solution of trifluoroacetic acid/hexafluoroisopropanol (TFA:HFIP) (1:1). After treatment, samples were subjected to vacuum filtration through a 96-well dot blot apparatus with an acetate cellulose membrane. Immunoblotting using anti Esp_N antibodies showed no retained aggregates. C) Detection of Esp amyloids in stool samples of mice subjected to a broad-spectrum antibiotics treatment to deplete the gut microbiota (∅). After antibiotics treatment, rEspN_PFF were orally administered for 2 days. The SDS resistant fraction purified from mice faecal samples were subjected to a filter retention assay. Immunoblotting was performed using anti Esp_N antibodies. D) Detection of CsgA and Esp amyloids purified from the insoluble fraction of *E. coli* VS39 pExport-CsgA (IF_CsgA) and *E. coli* VS39 pExport-EspN (IF_EspN) using the anti-amyloid oligomers (A11) and the anti-amyloid fibrils OC antibodies. Samples were treated (+FA) or not (-FA) with formic acid. E) Immunodetection of CsgA and Esp in the purified insoluble fraction using anti-His and anti-Esp antibodies, respectively. Mature fibres of CsgA and amyloid oligomers of Esp were marked with arrows. Monomers of CsgA and Esp were marked with asterisks.

10. For the ease of reading, it would be good to provide abbreviation expansions below Table S9

Author response:

As proposed by the Reviewer, we have added abbreviation expansions to Table S9 (now Table S7) and also to Tables S10 and Table S11 (now Table S8 and Table S9 respectively).

11. In the discussion it was mentioned “accumulation of endogenous phosphorylated α -Syn, a post-transcriptional modification associated with PD pathology”. Change post-transcriptional to post-translational.

Author response:

We apologize for the mistake. We have change post-transcriptional to post-translational.

12. It would be good to include page numbers.

Author response:

Done

References:

- Charpentier, E., Anton, A. I., Barry, P., Alfonso, B., Fang, Y., & Novick, R. P. (2004). Novel Cassette-Based Shuttle Vector System for Gram-Positive Bacteria. *Applied and Environmental Microbiology*, 70(10), 6076–6085.
- Creed, R. B., et al. (2022). Analysis of hemisphere-dependent effects of unilateral intrastriatal injection of α -synuclein pre-formed fibrils on mitochondrial protein levels, dynamics, and function. *Acta Neuropathologica Communications*, 1–19
- Dueholm, M. S., Søndergaard, M. T., Nilsson, M., Christiansen, G., Stensballe, A., Overgaard, M. T., et al. (2013). Expression of Fap amyloids in *Pseudomonas aeruginosa*, *P. fluorescens*, and *P. putida* results in aggregation and increased biofilm formation. *MicrobiologyOpen*, 2(3), 365–382.
- Polinski, N. K., et al. (2018). Best practices for generating and using alpha-synuclein pre-formed fibrils to model Parkinson's disease in rodents. *Journal of Parkinson's Disease*, 8(2), 303–322.
- Taglialegna, A., Navarro, S., Ventura, S., Garnett, J. A., Matthews, S., Penadés, J. R., et al. (2016). Staphylococcal Bap proteins build amyloid scaffold biofilm matrices in response to environmental signals. *PLoS Pathogens*, 12(6), e1005711.
- Taglialegna, A., Matilla-Cuenca, L., Dorado-Morales, P., Navarro, S., Ventura, S., Garnett, J. A., et al. (2020). The biofilm-associated surface protein Esp of *Enterococcus faecalis* forms amyloid-like fibers. *NPJ Biofilms and Microbiomes*, 6(1), 15–12
- Villanueva, M., García, B., Valle, J., Rapun, B., Ruiz de Los Mozos, I., Solano, C., et al. (2018). Sensory deprivation in *Staphylococcus aureus*. *Nature Communications*, 9(1), 523.
- Villanueva, M., Roch, M., Lasa, I., Renzoni, A., & Kelley, W. L. (2021). The role of ArlRS and VraSR in regulating ceftaroline hypersusceptibility in methicillin-resistant *Staphylococcus aureus*. *Antibiotics*, 10(7), 821.

**Reviewer #2. Microbiome and neuro diseases / alpha Syn.
(Remarks to the Author):**

This is a conceptually interesting article describing the ability of biofilm associated proteins (BAPs) to interact with alpha-synuclein and accelerate its accumulation in model systems. This work builds on a plethora of existing literature surrounding interactions between heterologous amyloidogenic proteins (aSyn, Abeta, Tau, SAA, IAP, etc, etc). Cross-kingdom interactions are more limited to studies of curli/CsgA from Enterobacteriaceae and FapC in Pseudomonas. This study extends the literature to include potential capabilities of the BAPs in an interaction with aSyn. While initially enthusiastic, given what is already known in the field regarding such cross-kingdom interactions, I was anticipating a more in depth and mechanistic study. In conjunction with some significant technical concerns, despite the conceptual interest, I would not support the publication of this manuscript in its current state.

**Major
Mechanism of BAP-synuclein interactions**

While the assays displayed begin to show the association of BAP with aSyn accumulation (and some aggregation), how this is occurring is not clear.

Do BAPs alter aSyn expression?

Do they impact proteostasis?

Are they actively cross-seeding? Unfortunately, even with the multiple model systems used, there is not much depth in this regard to begin interpreting how a BAP-induced acceleration might be occurring.

Author response:

We thank the reviewer for the highly valid comments related to the mechanisms of BAP- α -Syn interactions. To address these points, we have performed new experiments in an attempt to determine whether BAPs alter α -Syn expression, impact proteostasis, interact with α -Syn or induce an immune response.

A) For instance, in Fig 4A- it is unclear why there is no YFP detected in the transgenic worms treated with the null vector E. coli. Shouldn't expression by this constitutive muscle promoter be the same across groups, and thus we should see the same amount of total synuclein expression in the muscle tissue, rather than see a larger total amount with the different BAPs? Some clarity on the total amount of synuclein formed here across the different lines, as well as gene expression by qPCR (or similar) to understand where this striking effect might be occurring?

Author response:

The reviewer is correct in the observation that YFP expression was undetectable in the control worms. This is because we removed the green background of basal α -Syn expression by increasing the contrast of the images to highlight α -Syn aggregates. To avoid confusion in this regard, we have restored the original images in the new version of Fig. 4.

Figure 4. BAP-derived amyloidogenic domains promote human α -Syn aggregation in *C. elegans* model of PD A) Representative images of α -Syn muscle aggregates obtained by fluorescence microscopy in the head of *C. elegans* NL5901 fed *E. coli* VS39 expressing BAP amyloid domains. Scale bars, 10 μ m. B) Quantification of α -Syn muscle inclusions per worm (n= 20-25 worms per condition). C) Worm-thrashing representation as the number of bends per minute (BBPM) of NL5901 worms treated with BAP amyloids (n= 20-25 worms per condition). D) Representative images of α -Syn muscle aggregates obtained by fluorescence microscopy of *C. elegans* NL5901 fed pre-formed fibers of Bap_B (rBap-PFF) and Esp_N (rEspN-PFF), monomeric proteins rBapB-MON and rEspN-MON, and the nonamyloid domain rBap_CM2. Scale bars, 10 μ m. E) Quantification of α -Syn muscle inclusions per worm, (n= 15 worms per condition). Statistically significant differences were determined using Bonferroni's Multiple Comparison Test *p < 0.05, **p < 0.01, ***p < 0.001. Data are shown as means, and error bars are shown as the SE of means. F) Immunodetection of α -Syn:YFP in the protein fraction of *C. elegans* NL5901 fed with *E. coli* expressing Bap_B and Esp_N amyloids (upper panel). Arrow and * indicate α -Syn monomeric and sub-monomeric forms respectively. Anti-GFP antibody was used to detect α -Syn. Stained SDS-PAGE was used as loading control (lower panel).

Additionally, we have analysed the expression of α -Syn in worms treated with BAPs through qPCR. These results showed that BAP amyloids did not affect α -syn expression. We have included these results in the new Supplementary Fig. S6.

Panel B of figure S6. Effect of BAP amyloids on α -syn expression. qPCR measurement of α -syn mRNA level in day-10 *C. elegans* fed *E. coli* OP50 supplemented with rBapB-PFF and rEspN-PFF. Two biological replicates were performed. Statistically significant differences were determined using non parametric Kruskal Wallis Test.

B) In Fig 5A-C. Dose curves are needed, varying the amount of recombinant BAP fibrils, to identify concentration dependence and more conclusively show the potential acceleration. Do monomers of the BAPs have the same effect, or must these be in a PFF state? Controls without monomeric α Syn are also needed to see how the BAPs alone alter the ThT signal. Both these would help support interpretations of cross-seeding and begin to identify possible mechanistic activities (Are BAPs catalytic? Are they incorporated?)

Author response:

Considering the reviewer's comment, we analysed the aggregation kinetics of monomeric α -Syn (α -Syn-MON) in the presence of different amounts of rBapB-PFF seeds (1 μ M and 0.1 μ M) generated after 60 min of sonication, rBapB-PFF preformed fibrils that were not subjected to sonication (1 μ M and 0.1 μ M) and monomeric rBapB-MON (1 μ M). The results showed that the effect of rBap-PFF on α -Syn aggregation was dose-dependent and that small seeds produced by sonication of the preformed fibres were needed. The monomeric rBapB-MON protein had a weaker effect on α -Syn aggregation. We also controlled the aggregation kinetics of the seeds in the absence of the monomeric α -Syn. The results showed seeds of rBapB-PFF, rEspN-PFF and α -Syn-PFF, without monomeric α -Syn, did not alter the ThT signal. We have included these results as a new supplementary figure (Fig. S7). We have modified the text of the new version of the manuscript as follows:

Page 10, line 14:

“To investigate whether BAP-like amyloids can directly influence α -Syn aggregation in a cross-seeding process, we incubated seeds generated after sonication of rBapB-PFF or rEspN-PFF with monomeric α -Syn in vitro at concentrations lower than those at which they self-aggregated (Fig. S7A). The addition of preformed fibrils (α -syn-PFF) was used as a positive control. The results showed that Bap_B and Esp_N amyloids accelerated α -syn fibrilization in a similar way to that of α -syn-PFF (Fig. 5A-C). The effect of rBap-PFF on α -Syn aggregation was dose-dependent and small seeds produced by sonication of the preformed fibres were needed (Fig. S7B).”

Figure S7: Aggregation kinetics by measuring Th-T fluorescence over time. A) Th-T fluorescence of the seeds obtained after sonication (60 minutes) of rBapB-PFF, rEspN-PFF and α -Syn-PFF to show that the seeds alone do not alter Th-T signal. As positive control aggregation kinetic of monomeric α -Syn was shown. B) Aggregation kinetics of monomeric α -Syn in the absence or presence of the indicated amount of seeds obtained after sonication (60 minutes) of rBapB-PFF, rBapB-PFF nonsonicated or monomeric rBapB (rBapB-MON).

C) Do the BAPs co-localize with α Syn in the in vitro models? This would also help support a cross-seeding event, rather than BAP-mediated impacts on proteostasis or inflammatory pathways that could result in α Syn accumulation as well. In the discussion (pg 15, line 27-30) the authors state that they show a co localization of α Syn with BAPs, however this does not appear to be supported by any direct experimental data?

Author response:

The reviewer raises a very interesting inquiry. To determine BAP- α Syn colocalization, BAP amyloids were labelled with the protein aggregation dye ProteoStat (rBapB-PST) (Taglialegna et al., 2016). SH-SY5Y cells differentiated with retinoic acid were treated for 2 days with 10 μ g of rBapB-PST. Confocal analysis of ProteoStat labelling and antibody staining against α Syn, revealed the colocalisation of BAP with α Syn in SH-SY5Y cells. The rBapB-PST and α Syn signals appeared to overlap completely. To avoid possible complications with the rBapB-PST signal, we directly visualise the colocalisation of BAP and α Syn through immunofluorescence double staining. We again observed colocalisation of BAP and α Syn. We have included these results as Panel F of Figure 5.

Panel F of figure 5. SH-SY5Y cells were treated with preformed BAP amyloids, which were labelled (left panels) or not (right panels) with the aggregative dye ProteoStat (BapB-PST). After 10 days, cells were stained with anti- α Syn antibodies to show colocalization of BapB-PST and α Syn (left panels) or double stained with anti- α Syn and anti-Bap antibodies (right panels). Higher magnifications of the highlighted regions are shown in the lower panels.

We have modified the text of the new version of the manuscript as follows:

Page 11, line 4:

“We next tested whether BAP amyloids colocalize with α Syn. BAP amyloids labelled with the fluorescent dye ProteoStat (rBapB-PST) were used to treated SH-SY5Y cells¹⁵. By staining the α Syn with anti- α Syn antibodies, we observed the colocalization of BAP amyloids with α Syn (Fig. 5F). To avoid possible complication with the rBapB-PST signal, we directly visualize the colocalization of BAP and α Syn through immunofluorescence double staining. We also observed colocalization of BAP and α Syn, and the rBapB-PFF and α Syn signals appeared to overlap completely. (Fig. 5F).”

D) Further characterization of the in vivo outcomes would also help support possible mechanisms of action of BAPs on synuclein. These in vivo observations should be bolstered with further molecular characterization- western blots for pSyn and detergent soluble aSyn for instance, qPCR for aSyn expression, etc. All of which might support direct interactions with existing murine synuclein or effects on other cellular/molecular pathways which could more indirectly effect synuclein accumulation in mice.

Author response:

To further elucidate the mechanisms of action of BAPs on α -Syn, we analysed the α -Syn mRNA and protein expression levels in the midbrains of mice treated with BAP amyloids. Although the α -Syn RNA expression level was not affected, the α -Syn protein expression level was significantly greater (78%; $P \leq 0.05$) in the midbrains of mice treated with BAP amyloids than in those of control mice (Fig. 6I-K). In human-derived dopaminergic cells, α -Syn is predominantly degraded in lysosomes by chaperone-mediated autophagy (CMA). We measured the mRNA expression levels of 2 key proteins in the CMA pathway, LAMP-2A and Hsc70, in the midbrain. There was a significant decrease (55%; $P \leq 0.05$) in *lamp-2A* expression with a concomitant decrease in LAMP-2A protein expression levels (19%; $P \leq 0.05$) (Fig. 6I-K). Then, we wondered whether BAP amyloids would affect the degradation pathways of α -Syn. The mean half-life of α -Syn was analysed in the presence of BAP amyloids. We used cycloheximide to inhibit protein synthesis, and the cell samples were removed at 12-h intervals for up to 48 h. Under normal conditions, the mean half-life of α -Syn was 36.6 h (SD ± 1.1 h), which was not affected by the addition of Bap-MON (40.3 ± 2 h) (Fig. 6L and 6M). However, Bap-PFF significantly increased the mean half-life of α -Syn (57.8 ± 12 h), suggesting that α -Syn turnover was partially inhibited in the cells treated with BAP amyloids (Fig. 6L and 6M). The decrease in α -Syn turnover was associated with a decrease in LAMP-2A expression levels (21%) and a significant increase in α -Syn levels (103%; $P \leq 0.05$) (Fig. 6N and 6O). These results showed that CMA activity is compromised by the amyloid rBapB-PFF in human-derived dopaminergic cells and mouse models. These new results have been included as part of the new Fig. 6. We have discussed the results in the discussion section of the article.

Figure 6. Intracerebral inoculation of BAP-like amyloids enhances α -Syn pathophysiology in C57BL6/C3H F1 mice. A) Motor function of mice was assessed once a month during 7 months by quantifying time to turn around and move up toward the top of the platform in the negative geotaxis test, wirehang time, riding time (rotarod). Blue bars correspond to mice intrastrially injected with rBapB-PFF amyloids. Points represent individuals. Bars represent the means and standard errors. Data are analyzed by two-tailed t-test. * $p < 0.05$, ** $p < 0.01$. Motor data are compiled from two independent cohorts. B) Striatal TH innervation was quantified by optical density. C) Stereological quantification of TH-immunoreactive neurons in the SNpc of mice treated with rBapB-PFF and PBS. D) Striatal NeuN was quantified by optical density. E) Images for TH staining in posterior striatum. F) TH staining of dopaminergic neurons in coronal sections of the SNpc of mice treated with rBapB-PFF and PBS. G) Representative images of intraneuronal phospho- α -Syn inclusions in the ipsilateral SNpc of rBapB-PFF injected mice. Scale bars represent 20 μm . H) Quantification of Ser129 phospho- α -Syn (P-Syn) densitometry in the contralateral (CL) and ipsilateral (IL) of midbrain. I) qPCR measurement of mRNA level of α -syn, *hsc70* and *lamp2A* genes from midbrain of mice treated with rBapB-PFF. J) Representative images of the western-blot analyzing α -Syn and LAMP-2A. K) Quantification of total α -Syn and LAMP-2A protein levels normalized to beta-actin in the midbrain mice treated with rBapB-PFF. Bars represent the standard deviations of the results of 5 experiments. L) Average of α -Syn half-life in SH-SY5Y cells incubated with rBapB-PFF and rBap-MON. Cells were treated with 50 $\mu\text{g/ml}$ cycloheximide to inhibit protein synthesis. Disappearance of α -Syn was quantified as the mean of three experiments. M) Representative images of western-blot analyzing α -Syn turnover during 48 hours. N) Representative images of the western-blot analyzing α -Syn and LAMP-2A expressed in cells incubated with rBapB-PFF and rBap-MON. O) Quantification of total α -Syn and LAMP-2A protein levels normalized to beta-actin in the SH-SY5Y cells treated with rBapB-PFF and rBap-MON. Bars represent the standard deviations of the results of 3 independent experiments. Data are shown as means, and error bars are shown as the SD of means. Statistically significant differences were determined using Mann–Whitney test. * $p < 0.05$, ** $p < 0.01$. Red bars correspond to mice intrastrially injected with rBapB-PFF amyloids. White bars correspond to mice intrastrially injected with PBS.

As an example, the endpoint observation of equivalent area of GFAP and Iba helps to support a claim that neuroinflammatory activation is not involved, however, earlier timepoints would better show this. BAPs may incite a strong, yet acute, inflammatory activation that triggers the subsequent degeneration and pathology, before being resolved by the endpoint where this Iba/GFAP equivalency is observed. Controls with non-amyloidogenic BAP would help to confirm this too.

Author response:

As suggested by the reviewer, we analysed microglial activation using Iba-1 staining at early time points. The results showed no marked differences in the microglial activation in the substantia nigra pars compacta (SNc) between mice after rBapB-PFF injection and those after PBS or rBap-MON injection. The lack of a neuroinflammatory response in the SNc 72 hours after the interstitial injection of rBapB monomer or rBapB-PFF and 7 months after the injection of rBapB-PFF excludes the possibility that Bap-mediated neurodegeneration was mainly associated with a neuroinflammatory response to Bap fibrils. We have included these results as part of Figure S11.

A

Cerebral structure	Injection site	Striatum	
	Time	24h	72h
Motor cortex			
Visual cortex			
Somatosensory cortex		++	+++
Hippocampus			
Entorrhinal cortex			++
Olfactory areas			
Striatum		+++	+++
Amygdala		+++	++
Thalamus		+	+
Hypothalamus		++	++
SNpc			

B

C

D

Figure S11. Analysis of the presence of rBap-PFF at different brain regions. A) Summary table of the presence of rBap-PFF on different brain structures. C57BL6/C3H F1 mice (n=10) received an injection of sonicated rBap-PFF (10 μ g/mice) into the dorsal striatum by stereotaxic injection. Mice were sacrificed at 24 h or 72 h post injection and the presence of Bap aggregates were determined by immunostaining using anti-Bap antibodies. B) rBap-PFF immunostaining in different brain sections. C) Representative images showing immunostaining of IBA-1 microglia/macrophages on mid-brain sections of mice treated with rBap-PFF and rBap-MON. TH-positive neurons are shown in red. D) Graphs showing quantification of the fluorescence intensity corresponding to Iba-1 (n=2). Error bars are shown as the SD of means.

Analysis of cell culture models for inflammatory pathway activation or cytokine release would also help to clarify a role for inflammatory activation.

Author response:

To investigate the role of BAP amyloids in inflammatory activation, we quantified the release of proinflammatory cytokines, including tumour necrosis factor alpha (TNF- α), interleukin-6 (IL-6) and interleukin-8 (IL-8) in the culture media of SH-SY5Y cells treated with preformed fibrils (BapB-PFF and EspN-PFF), monomeric proteins (BapB-MON and EspN-MON) and the nonamyloid domain Bap_CM2. As a positive control, cells were treated with LPS from *E. coli*. After 24-h of stimulation, the production of the cytokines TNF- α , IL-6 and IL-8 was not affected in SH-SY5Y cells treated with BAP amyloids. These results suggest that BAP amyloids do not induce an inflammatory pathway activation that triggers α -Syn aggregation. We also determined the cytotoxic effects of rBap-PFF and rEspN-PFF on SH-SY5Y cells. After 48 h and 10 days of the treatment with BAP amyloids, a lactose dehydrogenase (LDH) assay was performed to evaluate cell death. The results showed that BAP amyloids did not affect cell viability. We have included these results as Supplementary Fig. S8.

Figure S8: Effect of BAP-amyloids on cell viability and cytokine release. A) Differentiated SH-SY5Y cells were treated for 2 days with 0.02 μ g/ml of rBapB-PFF and rEspN-PFF. LDH assay was used to determine the cell viability (% cell death in comparison with the positive control). Data are expressed as mean \pm SD (n=3). B) Differentiated SH-SY5Y cells were treated for 2 days with 0.02 μ g/ml of BapB-PFF and EspN-PFF, monomeric proteins (BapB-MON and EspN-MON), the nonamyloid domain Bap_CM2 and LPS (0.05 μ g/ml). The concentration of IL-8, IL-6 and TNF- α cytokines in the supernatant was measured by ELISA. Values are expressed in pg/ml and represent the mean \pm SD of triplicate analyses.

Presence of BAPs in human microbiomes

A) The mining of the existing, large scale metagenomic study of PD is very useful. I am slightly confused regarding the analysis however. As an example, the authors show in Fig 7 that the BAP encoded by *Lactobacillus johnsonii* is present in 83 PD vs 17 Control samples. However, the study these data come from states that *L. johnsonii* is present in only 11 PD vs 4 Control samples (Supplementary Data Table 1 of the published, cited study). Similar discrepancies appear with *E. faecalis* (44 vs 7 here and 42 vs 11 in the cited study), and likely others. Clarity is needed in the differences in the analysis that would detect, for instance, so much more *L. johnsonii* in the same dataset (8x more here). In the case of LJ, are the BAP_LJ gene coming from other similar species and being combined? The reason for prevalence differences are too unclear.

Author response:

The reviewer is correct in their observations, and we regret the confusion created in this regard. In a metagenomic study of PD (Wallen et al. 2023), the whole-genome shotgun read sequences were profiled using MetaPhlan3 with an accompanying database of ~1.1 M unique clade-specific marker genes. Although MetaPhlan has proven to be an excellent tool for identifying and quantifying bacterial presence, it does not consider genes in the accessory genome. In our analyses, we assumed that BAP-like genes are on the accessory genome and therefore could not be assessed via regular whole-genome interrogation or a 16S metagenomics approach. BAP-like genes could be horizontally transferred between bacteria, that is, not to their offspring or the same bacterial species. Therefore, we decided not to focus on the bacterial species rather than on the presence of BAP-encoding genes. The raw data from the Wallen et al. study were realigned against a custom database including conserved regions of BAP-like genes from representative bacterial species often present in the gut microbiota. We established a robust identification method in which the read alignment, based on the BWA algorithm, must uniquely identify the conserved region of BAP-like genes. As a result, the output data reflect the BAP-like gene frequencies corresponding to a homologous BAP gene of a certain species. However, this result cannot be considered the frequency of such bacterial species because a homologous BAP gene could be present in additional closely related species because of horizontal gene transfer. This might explain the discrepancies noted by the reviewer. To avoid any confusion in this regard, we refer to the presence of the BAP genes encoding Bap_FM, Esp, Bap_EFm, Bap_ECo, Bap_EFg and Bap_LJ, and not the presence of the bacteria.

B) Perhaps showing the bacterial species specifically and the predicted encoded BAPs side by side would help to clarify the differences between the presence of the organism and the genes of interest. I do understand that orthologs may be encoded by species that are not fully annotated and that strains of the same species may not have the specific gene. In those cases, the authors should be careful in their discussion of bacterial species vs the BAP itself (such as on pg 12 lines 7-11, re: Fig 7).

Author response:

We propose that a specific acquired gene encoding BAP amyloids could be present in the gastrointestinal microbiome and that this presence, in the long term, could be statistically significant concerning PD predisposition. As mentioned in the previous response, since the BAP genes are located in the accessory genomes and, in addition, the same homologous BAP gene could be encoded in different closely related species, we considered that listing the bacterial species and the *bap* genes side by side would not be useful. We have included changes in the manuscript for the reader to understand that we are only studying the presence of BAP-like genes and not the bacteria itself.

C) A discussion around the implications of the prevalence of BAPs between PD and control, vs the limited differences in actual abundance of those BAPs in those who are carrying them, irrespective of their PD status. Are the presence of BAP a potential trigger or a consequence?

Author response:

In the reanalysis of the large-scale metagenomic study of PD, the prevalence and abundance of BAP-encoding genes were investigated in a cohort of 490 PD patients and 234 neurologically healthy controls. Our results revealed an increase in the prevalence of certain genes encoding BAP amyloids in patients with PD. In addition, the abundance of *bap* genes that were most significantly represented in the PD group was calculated as the log₂ of RPKM. To identify statistically significant differences in RPKM, we used MaAsLin2 test and adjusted by the Benjamini-Hochberg false discovery rate (FDR) method. We found that the abundance was significantly greater in the PD group than in the control group. To highlight these results, we have indicated the significant differences in the violin graphs from Fig. 7. On the other hand, the reviewer points out an interesting observation, whether the of BAP genes underlie the pathology

or, on the contrary, they are a result of the illness. In our opinion, the presence of *bap* genes can be considered as a risk factor triggering PD pathology. However, the fact that BAP amyloids were detected in the microbiome of the control group suggests that external factors will be necessary to develop the disease. One possibility is that pathogenic amyloids could need to reach a threshold concentration in the intestine to influence α -Syn. The threshold concentration of BAP amyloids may also have collateral effects on the intestine, such as triggering a perturbation of the gut barrier, modulating the physiology of the consortium of gut bacteria by inducing changes in the bacterial composition or promoting interspecies amyloid aggregation, all of which may encourage an environment conducive to disease development. Therefore, not only the presence of amyloids but also the abundance of amyloids could be relevant causes of the pathology. We have discussed this possibility in the discussion section of the new version of the article.

D) There are also some discrepancies between the sequencing and the blotting in Fig 1 vs Fig 3. For instance, MB23984 has no reactivity for any of the tested amyloids (including EspN and BapLA), yet it was one of the higher samples for gene content with EspN. This is true for other samples as well – MB24377 for instance is also high in EspN and BapLA by gene content, but shows no signal in the detection assay. How well do the sequencing data align with the protein measurements? It suggests that encoding does not correspond to functional protein here, or that perhaps the amyloid staining in Fig 3 is not reacting to the bacterial amyloids.

Author response:

The reviewer is completely correct. Sequencing data provide information about the genetic load in the sample. However, the presence of a gene does not guarantee protein expression. Even if the protein is expressed, it does not imply the formation of amyloid structures. It is important to note that BAPs are facultative amyloids. They are secreted in a functional globular folded state and require certain conditions to change their conformation to an amyloid fold. Therefore, neither the presence of the gene nor the presence of the protein guarantees the formation of amyloid structures. The protocol used in this study is based on the purification of the insoluble fraction in which only amyloid structures can be detected. We ruled out the possibility that the amyloid stain in Fig. 3 did not react to the bacterial amyloids for the following reasons: i) The material purified and detected with the antibodies (Fig. 3) is susceptible to formic acid treatment (Fig. S5B). ii) Only the SDS-resistant fraction that were purified from faecal samples of mice colonized with EspN-PFF amyloids reacted with anti-Esp antibodies (Fig S5C). iii) The conformational amyloid antibody A11 recognise intermediate amyloid folding stages of Esp (Fig S5D).

E) It is fairly well established that injection of protein aggregates of various types into the brain, or other anatomical sites, can result in amyloid pathology. Given these are proteins produced in the gut, what is the biological relevance of injecting them into the brain?

Author response:

The gut–brain axis provides a bidirectional way of communicating with the central nervous system (CNS) through several mechanisms including neuroendocrine cells, the activation of immune cells and the enteric nervous system, through which the vagus nerve acts as the most important neural communication pathway. During homeostasis, bacteria may stimulate the afferent neurons of the enteric nervous system, and the resulting vagal signals originating from the gut induce anti-inflammatory responses, preventing infectious and inflammatory states that might otherwise be caused by pathological microorganisms. However, certain factors, such as age and/or altered gut microbiota, may disrupt gut barrier function, commonly known as “leaky gut”, which presumably allows the translocation of microbiota-derived products that may interact with the enteric neurons and, more excitingly, reach the brain themselves. Transmission from the gut to brain has been described for prions in transmissible spongiform encephalopathies. Upon oral ingestion, prions can survive the process of digestion and become incorporated into the gut. In the intestinal epithelium, prions interact with immune cells and accumulate in follicular dendritic cells and other lymphoid follicles. Then, the prions might move to the enteric nervous system, where they finally spread to the CNS. Similar spreading mechanisms may be responsible for the

propagation of other amyloid proteins that cause neurodegenerative disorders. Misfolded proteins can also move between cells and seed the aggregates of their normal conformers. Assuming that amyloids can migrate from the gut to the brain through the gut-brain axis, in this study, we used direct intracerebral inoculation of BAP preformed amyloids (rBapB_PFF) in wild-type nontransgenic mice. However, we agree with the reviewer that it would be interesting to investigate whether long-term gut exposure to pathological amyloids would affect α -Syn pathology in the intestine and the brain. We have modified the text of the new version of the manuscript to clarify the relevance of injecting BAP amyloid into the brain.

Page 18, line 25:

“Assuming that amyloids can migrate from the gut to the brain through the gut-brain axis, in this study we forced this scenario using direct intracerebral inoculation of BAP preformed amyloids (rBapB_PFF) in wild-type nontransgenic mice.”

Specificity for the amyloidogenic properties of the BAPs

A) In many instances, BAPs are compared to PBS buffer controls in all of the model systems, from the *in vitro* aggregation assays to the mouse intracerebral injections. Rather than PBS alone, perhaps the use of a BAP that does not form amyloid as readily (like CM2?) could be a useful control for amyloidogenic capacity and specificity or a mutant BAP that does not form amyloid.

Author response:

The reviewer suggests a very interesting control. We included the nonamyloid domain Bap_CM2 as a negative control in the of α -Syn aggregation experiments using the *C. elegans* and SH-SY5Y models. As shown in the new versions of Figures 4 and 5, Bap_CM2 did not affect α -Syn aggregation.

B) Similarly, purified, monomeric BAP (rather than PFF) would be essential in the cell culture and mouse systems, building on what is understood in the aSyn PFF models. Do monomers never induce pathology? Or do they simply take longer?

Author response:

As mentioned above, we included new controls for α -Syn aggregation, including the nonamyloid domain Bap_CM2, and the monomeric proteins rBapB-MON and rEspN-MON. As shown in the new Figures 4 and 5, rBapB-MON and rEspN-MON did not affect α -Syn aggregation in either the *C. elegans* or SH-SY5Y models. Based on these results, we also expected that BAP monomers would not affect α -Syn aggregation *in vivo*. However, we cannot rule out that monomeric proteins might aggregate *in vivo* over time, causing synucleopathologies in the future. Although this is an interesting topic, we believe that it is beyond the scope of this paper and will be the subject of future research. This possibility has been included in the Discussion of the new version of the article.

Page 19, line 3

“Although this effect has been observed using preformed BAP fibres, it cannot be ruled out that monomeric proteins may aggregate in vivo over time causing synucleopathologies in the future.”

Minor (generally in order of appearance)

In Fig 2B there are no indications of which colonies 1-26 are the controls and the tested proteins. I think it corresponds to the list presented in Supplemental Fig S2, but this was difficult to interpret as a reader.

Author response:

As recommended by the reviewer we have included the list of the *E. coli* strains that export BAP domains in the main figure 2B. We have indicated the number corresponding to the positive and negative controls in the figure legend.

Fig S2 shows significance compared to the blank, but this seems a bit unfair. Wouldn't the better comparison be to the vector/expression controls of BapA and EspRC?

Author response:

There is a misunderstanding on the part of the reviewer. *E. coli* strains expressing BAP-derived domains were compared with *E. coli* without the pExport plasmid (Ø) and not with the blank (no bacteria). This point has been mentioned in the Figure legends to avoid any confusion.

The relevance of anti-His immunogold reactivity is unclear to me Fig S3. Is this region histidine rich? Perhaps a phrase to explain to non-experts the relevance of this observation in the results.

Author response:

We thank the reviewer for this appreciation. To explore the amyloid-forming propensity of BAP domains, we used the C-DAG system. The plasmid used in this system (pExport) allows the expression of the epitope of interest fused to the first 42 residues of the signal sequence of CsgA (ssCsgA) and the 6xHistidine tag. Therefore, amyloid fibres can be detected using anti-His antibodies. We have explained this point in the Figure legend of the new Fig. S2.

Fig 3- controls are needed for specificity. Stool samples from a microbe free environment (germ-free mice?) for instance? To ensure that the antibodies indeed do not react to epitopes derived from host tissues. Cross-reactivity between amyloid-detecting antibodies can occur. Given the reactivity comes almost completely from individuals with IBS, could this be a reaction to something from the host rather than the microbes? Synuclein? Also a technical question, as I am not familiar with these specific types of blots. But in western blotting, the lighter color can sometimes indicate too strong reactivity that then blocks the detection substrate. How is equal loading confirmed? Could samples be run with and without a formic acid treatment to ensure some specificity of the amyloid signal?

Author response:

To ensure that the antibodies do not react to epitopes derived from host tissues, the reviewer proposes the use of stool samples from germ-free mice. Considering the difficulties of working with germ-free mice, we used a broad-spectrum antibiotic treatment to deplete the gut microbiota of the mice. We purified the insoluble protein fraction of the microbiota of the treated mice. As a positive control, we used stool samples from mice that were colonised with EspN-PFF after oral administration. The presence of SDS-stable amyloids was determined using a filter retention assay. Immunological detection using antibodies against Esp_N did not detect amyloid-like species in the stool samples of mice with a depleted microbiota, and the strains only reacted to samples from mice that were orally administered EspN-PFF amyloids. This result suggested that the antibodies react with Esp-derived amyloids and not with epitopes derived from host tissues (at least from mice). These results have been included in the new section of Figure S5 (Fig. S5C).

To confirm the specificity of the amyloids purified from human stool samples, SDS-resistant amyloids were treated with formic acid (100%) or a solution of trifluoroacetic acid/hexafluoroisopropanol (1:1). The treated samples were subjected to vacuum filtration through a 96-well dot blot apparatus with an acetate cellulose membrane. Immunoblotting using anti-EspN antibodies revealed no retained aggregates in faecal samples treated with formic acid or a solution of trifluoroacetic acid/hexafluoroisopropanol. These results reinforce the amyloid nature of the BAP aggregates purified from human samples. We have included this result in the new version of the Supplementary material (Fig. S5B).

Prompted by the reviewer's comment, we also conducted an assay to analyse the binding of the A11 antibody to Esp amyloids. We purified the insoluble fraction from a culture of VS39 pExport-Esp, using the protocol of Dueholm, M. S., et al. The results showed that the A11 antibody can recognise intermediate amyloid folding stages of Esp, which were sensitive to formic acid, while the OC antibody slightly detected Esp mature fibres purified from the insoluble fraction (Fig. S5D and S5E). These results suggest that the signal detected by A11 antibody in the insoluble fraction

of the human fecal samples could correspond to intermediate amyloid oligomers of Esp. We have included these results as part of the new version of Figure S5.

Finally, the reviewer asked how equal loading was confirmed. We did not use a loading control, instead, all the fecal samples were normalised to 32.4 mg/ml of wet weight in PBS and 10 ml.

Figure S5. BAP amyloid-like structures are present in the human fecal microbiota. A) Quantification of BAP-related amyloids from microbiota. The relative signal intensity of the blots was determined by densitometry using Image Lab Software. GraphPad Prism Version 6 was used for density analysis. B) Insoluble protein fractions purified from human stool samples were treated with 100% formic acid (FA) or a solution of trifluoroacetic acid/hexafluoroisopropanol (TFA:HFIP) (1:1). After treatment, samples were subjected to vacuum filtration through a 96-well dot blot apparatus with an acetate cellulose membrane. Immunoblotting using anti Esp_N antibodies showed no retained aggregates. C) Detection of Esp amyloids in stool samples of mice subjected to a broad-spectrum antibiotics treatment to deplete the gut microbiota (Ø). After antibiotics treatment, rEspN_PFF were orally administered for 2 days. The SDS resistant fraction purified from mice faecal samples were subjected to a filter retention assay. Immunoblotting was performed using anti Esp_N antibodies. D) Detection of CsgA and Esp amyloids purified from the insoluble fraction of *E. coli* VS39 pExport-CsgA (IF_CsgA) and *E. coli* VS39 pExport-EspN (IF_EspN) using the anti-amyloid oligomers (A11) and the anti-amyloid fibrils OC antibodies. Samples were treated (+FA) or not (-FA) with formic acid. E) Immunodetection of CsgA and Esp in the purified insoluble fraction using anti-His and anti-Esp antibodies, respectively. Mature fibres of CsgA and amyloid oligomers of Esp were marked with arrows. Monomers of CsgA and Esp were marked with asterisks.

PBS controls in Fig 5C do not appear to have the same N displayed as in 5B (2 vs 3).

Author response:

The reviewer is correct. We have modified Figure 5 accordingly.

Fig 5H, whole blots are needed. Are oligomeric forms of aSyn observed in these fractions? Was pS129 probed from the same blot as the total aSyn? Are the observed bands forming at the same MW between the two targets? This would also help support possible

mechanistic inferences.

Author response:

We thank the reviewer for this suggestion. Whole SDS-page and western-blot are now included as supplementary Figure S10. High molecular weight α -Syn have been shown in the whole blots. PS129 and total α -Syn were probed using different blots.

The quantification of pSyn density in Fig6H is not clear. Given the sparsity of aggregates seen in Fig 6G, counts should be provided- are these in TH+ cells?

Author response:

In most animals the aggregates are really small and difficult to identify subsequently, the aggregate count would be inaccurate. For this reason, we select a more accurate approach analyzing the pSyn density as previously reported (Gabrielyan L, Liang H, Minalyan A, Hatami A, John V, Wang L. Behavioral Deficits and Brain α -Synuclein and Phosphorylated Serine-129 α -Synuclein in Male and Female Mice Overexpressing Human α -Synuclein. J Alzheimers Dis. 2021;79(2):875-893.)

Pg 13, lines 16, 20 (and possibly elsewhere)- Gram should be capitalized.

Author response:

We modified the text accordingly.

Citation 77 does not support a prion like spread of BAPs or any protein (discussion, pg 16 line 11)

Author response:

The reviewer is correct. We have removed the reference from the text.

Reviewer #3. Bacterial amyloids / Structural and functional analyses. (Remarks to the Author):

Fernández-Calvet and coworkers report a comprehensive study the prevalence of Biofilm associated protein (Bap) – like bacterial amyloid sequences in the human intestinal microbiota, and evaluate their potential neurodegenerative activity.

Through a phylogenomic mining of sequenced gut microbiomes the authors identify a series of Bap-like proteins and map associated candidate amyloidogenic degradation fragments. They then analyse their prevalence in the metagenomes of 35 individuals with irritable bowel syndrome and 14 control individuals, finding a high prevalence for some of the identified proteins. A selection of the identified protein fragments are cloned into a curli-based export system to monitor production of extracellular amyloid-like fibers, and cloned for recombinant expression. ThT, CR, EM and FTIR are used to monitor the presence and extend of amyloid formation on cells or in vitro using purified recombinant protein fragments.

The authors demonstrate that some of the identified Bap fragments are amyloidogenic and that stools of patients have variable levels of A11-reactive fibers. They then go on to evaluate the cross-seeding activity towards alpha-Synuclein in vitro, as well as in cellulo and in mice. Doing so, the authors identify the relatively widespread presence of amyloidogenic proteins in the human gut microbiome, and demonstrate their principle potential to facilitate the formation of amyloidogenic deposits of alpha-Syn.

The setup and findings of the study are novel and will be of interest to the growing community seeking to determine the potential of gut microbiota and their proteomes to enhance human protein aggregation disorders and associated pathologies. The study is generally well executed and documented. A few points need further clarification prior to publication.

Author response:

We would like to express our gratitude for the reviewer's critical evaluation of our manuscript. We have taken on board their suggestions and believe that the revised submission has been greatly improved as a result.

Specific points of attention:

1) The alpha-Syn cross-seeding experiments use 1 μ M Bap pre-formed fibrils or 10 nM active human recombinant a-Syn-PFF. Did the authors try lower concentrations of rBap? It would be important to determine if there is a critical concentration to the Bap – a-Syn cross-seeding activity.

Author response:

We thank the reviewer for this relevant question that have been also intreated by the other reviewers. We have analyzed the aggregation kinetics of the monomeric α -Syn in the presence of 1 μ M and 0.1 μ M of rBapB-PFF seeds. We also included rBapB-PFF nonsonicated and monomeric rBapB (1 μ M). The results showed that the effect of rBap-PFF on α -sSyn aggregation was dose-dependent and that small seeds produced by sonication of the preformed fibres were needed. The monomeric rBapB-MON protein had a small effect on α -syn aggregation. We have included as a control, the aggregation kinetic of rBap-PFF, rEsp-PFF and α -Syn-PFF seeds without monomeric α -Syn. The results showed that seeds of rBapB-PFF, rEspN-PFF and α -Syn-PFF did not alter Th-T signal. We have included these results as Supplementary Figure S7. We have modified the text of the new version of the manuscript as follows:

Page 10, line 14

“To investigate whether BAP-like amyloids can directly influence α -syn aggregation in a cross-seeding process, we incubated seeds generated after sonication of rBapB-PFF or rEspN-PFF with monomeric α -syn in vitro at concentrations lower than those at which they self-aggregated

(Fig. S7A). The addition of preformed fibrils (α -syn-PFF) was used as a positive control. The results showed that Bap_B and Esp_N amyloids accelerated α -syn fibrilization in a similar way to that of α -syn-PFF (Fig. 5A-C). The effect of rBap-PFF on α -syn aggregation was dose-dependent and small seeds produced by sonication of the preformed fibres were needed (Fig. S7B).”

Figure S7. Aggregation kinetics by measuring Th-T fluorescence over time. A) Th-T fluorescence of the seeds obtained after sonication (60 minutes) of rBapB-PFF, rEspN-PFF and α -Syn-PFF to show that the seeds alone do not alter Th-T signal. As positive control aggregation kinetic of monomeric α -Syn was shown. B) Aggregation kinetics of monomeric α -Syn in the absence or presence of the indicated amount of seeds obtained after sonication (60 minutes) of rBapB-PFF, rBapB-PFF nonsonicated or monomeric rBapB (rBapB-MON).

2) Furthermore, in the cellular or in vivo experiments, cells were exposed to 10 µg rBap-PFF / well (please provide a concentration) or mice were injected with 10 µg rBap-PFF / mouse, resulting in a quite high local concentration. How does this relate to expected circulating concentrations of rBap fragments? Did the authors try lower doses? It is advisable to include a discussion on the concentrations used and how they relate to expected in vivo conditions, and whether there is any critical minimal concentration to induce disease.

Author response:

The reviewer makes an interesting observation regarding the concentrations of BAP used and how this relates to the expected circulating concentration of rBap fragments. Although this is a relevant point, we consider it extremely difficult to correlate the concentration used in the experiments, with the expected *in vivo* conditions. We selected the rBap-PFF concentration based on previous studies that used α -Syn preformed fibres in similar *in vivo* and cellular models (Luk, et al 2012; Paumier et al., 2012). For the *in vivo* experiment, we performed a preliminary assay using different concentrations of rBapB-PFF administered at different brain locations. In the experiment, 2, 5, and 10 µg of BapB-PFF was injected into the olfactory bulb, intracortex, or intrastriatal region, respectively, of C57BL6/C3H F1 mice. Mice were sacrificed at 24 h and 72 h postinjection, and the presence of rBapB-PFF in different brain sections was determined by immunostaining. The results showed that the rBapB-PFF aggregates were stable at 72 h postinjection, and were localized at regions other than the injection site (see Response Figure R3). Since previous studies have demonstrated that intrastriatal injections of fibrillar α -Syn into mice

induced Parkinson's disease (PD)-like Lewy body (LB) pathology, we used the same model to determine whether BAP amyloids induce α -Syn pathology. We have included the results corresponding to the intrastriatal injection of rBap-PFF in Supplementary Figure S11.

A

Injection site	Cortex		Olfactory bulb		Striatum	
Concentration BAP	5 μ g/mice		2 μ g/mice		10 μ g/mice	
Time	24 h	72 h	24 h	72 h	24 h	72 h
Cerebral structure						
Motor cortex				++		
Visual cortex		++				
Somatosensory cortex	+		+	++	++	+++
Hippocampus	+++	+++				
Entorrhinal cortex	++	++				++
Olfactory areas			+++			
Striatum			+	++	+++	+++
Amygdala					+++	++
Thalamus		+	+		+	+
Hypothalamus			+		++	++
SNpc	+	+	+	+		

B

Response Figure R3. Analysis of the presence of rBap-PFF at different brain regions. A) Summary table of the presence of rBap-PFF aggregates. Three groups of mice (n=10) received an injection of sonicated rBap-PFF into the cortex (5 μ g/mice), dorsal striatum (10 μ g/mice) and olfactory bulb (2.5 μ g/mice). Mice were sacrificed at 24h or 72h post injection and the presence of Bap aggregates were determined by immunostaining using anti-Bap antibodies. B) rBap-PFF immunostaining in different sections of brains of mice sacrificed at 72h following rBap-PFF injection.

As recommended by the reviewer, we indicate the concentrations of rBap-PFF used in the cellular experiments: 0.02 μ g/ μ l, and in *in vivo*: 10 μ g/mouse.

We have also included a discussion on the concentrations used and how they relate to expected *in vivo* conditions.

Page 17 line 5:

“The fact that BAP amyloids were detected in the microbiome of the control group suggests that pathogenic amyloids may need to reach a threshold concentration in the intestine to have an effect on α -Syn. This hypothesis was supported by the fact that cross-seeding of α -Syn is concentration dependent (Fig. S7). The threshold concentration of BAP amyloids could also have collateral effects on the intestine, such as triggering a perturbation of the gut barrier,

modulating the physiology of the consortium of gut bacteria by inducing changes in the bacterial composition or promoting interspecies amyloid aggregation, all of which may encourage an environment conducive to disease development.”

3) Control animals were injected with PBS. Can the authors exclude an inflammatory activity of endotoxin co-purified with their recombinant Baps? The authors should include an injection of a recombinant non fiber forming Bap fragment as a control.

Author response:

We thank the Reviewer for their very valid comments regarding the possibility that the copurification of endotoxins with recombinant BAP proteins may have inflammatory activity. To address this point, we have added several new experiments to the manuscript. To determine the presence of endotoxin copurified with recombinant BAP proteins we determined the cytotoxic effects of rBap-PFF and rEspN-PFF on SH-SY5Y cells. We treated differentiated SH-SY5Y cells with 0.02 $\mu\text{g}/\mu\text{l}$ of rBap-PFF and rEspN-PFF. After 48 h and 10 days of the treatment a lactose dehydrogenase (LDH) assay was performed to evaluate cell death. The results showed that none of the BAP amyloids altered cell viability. We have included this result in the new Supplementary Figure S8A. We also investigated the production of the proinflammatory cytokines, tumour necrosis factor alpha (TNF- α), interleukin-6 (IL-6) and interleukin-8 (IL-8) in the culture media of SH-SY5Y cells treated with preformed fibrils (BapB-PFF and EspN-PFF), monomeric proteins (BapB-MON and EspN-MON) and the nonamyloid domain Bap_CM2. As a positive control, cells were treated with LPS from *E. coli*. After 24 h of stimulation, the production of the cytokines TNF- α , IL-6 and IL-8 was not affected in SH-SY5Y cells treated with BAP amyloids. These results suggest that BAP amyloids do not active an inflammatory pathway that triggers α -Syn aggregation. We have included these results as part of the Supplementary Fig. S8B.

Figure S8: Effect of BAP-amyloids on cell viability and cytokine release. A) Differentiated SH-SY5Y cells were treated for 2 days with 0.02 $\mu\text{g}/\text{ml}$ of rBapB-PFF and rEspN-PFF. LDH assay was used to determine cell viability (% cell death in comparison with the positive control). Data are expressed as mean \pm SD (n=3). B) Differentiated SH-SY5Y cells were treated for 2 days with 0.02 $\mu\text{g}/\text{ml}$ of preformed fibrils (BapB-PFF and EspN-PFF), monomeric proteins (BapB-MON and EspN-MON), the nonamyloid domain Bap_CM2 and LPS (0.05 $\mu\text{g}/\text{ml}$). The concentration of IL-8, IL-6 and TNF- α cytokines in the supernatant was measured by ELISA. Values are expressed in pg/ml and represent the mean \pm SD of triplicate analyses.

Finally, we analysed microglial activation using Iba-1 staining at early time points. The results showed no marked differences in the microglial activation in the brains of mice after rBap-PFF injection compared with those after PBS or rBap-MON injection. These results, suggested that Bap-mediated neurodegeneration was not associated with a neuroinflammatory response to Bap fibrils. We have included these results as part of Figure S11.

Figure S11. Analysis of the presence of rBap-PFF at different brain regions. A) Summary table of the presence of rBap-PFF on different brain structures. C57BL/6/C3H F1 mice (n=10) received an injection of sonicated rBap-PFF (10 μ g/mice) into the dorsal striatum by stereotaxic injection. Mice were sacrificed at 24 h or 72 h post injection and the presence of Bap aggregates were determined by immunostaining using anti-Bap antibodies. B) rBap-PFF immunostaining in different brain sections. C) Representative images showing immunostaining of IBA-1 microglia/macrophages on mid-brain sections of mice treated with rBap-PFF and rBap-MON. TH-positive neurons are shown in red. D) Graphs showing quantification of the fluorescence intensity occupied by positive Iba-1 (n=2). Error bars are shown as the SD of means.

Additional points of attention:

Page 3 - Ln 24-26: The term ‘facultative’ bacterial amyloid was coined by Van Gerven et al. 2018 J. Mol. Biol. 430, 3657–3684.

Author response:

We totally agree with the reviewer. We have included the reference.

Page 3 - Ln32: gram-positive and gram-negative should be Gram-positive and Gram-negative. Check throughout text.

Author response:

We modified the text accordingly.

Page 4 – Ln 3: The authors write: “Amyloid protein folding patterns are highly conserved in human and bacterial amyloids.” What is this statement based on? More caution should be made in this statement. To date, the only structures available are for the obligate bacterial amyloid curli (Sleutel et al. 2023, Nat Commun 17;14(1):2822), which forms beta-solenoids rather than the serpentine amyloid folds seen in pathological amyloids. This may be different in facultative bacterial amyloids, where the amyloid state is an off-pathway beta aggregation of often protein fragments, and does not form the native fold of the amyloidogenic proteins. There are however no structural studies to date to confirm or disconfirm this claim. This statement on structural similarity is repeated in the Discussion, Page 15, Ln10-11. Statements on cross-seeding ability are correct, but any claims of structural similarity should be toned down and be rephrased.

Author response:

We have rewritten this statement in the Introduction and Discussion sections. Thank you for the suggestion. Now it states:

Page 4 line 1:

“Recent advances in determining amyloid structure have revealed a high diversity of fibre architectures²². However, many features are shared by most amyloid structures, which opens the possibility of a promiscuous interspecies of amyloids.”

Page 17, line 14:

“Although recent studies have shown the existence of a wide variety of amyloid fibre architectures²², bacterial functional amyloids can be considered as canonical molecular mimics of human molecules that can play a pathological role in the central nervous system (CNS) in certain situations, either by nucleating the polymerisation of human amyloid proteins (cross-seeding) or priming an inflammatory response^{47,49,72}.”

Ref. 21 needs formatting: “Rogers, G. B. et al. From gut dysbiosis to altered brain function and mental illness: mechanisms and pathways. 21, 738–748 (2016).”

Author response:

The reference has been properly formatted
Rogers, G. B., Keating, D. J., Young, R. L., Wong, M.-L., Licinio, J., & Wesselingh, S. (2016). From gut dysbiosis to altered brain function and mental illness: Mechanisms and pathways. Molecular Psychiatry, 21(6), 738–748.

References:

Luk, K. C., Kehm, V., Carroll, J., Bin Zhang, O’Brien, P., Trojanowski, J. Q., & Lee, V. M. Y. (2012). Pathological α -Synuclein Transmission Initiates Parkinson-like Neurodegeneration in Nontransgenic Mice. *Science (New York, NY)*, 338(6109), 949–953.

Paumier, K. L., Luk, K. C., Manfredsson, F. P., Kanaan, N. M., Lipton, J. W., Collier, T. J., et al. (2015). Intrastratial injection of pre-formed mouse α -synuclein fibrils into rats triggers α -synuclein pathology and bilateral nigrostriatal degeneration. *Neurobiology of Disease*, 82, 185–199.

REVIEWERS' COMMENTS

Reviewer #1 (Remarks to the Author):

The author's have thoughtfully addressed our comments. There are a few minor comments to consider.

Minor comments to the authors:

1. Page 3, line 15 change 'csgACB' to csgBAC
2. Page 27, line 30 change "bards" to bars.
3. Figure S6A What is the band between 35- 48 kDa in the control sample?
4. In Figure S10 the blot using anti-PS a-Syn antibodies shows the presence of prominent high molecular weight (HMW) a-Syn \sim 135 kDa. However, the HMW band is absent in the blot probed with a-Syn antibodies. Is it known that phosphorylated a-Syn (P-S129) form SDS stable higher molecular weight species? Can the authors comment on that?

Reviewer #2 (Remarks to the Author):

The authors should be commended on their detailed response and, to me, the manuscript is certainly in a significantly improved state. I do think this is an interesting study that will be useful for the field.

I have two **very** minor concerns that might be worthwhile to address prior to publication.

- 1) New Figure S11, demonstrating spread of BAPs to various brain regions following intra-striatal injection. 24hr spread seems very fast based on other reports (with synuclein PFFs), the authors could show some negative control for specificity of the anti-BAP staining in these tissues.

2) In the rationale for using intra-striatal injections (rather than intestinal injections or exposures) the authors state: "Assuming that amyloids can migrate from the gut to the brain through the gut-brain axis, [...]"

I would suggest revising the term "gut-brain axis" here. This general refers to a conceptual axis of vagal, spinal, hormonal, immune, and metabolic signals between the gut and brain. It may instead to be more concrete in stating the predicted anatomical paths that gut to brain amyloid spread *may* occur by (for instance spinal or vagal neurons: PMIDs 31254094, 38063197), if this is what the authors refer to.

Reviewer #3 (Remarks to the Author):

I commend the authors on the elaborate revision of the manuscript and taking up the requested control experiments. The revised manuscript successfully address my previous points of concern.

The study brings important new insights to show that bacterial BAP-like sequences may have a PD-inducing potential. The authors show an association with BAP occurrence and PD disease status, though that does not in itself proof any causality. The manuscript also convincingly shows that BAP fragments can induce a-syn aggregation in vitro and in vivo, and that cerebral injection of BAP fibrils can induce PD symptoms. Whether intestinal BAP deposits can reach the brain, and do so in sufficient amounts to induce PD remains unclear. Nevertheless, this is an important first report in a possible BAP - PD link, and the authors' discussion is sufficiently balanced and cautious to not over interpret results.

REVIEWERS' COMMENTS

Reviewer #1 (Remarks to the Author):

The author's have thoughtfully addressed our comments. There are a few minor comments to consider.

We would like to thank reviewer for taking the time to review the revised version of the manuscript. We appreciate the reviewer's comments. We have addressed all the minor concerns raised by the reviewer.

Minor comments to the authors:

1. Page 3, line 15 change 'csgACB' to csgBAC

Author response:

Done

2. Page 27, line 30 change "bards" to bars.

Author response:

Done

3. Figure S6A What is the band between 35- 48 kDa in the control sample?

Author response:

This band correspond to lower molecular weight species that go down to a submonomeric form. These forms have been previously observed in *C. elegans* Model NL5901 (Punc-54:: α -syn::YFP) (Goya, M. E., et al. (2020). We have included a clarification in the figure caption.

4. In Figure S10 the blot using anti-PS a-Syn antibodies shows the presence of prominent high molecular weight (HMW) α -Syn ~ 135 kDa. However, the HMW band is absent in the blot probed with a-Syn antibodies. Is it known that phosphorylated α -Syn (P-S129) form SDS stable higher molecular weight species? Can the authors comment on that?

Author response:

We appreciate the observation made by the reviewer. For the detection of phosphorylated α -Syn (P-S129) we have used the commercial antibody ab51253. Antibody ab51253 shows high specificity for pS129- α Syn detection (Fayyad, M., et al., 2020). However, it has been reported to occasionally stain an unknown high-molecular weight protein that is not α -Syn (~100 KDa) in immunoblots (Delic, V., et al., 2018). Moreover, the unspecificity is supported by the lack of these bands in the blot probed with α -Syn antibodies. We modified the figure S10 accordingly.

References

Delic, V., Chandra, S., Abdelmotilib, H., Maltbie, T., Wang, S., Kem, D., et al. (2018). Sensitivity and specificity of phospho-Ser129 α -synuclein monoclonal antibodies. *Journal of Comparative Neurology*, 526(12), 1978–1990. doi.org/10.1002/cne.24468

Fayyad, M., Majbour, N. K., Vaikath, N. N., Erskine, D., El-Tarawneh, H., Sudhakaran, I. P., et al. (2020). Generation of monoclonal antibodies against phosphorylated α -Synuclein at serine 129: Research tools for synucleinopathies. *Neuroscience Letters*, 725, 134899. doi.org/10.1016/j.neulet.2020.134899

Goya, M. E., Xue, F., Sampedro-Torres-Quevedo, C., Arnaouteli, S., Riquelme-Dominguez, L., Romanowski, A., et al. (2020). Probiotic *Bacillus subtilis* Protects against α -Synuclein Aggregation in *C. elegans*. *CellReports*, 30(2), 367–380.e7. doi.org/10.1016/j.celrep.2019.12.078

Reviewer #2 (Remarks to the Author):

The authors should be commended on their detailed response and, to me, the manuscript is certainly in a significantly improved state. I do think this is an interesting study that will be useful for the field.

We thank the reviewer for these kind words. We are glad we have been able to provide satisfying answers to their questions and comments. We also believe the manuscript is certainly in a significantly improved state.

I have two *very* minor concerns that might be worthwhile to address prior to publication.

1) New Figure S11, demonstrating spread of BAPs to various brain regions following intra-striatal injection. 24hr spread seems very fast based on other reports (with synuclein PFFs), the authors could show some negative control for specificity of the anti-BAP staining in these tissues.

Author response:

We agree with the reviewer to include a control for specificity of the anti-BAP staining. For that, C57BL6/C3H F1 mice received intra-striatal injection of PBS or BapB-PFF. After 72 h post-injection, mice were sacrificed and striatal sections were stained with anti-Bap antibodies. Results showed the presence of dense aggregate signals in the brain of mice following intra-striatal injection of BapB-PFF but not in the brains of mice after injection of PBS. These results confirm the specificity of the anti-BAP staining in striatal sections. We have included these results as part of supplementary figure S11c.

Figure S11. Analysis of the presence of rBap-PFF at different brain regions. **c** Control for anti-Bap antibodies specificity. Mice that received PBS (Ø) by stereotaxic injection did not show a Bap-specific staining, while mice that received rBap-PFF showed clear Bap-specific staining.

2) In the rationale for using intra-striatal injections (rather than intestinal injections or exposures) the authors state: "Assuming that amyloids can migrate from the gut to the brain through the gut-brain axis, [...]" I would suggest revising the term "gut-brain axis" here. This general refers to a conceptual axis of vagal, spinal, hormonal, immune, and metabolic signals between the gut and brain. It may instead be more concrete in stating the predicted anatomical paths that gut to brain amyloid spread *may* occur by (for instance spinal or vagal neurons: PMIDs 31254094, 38063197), if this is what the authors refer too.

Author response:

Following the reviewer's suggestion, we have modified the text accordingly:

"Assuming that amyloids may migrate from the gut to the brain through anatomical paths, such as spinal or vagal neurons^{87, 88}, in this study we forced this scenario using direct intracerebral inoculation of BAP preformed amyloids (rBapB_PFF) in wild-type nontransgenic mice"

Reviewer #3 (Remarks to the Author):

I commend the authors on the elaborate revision of the manuscript and taking up the requested control experiments. The revised manuscript successfully address my previous points of concern.

The study brings important new insights to show that bacterial BAP-like sequences may have a PD-inducing potential. The authors show an association with BAP occurrence and PD disease status, though that does not in itself proof any causality. The manuscript also convincingly shows that BAP fragments can induce a-syn aggregation in vitro and in vivo, and that cerebral injection of BAP fibrils can induce PD symptoms. Whether intestinal BAP deposits can reach the brain, and do so in sufficient amounts to induce PD remains unclear. Nevertheless, this is an important first report in a possible BAP - PD link, and the authors' discussion is sufficiently balanced and cautious to not over interpret results.

We genuinely appreciate the reviewer's dedication to our work and thank for appreciating the significance of this work.